# 🦁 Linear Attention for Efficient Bidirectional Sequence Modeling

**Arshia Afzal**[*], **Elias Abad Rocamora**, **Leyla Naz Candogan**,
**Pol Puigdemont, Francesco Tonin, Yongtao Wu, Mahsa Shoaran,**
**Volkan Cevher**

**EPFL**

## Abstract

Linear Transformers and State Space Models have emerged as efficient alternatives to softmax Transformers for causal sequence modeling, enabling parallel training via matrix multiplication and efficient RNN-style inference. However, despite their success in causal tasks, no unified framework exists for applying Linear Transformers to bidirectional sequence modeling. We introduce LION, the first framework to systematically extend Linear Transformers to the bidirectional setting. LION generalizes three core representations commonly used in the causal case—full **Li**near Attenti**on**, bidirectional RNN, and chunkwise parallel form—to the bidirectional setting. These forms are theoretically equivalent and enable models to exploit the strengths of each during training and inference. We prove that a broad class of Linear Transformers can be extended using LION and validate our framework via three core examples based on the choice of decay type: LION-LIT, the bidirectional extension of [25]; LION-D, based on [44]; and LION-S, a variant using selective decay [34, 13]. Across standard bidirectional tasks, LION enables models to match or exceed the performance of softmax Transformers, while offering significantly faster training and more efficient inference than existing State Space Models.

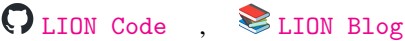 LION Code , 📚 LION Blog

## 1 Introduction

Softmax Transformers [48] are widely used in sequence modeling tasks such as causal language modeling [4, 11] due to their high performance and parallelized training. However, their quadratic computational cost is often limiting [45], increasing interest in Recurrent Neural Network (RNN)-like models for inference. Causal Linear Transformers addresses this by replacing softmax attention with linear attention, which is equivalent to an RNN with a two-dimensional hidden state [25]. Causal Linear Transformers enjoy fast training via matrix multiplication and efficient RNN-style inference.

To improve the performance of Linear Transformers, several variants of linear attention and state space models (SSMs) have been proposed by incorporating fixed [44, 34, 15] or input-dependent [13, 6, 51] decay factors into the RNN recurrence. These recent Linear Transformers match the performance of softmax Transformers in causal tasks, while retaining efficient and fast inference.

Several real-world tasks are inherently bidirectional and benefit from bidirectional sequence modeling. Examples include DNA and protein modeling [17], computer vision [39], and biological and chemical sequence modeling [42]. In these domains, bidirectional models often outperform causal ones, for

---

[*]***Corresponding author: arshia.afzal@epfl.ch**. Arshia Afzal, Elias Abad Rocamora, Leyla Naz Candogan, Pol Puigdemont, Francesco Tonin, Yongtao Wu, and Volkan Cevher are with LIONS@EPFL. Mahsa Shoaran is with Integrated Neurotechnologies Laboratory (INL)@EPFL.

instance, Bi-Mamba outperforms Mamba in DNA modeling [39]. This motivates the development of architectures specifically designed for bidirectional sequence modeling

*Unlike causal sequence modeling, transformers with linear attention remain largely unexplored for bidirectional sequence modeling.* Current bidirectional SSMs are primarily based on Mamba [53, 21] and are mostly designed for vision tasks [28, 21] and have not been generalized to the broader class of Linear Transformers. These models typically apply their causal forms in both forward and backward directions (e.g., dual scans in Vim or two separate SSDs in Hydra), which fails to leverage the natural priors of bidirectional modeling namely, the availability of the entire sequence during both training and inference. As a result, their training speed lags behind that of softmax Transformers [9, 18].

To show how Linear Transformers can be applied to bidirectional modeling and explore their capabilities, we present the LION framework, a general approach for extending Linear Transformers to bidirectional sequence modeling. LION supports three theoretically equivalent representations—full attention (for maximum training speed), bidirectional RNN (for memory-efficient inference), and chunkwise parallel form (balancing speed and memory)—mirroring the efficiency and speed advantages of Linear Transformers in the causal tasks.

We show that a wide range of Linear Transformers can be extended to bidirectional sequence modeling via the LION framework (Table 1). For experiments, we focus on three representative running examples that cover the different types of decay factors: *(a)* LION-LIT , extension of vanilla Linear Transformer without decay [25]; *(b)* LION-D , extension of RetNet with learnable, non-selective decay [44]; and *(c)* LION-S , which uses selective, input-dependent decay [34].

All LION models are **trained using full attention**, enabling the highest training speed among existing SSMs, and comparable to softmax Transformers. For inference, LION offers three representations: (1) RNN for maximum efficiency, (2) full attention for maximum speed, and (3) a chunkwise form to balance memory and speed. We validate these advantages through extensive experiments on standard bidirectional tasks across multiple model scales. LION enables training up to $10\times$ faster than SSMs (e.g., Vim [53]) while matching the performance of SSMs and softmax Transformers. It also achieves superior inference efficiency through its RNN form, as shown in Figure 3.

## 2 Background

**Notation.** Matrices (vectors) are denoted by uppercase (lowercase) boldface letters, such as $\mathbf{X}$ for matrices and $\mathbf{x}$ for vectors. Scalars are represented by lowercase letters, e.g., $x$ and $\odot$ is Hadamard product.

**Causal Linear Transformers: Transformers with Linear Attention**

Given a sequence $\mathbf{X} = [\mathbf{x}_1, \mathbf{x}_2, \ldots, \mathbf{x}_L]^\top \in \mathbb{R}^{L \times d}$, a single-head softmax attention is defined as:

$$(\mathbf{q}_i, \mathbf{k}_i, \mathbf{v}_i) = (\mathbf{x}_i \mathbf{W_q}, \mathbf{x}_i \mathbf{W_k}, \mathbf{x}_i \mathbf{W_v}), \quad \mathbf{y}_i = \sum_{j=1}^{i} \frac{\exp(\mathbf{q}_i^\top \mathbf{k}_j)}{\sum_{p=1}^{i} \exp(\mathbf{q}_i^\top \mathbf{k}_p)} \mathbf{v}_j, \tag{1}$$

where $\mathbf{x}_i, \mathbf{q}_i, \mathbf{k}_i, \mathbf{v}_i, \mathbf{y}_i \in \mathbb{R}^d$ and the weights $\mathbf{W_q}, \mathbf{W_k}, \mathbf{W_v} \in \mathbb{R}^{d \times d}$ with $d$ being the projection dimension. With $\mathbf{Q} := [\mathbf{q}_1, \ldots, \mathbf{q}_L]^\top$, $\mathbf{K} := [\mathbf{k}_1, \ldots, \mathbf{k}_L]^\top$, $\mathbf{V} := [\mathbf{v}_1, \ldots, \mathbf{v}_L]^\top \in \mathbb{R}^{L \times d}$, To enable parallel training, the attention output can be expressed in the following matrix form:

$$\mathbf{Y} = \text{softmax}\left(\mathbf{Q}\mathbf{K}^\top + \mathbf{M}^C\right)\mathbf{V}, \tag{2}$$

where $\mathbf{M}^C \in \{-\infty, 0\}^{L \times L}$ is a causal mask for preventing future tokens to attend to past. In contrast, equation 1 is used during inference for generating or processing tokens. However, for causal Transformers [27], employing equation 1 requires storing the previous $L$ tokens to attend to the latest token during inference (i.e., the KV cache). This approach is less efficient than RNNs, where only the state is stored regardless of the previous sequence (*cf.*, [32]).

Causal Linear Attention replaces the exponential kernel $\exp(\mathbf{q}_i^\top \mathbf{k}_j)$ with feature map function $\phi(\mathbf{q}_i)^\top \phi(\mathbf{k}_j)$ where $\phi(\cdot) : \mathbb{R}^n \to \mathbb{R}^d$ maps the input to a higher-dimensional space [25]. For simplicity of notation, we use $\mathbf{q}_i := \phi(\mathbf{W_q}\mathbf{x}_i)$ and $\mathbf{k}_i := \phi(\mathbf{W_k}\mathbf{x}_i)$ in the sequel. This formulation allows the linear transformer to be expressed as an RNN with linear recurrence [2], which eliminates the need

---

[2]This models with 2D hidden state also known as "fast weights" [20, 41] and their connection to transformers were explored in [40].

to store previous tokens in inference, while enabling parallelized training via matrix multiplication:

$$\mathbf{Y} = \text{SCALE}\left(\mathbf{Q}\mathbf{K}^\top \odot \mathbf{M}^C\right)\mathbf{V}, \quad \mathbf{S}_i = \mathbf{S}_{i-1} + \mathbf{k}_i\mathbf{v}_i^\top, \quad \mathbf{z}_i = \mathbf{z}_{i-1} + \mathbf{k}_i, \quad \mathbf{y}_i = \frac{\mathbf{q}_i^\top \mathbf{S}_i}{\mathbf{q}_i^\top \mathbf{z}_i} \quad (3)$$

Here, the causal mask $\mathbf{M}^C \in \{0,1\}^{L \times L}$ is a lower triangular matrix that enforces causal constraints. $\text{SCALE}(\cdot)$ denotes the scaling of the attention matrix across its rows ($\text{SCALE}(\mathbf{A})_{ij} = \frac{\mathbf{A}_{ij}}{\sum_{p=1}^{L} \mathbf{A}_{ip}}$) and $\mathbf{S}_i \in \mathbb{R}^{d \times d}$ and $\mathbf{z}_i \in \mathbb{R}^d$ are the hidden state matrix and the hidden state scaling vector. Eq. (3) can also written as $\mathbf{Y} = \mathbf{P}\mathbf{V}$ with $\mathbf{P} \in \mathbb{R}^{L \times L}$ known as sequence mixer [21].

To enhance the performance of linear attention and incorporate relative position encodings into the recurrence, decay factors ($\lambda_i$) have been introduced into the state update:

$$\mathbf{Y} = \text{SCALE}\left(\mathbf{Q}\mathbf{K}^\top \odot \mathbf{M}^C\right)\mathbf{V}, \quad \mathbf{S}_i = \lambda_i \mathbf{S}_{i-1} + \mathbf{k}_i\mathbf{v}_i^\top. \quad (4)$$

The mask $\mathbf{M}^C \in \mathbb{R}^{L \times L}$ is a lower-triangular mask generated based on the input-dependent (selective) parameter $\lambda_i$ as bellow:

$$\mathbf{M}_{ij}^C = \begin{cases} \Pi_{k=j+1}^{i} \lambda_k, & i \geq j; \\ 0, & i < j. \end{cases} \quad (5)$$

Several Linear Transformers have been developed based on equation 4 [5, 34, 3, 44, 51], and can be unified under the form:

$$\mathbf{S}_i = \boldsymbol{\Lambda}_i \mathbf{S}_{i-1} + \boldsymbol{\gamma}_i \mathbf{k}_i \mathbf{v}_i^\top, \quad \mathbf{z}_i = \boldsymbol{\Lambda}_i \mathbf{z}_{i-1} + \boldsymbol{\gamma}_i \mathbf{k}_i \quad \text{SCALED} : \mathbf{y}_i = \frac{\mathbf{q}_i^\top \mathbf{S}_i}{\mathbf{q}_i^\top \mathbf{z}_i}, \quad \text{NON-SCALED} : \mathbf{y}_i = \mathbf{q}_i^\top \mathbf{S}_i. \quad (6)$$

Here, $\boldsymbol{\Lambda}_i$ and $\boldsymbol{\gamma}_i$ denote model-specific decay factors and stabilizers. In this study, we focus on scalar or diagonal forms of $\boldsymbol{\Lambda}_i$, corresponding to the $\text{TC}^0$ class as stated in Merrill et al. [31], which encompasses most Linear Transformers. This unified view is also referred to as Linear Recurrent Models [52].

**Chunkwise parallel form of causal Linear Transformers**

Causal Linear Transformers balance the memory-speed tradeoff during training by employing chunkwise parallelization [51, 52, 6]. In this approach, the input sequence $\mathbf{X} \in \mathbb{R}^{L \times d}$ and the corresponding query, key, and value vectors are split into $\frac{L}{C}$ non-overlapping chunks, each of size $C$.

Let $\mathbf{Q}_{[i]}, \mathbf{K}_{[i]}, \mathbf{V}_{[i]} \in \mathbb{R}^{C \times d}$ represent the chunked query, key, and value matrices, and define the chunk-level hidden state after processing $i$ chunks as $\mathbf{S}_{[i]} \in \mathbb{R}^{d \times d}$. The causal sequence mixer which maps the input to output $\mathbf{Y} = \mathbf{P}\mathbf{V}$ is expressed as:

$$\mathbf{S}_{[i]} = \mathbf{S}_{[i-1]} + \underbrace{\sum_{j=iC+1}^{i(C+1)} \mathbf{k}_j^\top \mathbf{v}_j}_{\mathbf{K}_{[i]}^\top \mathbf{V}_{[i]}} \quad (7)$$

$$\mathbf{Y}_{[i]} = \underbrace{\mathbf{Q}_{[i]}\mathbf{S}_{[i]}}_{\text{inter}} + \underbrace{\left(\mathbf{Q}_{[i]}\mathbf{K}_{[i]}^\top \odot \mathbf{M}^C\right)\mathbf{V}_{[i]}}_{\text{intra}}$$

$$\mathbf{Y} = \mathbf{P}\mathbf{V}$$

The above form is the chunkwise form of Linear Transformer without decay factor and the mask $\mathbf{M}^C \in \{0,1\}^{L \times L}$ is the causal mask. Chunking is essential for training Linear Transformers and SSMs with decay factors, such as Mamba-2 [6] and GLA [51] even more, since treating the entire sequence mixer $\mathbf{P}$ as a single intra-chunk block, is numerically unstable. This instability arises from computing the cumulative product of decay factors $\prod_{t=1}^{L} \boldsymbol{\Lambda}_t$ and their inverse $\prod_{t=1}^{L} \boldsymbol{\Lambda}_t^{-1}$ over the entire sequence, which can overflow or underflow even in logarithmic space[3]. Consequently, SSMs like Mamba-2 [6] need chunking for training stability, which limits their train speed, particularly for short sequences [26]. Chunkwise form has a complexity of $O(LCd)$ and requires $O\left(\frac{L}{C}\right)$ sequential steps, as opposed to RNNs with a complexity of $O(L)$ and $O(L)$ steps, and attention with $O(L^2)$ complexity and $O(1)$ steps [52].

## 3  LION: Unified framework for bidirectional Linear Transformers

In this section, we introduce the full linear attention (Eq. (8)), its equivalent bidirectional RNN form (Eq. (19)), and the chunkwise formulation (Eq. (21)). These are presented together to highlight their

---

[3]In log space, $\prod_{t=1}^{L} \boldsymbol{\Lambda}_t^{-1} = \exp\left(-\sum_{t=1}^{L} \log(\boldsymbol{\Lambda}_t)\right)$, yet instability persists, as discussed in the official Mamba2 blog post here. Chunking is therefore required for stable training.

equivalence and demonstrate the range of inference strategies supported by our framework. We begin with the scalar decay case due to its simplicity and effectiveness.

## 3.1 LION **Full Linear Attention**

In bidirectional tasks, the entire sequence is available during both training and inference. For softmax-based attention, this results in the form:

$$\mathbf{Y} = \text{softmax}\left(\mathbf{Q}\mathbf{K}^{\top}\right)\mathbf{V}$$

similar to Eq. (2) but without causal masking. Inspired by the natural formulation of full softmax attention and casual Linear Attention Eq. (4), we formulate the Full Linear Attention as:

$$\mathbf{Y} = \text{SCALE}\left(\mathbf{Q}\mathbf{K}^{\top} \odot \mathbf{M}\right)\mathbf{V} \tag{8}$$

The key construction part is to define the bidirectional mask $\mathbf{M}$ in a way that mirrors its properties in the causal setting. To motivate this, we begin with a remark on the structure of $\mathbf{M}^{C}$ in causal setting:

**Remark.** Causal Linear Transformers encode relative positional information through their decay factors, which is reflected in their causal masks $\mathbf{M}^{C}$ (Eq. (4)) [44, 6]. Specifically, the causal mask between tokens $i$ and $j$ is given by $\mathbf{M}_{ij}^{C} = \lambda_{j+1}\lambda_{j+2}\ldots\lambda_{i}$, representing the product of all decays between positions $j$ and $i$ ($i \geq j$).

Inspired by the above remark, we define the mask $\mathbf{M}$ for bidirectional sequence modeling such that $\mathbf{M}_{ij}$ equals the product of all decay factors between tokens $i$ and $j$. Formally, the selective $\boxed{\mathbf{M}}$, learnable fixed decay $\boxed{\mathbf{M}}$, and all-ones $\boxed{\mathbf{M}}$ masks can be written as:

$$\mathbf{M}_{ij} = \begin{cases} \Pi_{k=j}^{i-1}\lambda_{k}, & i > j \\ 1 & i = j \\ \Pi_{k=i+1}^{j}\lambda_{k}, & i < j. \end{cases} \qquad \mathbf{M}_{ij} = \lambda^{|i-j|}, \qquad \mathbf{M}_{ij} = 1. \tag{9}$$

Therefore, in the most general form (selective decay), the output of Full Linear Attention is:

$$\mathbf{Y} = \text{SCALE}\left(\underbrace{\begin{pmatrix} \mathbf{q}_1^{\top}\mathbf{k}_1 & \mathbf{q}_1^{\top}\mathbf{k}_2 & \cdots & \mathbf{q}_1^{\top}\mathbf{k}_L \\ \mathbf{q}_2^{\top}\mathbf{k}_1 & \mathbf{q}_2^{\top}\mathbf{k}_2 & \cdots & \mathbf{q}_2^{\top}\mathbf{k}_L \\ \vdots & \vdots & \ddots & \vdots \\ \mathbf{q}_L^{\top}\mathbf{k}_1 & \mathbf{q}_L^{\top}\mathbf{k}_2 & \cdots & \mathbf{q}_L^{\top}\mathbf{k}_L \end{pmatrix}}_{\mathbf{A} = \mathbf{Q}\mathbf{K}^{\top}} \odot \underbrace{\begin{pmatrix} 1 & \lambda_2 & \lambda_2\lambda_3 & \cdots & \lambda_2\cdots\lambda_L \\ \lambda_1 & 1 & \lambda_3 & \cdots & \lambda_3\cdots\lambda_L \\ \lambda_1\lambda_2 & \lambda_2 & 1 & \cdots & \lambda_4\cdots\lambda_L \\ \vdots & \vdots & \vdots & \ddots & \vdots \\ \lambda_{L-1}\cdots\lambda_1 & \lambda_{L-1}\cdots\lambda_2 & \lambda_{L-1}\cdots\lambda_3 & \cdots & 1 \end{pmatrix}}_{\mathbf{M}}\right)\begin{pmatrix} \mathbf{v}_1^{\top} \\ \mathbf{v}_2^{\top} \\ \mathbf{v}_3^{\top} \\ \vdots \\ \mathbf{v}_L^{\top} \end{pmatrix}, \tag{10}$$

where we use $\boxed{\phantom{x}}$ for upper triangular elements, $\boxed{\phantom{x}}$ for lower triangular elements, and $\boxed{\phantom{x}}$ for the diagonal of the attention matrix and mask. The selective and fixed learnable masks in Eq. (9) have structured forms enabling efficient implementation via matrix multiplications. For the vanilla Linear Transformer [25], due to the absence of decay factor, the mask simplifies to an all-ones matrix and can be omitted.

**Selective Mask:** To build the selective mask $\mathbf{M}$, we decompose it into lower and upper triangular components (including the diagonal), denoted by $\mathbf{M}^{F}$ and $\mathbf{M}^{B}$, respectively. Since both masks are rank-1 semi-separable (see proof in Appendix B.8) [21], they can be constructed efficiently via matrix multiplication. In the bidirectional setting, where all decay factors $\lambda_i$ are known, we stack them into a vector $\boldsymbol{\lambda} \in \mathbb{R}^{L}$, analogous to stacking queries and keys into $\mathbf{Q}$ and $\mathbf{K}$. The cumulative product $\mathbf{L}^{F} = \text{cumprod}(\boldsymbol{\lambda})$, where $\mathbf{L}_i^{F} = \prod_{k=0}^{i}\lambda_k$, is used to construct the lower triangular mask $\mathbf{M}^{F}$. For the upper triangular mask, we flip the decay vector be reversing the order of the sequence, and compute $\mathbf{L}^{B} = \text{cumprod}(\text{FLIP}(\boldsymbol{\lambda}))$. The two masks are then given by:

$$\mathbf{M}^{F} = \text{Tril}(\mathbf{L}^{F}\frac{1}{\mathbf{L}^{F\top}}), \quad \mathbf{M}^{B} = \text{Triu}(\mathbf{L}^{B}\frac{1}{\mathbf{L}^{B\top}}), \tag{11}$$

where Tril($\cdot$) and Triu($\cdot$) extract the lower and upper triangular parts, respectively.

For numerical stability, we compute the masks in log-space by defining $\mathbf{a} = \log(\boldsymbol{\lambda})$, where $\mathbf{a} \in \mathbb{R}^L$ originates from Zero-Order Hold (ZOH) discretization (see Appendix B.11). The cumulative products $\mathbf{L}^F$ and $\mathbf{L}^B$ can then be computed as cumulative sums in log-space, followed by exponentiation:

$$\mathbf{D}^F = \texttt{cumsum}(\mathbf{a}), \mathbf{D}^B = \texttt{cumsum}(\text{FLIP}(\mathbf{a})), \quad \mathbf{L}^F = \exp(\mathbf{D}^F), \mathbf{L}^B = \exp(\mathbf{D}^F)$$

The full selective mask (Eq. (9)) is then assembled as $\boxed{\mathbf{M}} = \mathbf{M}^F + \mathbf{M}^B - \mathbf{I}$, as shown in Eq. (13).

**Learnable Fixed Mask:** For learnable fixed decay factor, the mask $\boxed{\mathbf{M}}$ forms a Kac–Murdock–Szegö (KMS) matrix [23], and stable when $0 < \lambda < 1$. Our codes for generating the masks are provided in Appendix C.17).

**All-ones Mask:** As for the case of vanilla Linear Transformer, due to the absence of decay ($\lambda = 1$), the mask is omitted and simplified to $\boxed{\mathbf{M}} = 1$.

### 3.2 LION **RNN: Equivalent bi-directional RNN for Full Linear Attention**

We first show that the naive summation of two separate linear transformers suffers from double counting and imbalanced attention. Considering two linear transformers, we have:

$$\mathbf{S}_i^F = \sum_{j=1}^{i} \mathbf{k}_j \mathbf{v}_j^\top, \quad \mathbf{S}_i^B = \sum_{j=i}^{L} \mathbf{k}_j \mathbf{v}_j^\top, \quad \mathbf{y}_i = \mathbf{q}_i^\top \mathbf{S}_i^F + \mathbf{q}_i^\top \mathbf{S}_i^B = \mathbf{q}_i^\top (\mathbf{k}_i \mathbf{v}_i^\top + \sum_{j=1}^{L} \mathbf{k}_j \mathbf{v}_j^\top).$$

This summation causes *imbalanced attention*, i.e., $\mathbf{Y} = ((\mathbf{I} + \mathbf{1}) \odot \mathbf{QK}^\top)\mathbf{V}$, due to double counting of the diagonal and underperforms balanced version as shown in Table 16. To address this, we construct a bidirectional RNN formulation for balanced attention (Eq. (55)), by decomposing the attention matrix $\mathbf{A}$ and mask $\mathbf{M}$ into lower and upper triangular parts:

$$\underbrace{\begin{pmatrix} \mathbf{q}_1^\top \mathbf{k}_1 & \mathbf{q}_1^\top \mathbf{k}_2 & \cdots & \mathbf{q}_1^\top \mathbf{k}_L \\ \mathbf{q}_2^\top \mathbf{k}_1 & \mathbf{q}_2^\top \mathbf{k}_2 & \cdots & \mathbf{q}_2^\top \mathbf{k}_L \\ \vdots & \vdots & \ddots & \vdots \\ \mathbf{q}_L^\top \mathbf{k}_1 & \mathbf{q}_L^\top \mathbf{k}_2 & \cdots & \mathbf{q}_L^\top \mathbf{k}_L \end{pmatrix}}_{\mathbf{A} = \mathbf{QK}^\top} = \underbrace{\begin{pmatrix} \frac{1}{2}\mathbf{q}_1^\top \mathbf{k}_1 & & & \\ \mathbf{q}_2^\top \mathbf{k}_1 & \frac{1}{2}\mathbf{q}_2^\top \mathbf{k}_2 & & \\ \vdots & \vdots & \ddots & \\ \mathbf{q}_L^\top \mathbf{k}_1 & \mathbf{q}_L^\top \mathbf{k}_2 & \cdots & \frac{1}{2}\mathbf{q}_L^\top \mathbf{k}_L \end{pmatrix}}_{\mathbf{A}^F} + \underbrace{\begin{pmatrix} \frac{1}{2}\mathbf{q}_1^\top \mathbf{k}_1 & \mathbf{q}_1^\top \mathbf{k}_2 & \cdots & \mathbf{q}_1^\top \mathbf{k}_L \\ & \frac{1}{2}\mathbf{q}_2^\top \mathbf{k}_2 & \cdots & \mathbf{q}_2^\top \mathbf{k}_L \\ & & \ddots & \vdots \\ & & & \frac{1}{2}\mathbf{q}_L^\top \mathbf{k}_L \end{pmatrix}}_{\mathbf{A}^B} \quad (12)$$

$$\underbrace{\begin{pmatrix} 1 & \lambda_2 & \lambda_2\lambda_3 & \cdots & \lambda_2\cdots\lambda_L \\ \lambda_1 & 1 & \lambda_3 & \cdots & \lambda_3\cdots\lambda_L \\ \lambda_1\lambda_2 & \lambda_2 & 1 & \cdots & \lambda_4\cdots\lambda_L \\ \vdots & \vdots & \vdots & \ddots & \vdots \\ \lambda_{L-1}\cdots\lambda_1 & \lambda_{L-1}\cdots\lambda_2 & \lambda_{L-1}\cdots\lambda_3 & \cdots & 1 \end{pmatrix}}_{\mathbf{M}} = \underbrace{\begin{pmatrix} 1 & & & & \\ \lambda_1 & 1 & & & \\ \lambda_1\lambda_2 & \lambda_2 & 1 & & \\ \vdots & \vdots & \vdots & \ddots & \\ \lambda_{L-1}\cdots\lambda_1 & \lambda_{L-1}\cdots\lambda_2 & \lambda_{L-1}\cdots\lambda_3 & \cdots & 1 \end{pmatrix}}_{\mathbf{M}^F} + \underbrace{\begin{pmatrix} 1 & \lambda_2 & \lambda_2\lambda_3 & \cdots & \lambda_2\cdots\lambda_L \\ & 1 & \lambda_3 & \cdots & \lambda_3\cdots\lambda_L \\ & & 1 & \cdots & \lambda_4\cdots\lambda_L \\ & & & \ddots & \vdots \\ & & & & 1 \end{pmatrix}}_{\mathbf{M}^B} - \mathbf{I} \quad (13)$$

As in Eq. (12) and Eq. (13), the attention matrix and mask are split into lower ($\mathbf{A}^F, \mathbf{M}^F$) and upper triangular ($\mathbf{A}^B, \mathbf{M}^B$) matrices. The scaling operator divides each row of the attention matrix to its summed value, and hence equals to a diagonal matrix $\mathbf{C}^{-1}$ multiplied by the attention:

$$\mathbf{Y} = \big(\text{SCALE}(\mathbf{QK}^\top \odot \mathbf{M})\big)\mathbf{V} = (\mathbf{C}^{-1}(\mathbf{QK}^\top \odot \mathbf{M}))\mathbf{V}, \ \mathbf{C}_i = \mathbf{q}_i^\top \sum_{j=1}^{L} \mathbf{M}_{ij}\mathbf{k}_j. \quad (14)$$

Decomposing $\mathbf{C}$ into causal and non-causal parts as $\mathbf{C}_i = \mathbf{q}_i^\top \sum_{j=1}^{i} \mathbf{M}_{ij}\mathbf{k}_j + \mathbf{q}_i^\top \sum_{j=i}^{L} \mathbf{M}_{ij}\mathbf{k}_j - \mathbf{q}_i^\top \mathbf{k}_i$, we can similarly split the scaling matrix into two parts as:

$$\boxed{\mathbf{C}_i} = \underbrace{\mathbf{q}_i^\top \sum_{j=1}^{i} \mathbf{M}_{ij}\mathbf{k}_j - \frac{1}{2}\mathbf{q}_i^\top \mathbf{k}_i}_{\mathbf{C}_i^F} + \underbrace{\mathbf{q}_i^\top \sum_{j=i}^{L} \mathbf{M}_{ij}\mathbf{k}_j - \frac{1}{2}\mathbf{q}_i^\top \mathbf{k}_i}_{\mathbf{C}_i^B}$$

Since $\mathbf{A} = \mathbf{A}^F + \mathbf{A}^B$ and $\mathbf{M} = \mathbf{M}^F + \mathbf{M}^B - \mathbf{I}$, we can rewrite the attention output as:

$$\mathbf{Y} = \big(\text{SCALE}(\mathbf{QK}^\top \odot \mathbf{M})\big)\mathbf{V} = (\mathbf{C}^{-1}(\mathbf{QK}^\top \odot \mathbf{M}))\mathbf{V}$$

$$= (\boxed{\mathbf{C}^F} + \boxed{\mathbf{C}^B})^{-1}\big((\boxed{\mathbf{A}^F} + \boxed{\mathbf{A}^B}) \odot (\boxed{\mathbf{M}^F} + \boxed{\mathbf{M}^B} - \mathbf{I})\big)\mathbf{V} \quad (15)$$

$$= (\mathbf{C}^F + \mathbf{C}^B)^{-1}\big(\mathbf{A}^F \odot \mathbf{M}^F + \mathbf{A}^F \odot \mathbf{M}^B + \mathbf{A}^B \odot \mathbf{M}^F + \mathbf{A}^B \odot \mathbf{M}^B - \mathbf{A}^F \odot \mathbf{I} - \mathbf{A}^B \odot \mathbf{I}\big)\mathbf{V}.$$

Since the forward and backward recurrence matrices ($\mathbf{A}^F, \mathbf{A}^B$ for attention and $\mathbf{M}^F, \mathbf{M}^B$ for mask)

only share the diagonal with each other, and the diagonal of both forward and backward recurrence masks consists entirely of ones, we can simplify the above equation as follows:

$$\mathbf{Y} = (\mathbf{C}^F + \mathbf{C}^B)^{-1}\big(\mathbf{A}^F \odot \mathbf{M}^F + \underbrace{\mathbf{A}^F \odot \mathbf{M}^B}_{\mathbf{A}^F \odot \mathbf{I}} + \underbrace{\mathbf{A}^B \odot \mathbf{M}^F}_{\mathbf{A}^B \odot \mathbf{I}} + \mathbf{A}^B \odot \mathbf{M}^B - \mathbf{A}^F \odot \mathbf{I} - \mathbf{A}^B \odot \mathbf{I}\big)\mathbf{V}$$

$$= (\ \mathbf{C}^F + \mathbf{C}^B\ )^{-1}\big(\ \underbrace{(\mathbf{A}^F \odot \mathbf{M}^F)\mathbf{V}}_{\text{FORWARD}} + \underbrace{(\mathbf{A}^B \odot \mathbf{M}^B)\mathbf{V}}_{\text{BACKWARD}}\ \big). \tag{16}$$

As seen from Eq. (4), the FORWARD part above can be expressed as an RNN. We now demonstrate that the BACKWARD recurrence term can also be represented by the same RNN in reverse. We re-write the Eq. (16) by flipping the vector $\mathbf{V}$ as:

$$\underbrace{\begin{pmatrix} \frac{1}{2}\frac{\mathbf{q}_L^\top \mathbf{k}_L}{\mathbf{q}_L^\top \mathbf{z}_L} & & & \\ \frac{\mathbf{q}_{L-1}^\top \mathbf{k}_L}{\mathbf{q}_2^\top \mathbf{z}_L} & \frac{1}{2}\frac{\mathbf{q}_{L-1}^\top \mathbf{k}_{L-1}}{\mathbf{q}_2^\top \mathbf{z}_L} & & \\ \vdots & \vdots & \ddots & \\ \frac{\mathbf{q}_1^\top \mathbf{k}_L}{\mathbf{q}_1^\top \mathbf{z}_L} & \frac{\mathbf{q}_1^\top \mathbf{k}_{L-1}}{\mathbf{q}_1^\top \mathbf{z}_L} & \cdots & \frac{1}{2}\frac{\mathbf{q}_1^\top \mathbf{k}_1}{\mathbf{q}_1^\top \mathbf{z}_L} \end{pmatrix}}_{F(\mathbf{A}^B)} \odot \underbrace{\begin{pmatrix} 1 & & & \\ \lambda_L & 1 & & \\ \lambda_L\lambda_{L-1} & \lambda_{L-1} & 1 & \\ \vdots & \vdots & \vdots & \ddots \\ \lambda_L\cdots\lambda_2 & \lambda_L\cdots\lambda_3 & \lambda_L\cdots\lambda_4 & \cdots & 1 \end{pmatrix}}_{F(\mathbf{M}^B)} \begin{pmatrix} \mathbf{v}_L^\top \\ \mathbf{v}_{L-1}^\top \\ \mathbf{v}_{L-2}^\top \\ \vdots \\ \mathbf{v}_1^\top \end{pmatrix} \tag{17}$$

Above is the exact representations for the forward pass, as shown in Eq. (16), but with the tokens in reverse order. The matrices $\mathbf{A}^B$ and $\mathbf{M}^B$ are also modified to match the final flipped output using flipped input values $\mathbf{V}$ using functions $F(\mathbf{X}) = \mathbf{J}_L\mathbf{X}\mathbf{J}_L$ and $\text{FLIP}(\mathbf{X}) = \mathbf{J}_L\mathbf{X}$, where $\mathbf{J}_L$ is an $L$-dimensional exchange matrix, as detailed in Appendix B.4. Thus, the outputs of the forward and backward recurrences can be expressed as follows:

$$\mathbf{Y} = (\mathbf{C}^F + \mathbf{C}^B)^{-1}(\ \mathbf{Y}^F + \mathbf{Y}^B\ ), \text{ where} \tag{18}$$

$$\mathbf{Y}^F = (\mathbf{A}^F \odot \mathbf{M}^F)\mathbf{V}, \quad \mathbf{Y}^B = (\mathbf{A}^B \odot \mathbf{M}^B)\mathbf{V} = \text{FLIP}\Big(\big(F(\mathbf{A}^B) \odot F(\mathbf{M}^B)\big)\text{FLIP}(\mathbf{V})\Big).$$

**Theorem 3.1.** *(LION-RNN) Since Eq. (10) is the parallel form of the recurrence presented in Eq. (6), we can therefore express the equivalent recurrence for full attention Eq. (10) as follows:*

$$\begin{array}{ll}
\mathbf{S}_i^{F/B} = \lambda_i \mathbf{S}_{i-1}^{F/B} + \mathbf{k}_i \mathbf{v}_i^\top, & \text{(19a)} \qquad \mathbf{y}_i^{F/B} = \mathbf{q}_i^\top \mathbf{S}_i^{F/B} - \frac{\mathbf{q}_i^\top \mathbf{k}_i}{2}\mathbf{v}_i, \qquad \text{(20a)} \\[2mm]
\mathbf{z}_i^{F/B} = \lambda_i \mathbf{z}_{i-1}^{F/B} + \mathbf{k}_i, & \text{(19b)} \\[2mm]
c_i^{F/B} = \mathbf{q}_i^\top \mathbf{z}_i^{F/B} - \frac{\mathbf{q}_i^\top \mathbf{k}_i}{2}, & \text{(19c)} \qquad \text{OUTPUT: } \mathbf{y}_i = \frac{\mathbf{y}_i^F + \mathbf{y}_i^B}{c_i^F + c_i^B} \qquad \text{(20b)}
\end{array}$$

*The terms $\frac{1}{2}\mathbf{q}_i^\top \mathbf{k}_i \mathbf{v}_i$ and $\frac{1}{2}\mathbf{q}_i^\top \mathbf{k}_i$ are subtracted to avoid double counting.*

Several Linear Transformers can be generalized within our framework as shown in Table 1, where many causal recurrent models we adapt to the bidirectional setting. Since evaluating all models at all scales is infeasible, we focus on three representative examples based on the choice of decay $\lambda_i$:

- LION-LIT for $\lambda_i = 1$ which is bi-directional form of Vanilla Linear Transformer [25].

- LION-D for $\lambda_i = \lambda$ fixed decay, and bi-directional form of RetNet (with scaling) [44].

- LION-S , where $\lambda_i = \sigma(\mathbf{W}\mathbf{x}_i + b)$ is the bidirectional extension of GRFA [34] (with shifted SiLU activation) and also inspired by the selectivity of Mamba2.

For all three variants: LION-LIT, LION-D, and LION-S, we use *shifted and normalized SiLU activation* defined as $\phi(\mathbf{x}) = \frac{\text{SiLU}(\mathbf{x})+0.5}{\|\text{SiLU}(\mathbf{x})+0.5\|}$ [19]. To ensure stability in the recurrence ($0 < \lambda_i \leq 1$) [32], we set $\lambda_i = \sigma(\mathbf{W}_a^\top \mathbf{x}_i + b)$ for LION-S and $\lambda = \sigma(a)$ for LION-D, with $\sigma(\cdot)$ as the sigmoid and $a$ a learnable scalar. An ablation study on different choices is provided at Appendix C.3.

### 3.3 LION-chunk: Chunkwise parallel form of Full Linear Attention

Having established the Full Linear Attention (highest speed) and RNN (highest efficiency) representations, in this section we build the **chunkwise parallel form** form, which balances memory and speed. In the bidirectional case, chunkwise full attention only contains inter chunk leading to:

Table 1: Mapping Linear Transformers to their bidirectional forms using LION ▢, with both Full Linear Attention and equivalent bidirectional RNN representations. Proofs are provided in Appendix B.6. While RetNet does not originally include scaling, we include it for completeness. LION-S closely follows GRFA, differing only in the non-linearity $\phi(\cdot)$. Even non-diagnal models such as DeltaNet can exploit LION framework as detailed in Appendix B.7.

| Model | Causal Recurrence | LION Bi-directional RNN | LION Full Linear Attention |
|---|---|---|---|
| LinAtt [25] | $\mathbf{S}_i = \mathbf{S}_{i-1} + \mathbf{k}_i \mathbf{v}_i^\top$ 
 $\mathbf{y}_i = \mathbf{q}_i^\top \mathbf{S}_i$ | $\mathbf{S}_i^{F/B} = \mathbf{S}_{i-1}^{F/B} + \mathbf{k}_i \mathbf{v}_i^\top$ 
 $\mathbf{y}_i^{F/B} = \mathbf{q}_i^\top(\mathbf{S}_i^{F/B} - \frac{1}{2}\mathbf{k}_i\mathbf{v}_i)$, $\quad \mathbf{y}_i = \mathbf{y}_i^F + \mathbf{y}_i^B$ | $\mathbf{Y} = \mathbf{Q}\mathbf{K}^\top \mathbf{V}$ |
| + Scaling | $\mathbf{z}_i = \mathbf{z}_{i-1} + \mathbf{k}_i$ 
 $\mathbf{y}_i = \mathbf{q}_i^\top \mathbf{S}_i / \mathbf{q}_i^\top \mathbf{z}_i$ | $\mathbf{z}_i^{F/B} = \mathbf{z}_{i-1}^{F/B} + \mathbf{k}_i$, $\quad c_i^{F/B} = \mathbf{q}_i^\top(\mathbf{z}_i^{F/B} - \frac{1}{2}\mathbf{k}_i)$ 
 $\mathbf{y}_i^{F/B} = \mathbf{q}_i^\top(\mathbf{S}_i^{F/B} - \frac{1}{2}\mathbf{k}_i\mathbf{v}_i)$, $\quad \mathbf{y}_i = \frac{\mathbf{y}_i^F + \mathbf{y}_i^B}{c_i^F + c_i^B}$ LION-LIT | $\mathbf{Y} = \text{SCALE}(\mathbf{Q}\mathbf{K}^\top)\mathbf{V}$ |
| RetNet [44] | $\mathbf{S}_i = \lambda \mathbf{S}_{i-1} + \mathbf{k}_i \mathbf{v}_i^\top$ 
 $\mathbf{y}_i = \mathbf{q}_i^\top \mathbf{S}_i$ | $\mathbf{S}_i^{F/B} = \lambda \mathbf{S}_{i-1}^{F/B} + \mathbf{k}_i\mathbf{v}_i^\top$ 
 $\mathbf{y}_i^{F/B} = \mathbf{q}_i^\top(\mathbf{S}_i^{F/B} - \frac{1}{2}\mathbf{k}_i\mathbf{v}_i)$, $\quad \mathbf{y}_i = \mathbf{y}_i^F + \mathbf{y}_i^B$ | $\mathbf{Y} = (\mathbf{Q}\mathbf{K}^\top \odot \mathbf{M})\mathbf{V}$, 
 $\mathbf{M}_{ij} = \lambda^{|i-j|}$ |
| + Scaling | $\mathbf{z}_i = \lambda \mathbf{z}_{i-1} + \mathbf{k}_i$ 
 $\mathbf{y}_i = \mathbf{q}_i^\top \mathbf{S}_i / \mathbf{q}_i^\top \mathbf{z}_i$ | $\mathbf{z}_i^{F/B} = \lambda \mathbf{z}_{i-1}^{F/B} + \mathbf{k}_i$, $\quad c_i^{F/B} = \mathbf{q}_i^\top(\mathbf{z}_i^{F/B} - \frac{1}{2}\mathbf{k}_i)$ 
 $\mathbf{y}_i^{F/B} = \mathbf{q}_i^\top(\mathbf{S}_i^{F/B} - \frac{1}{2}\mathbf{k}_i\mathbf{v}_i)$, $\quad \mathbf{y}_i = \frac{\mathbf{y}_i^F + \mathbf{y}_i^B}{c_i^F + c_i^B}$ LION-D | $\mathbf{Y} = \text{SCALE}(\mathbf{Q}\mathbf{K}^\top \odot \mathbf{M})\mathbf{V}$, 
 $\mathbf{M}_{ij} = \lambda^{|i-j|}$ |
| Gated RFA [34] | $\mathbf{S}_i = \sigma(\mathbf{W}\mathbf{x}_i)\mathbf{S}_{i-1} + \mathbf{k}_i \mathbf{v}_i^\top$ 
 $\mathbf{z}_i = \sigma(\mathbf{W}\mathbf{x}_i)\mathbf{z}_{i-1} + \mathbf{k}_i$ 
 $\mathbf{y}_i = \mathbf{q}_i^\top \mathbf{S}_i / \mathbf{q}_i^\top \mathbf{z}_i$ | $\mathbf{S}_i^{F/B} = \sigma(\mathbf{W}\mathbf{x}_i)\mathbf{S}_{i-1}^{F/B} + \mathbf{k}_i \mathbf{v}_i^\top$ 
 $\mathbf{y}_i = \mathbf{q}_i^\top \mathbf{S}_i$ 
 $\mathbf{z}_i^{F/B} = \sigma(\mathbf{W}\mathbf{x}_i)\mathbf{z}_i^{F/B} + \mathbf{k}_i$, $\quad c_i^{F/B} = \mathbf{q}_i^\top(\mathbf{z}_i^{F/B} - \frac{1}{2}\mathbf{k}_i)$ 
 $\mathbf{y}_i^{F/B} = \mathbf{q}_i^\top(\mathbf{S}_i^{F/B} - \frac{1}{2}\mathbf{k}_i\mathbf{v}_i)$, $\quad \mathbf{y}_i = \frac{\mathbf{y}_i^F + \mathbf{y}_i^B}{c_i^F + c_i^B}$ LION-S | $\mathbf{Y} = \text{SCALE}(\mathbf{Q}\mathbf{K}^\top \odot \mathbf{M}^*)\mathbf{V}$, 
 $\mathbf{M}^*_{ij} = \begin{cases} \Pi_i^{j+1}\lambda_k & \text{if } i > j, \\ \Pi_{i+1}^j \lambda_k & \text{if } i < j, \\ 1 & \text{if } i = j, \end{cases}$ |
| Mamba-2 [6] | $\mathbf{S}_i = \lambda_t \mathbf{S}_{i-1} + \mathbf{k}_i \mathbf{v}_i^\top$ 
 $\mathbf{y}_i = \mathbf{q}_i^\top \mathbf{S}_i$ | $\mathbf{S}_i^{F/B} = \lambda_t \mathbf{S}_{i-1}^{F/B} + \mathbf{k}_i\mathbf{v}_i^\top$ 
 $\mathbf{y}_i^{F/B} = \mathbf{q}_i^\top(\mathbf{S}_i^{F/B} - \frac{1}{2}\mathbf{k}_i\mathbf{v}_i)$, $\quad \mathbf{y}_i = \mathbf{y}_i^F + \mathbf{y}_i^B$ | $\mathbf{Y} = (\mathbf{Q}\mathbf{K}^\top \odot \mathbf{M})\mathbf{V}$ 
 $\mathbf{M} = \mathbf{M}^*$ |
| RWKV-6[33] 
 Mamba [13] 
 GLA [51] | $\mathbf{S}_i = \text{Diag}(\lambda_i)\mathbf{S}_{i-1} + \mathbf{k}_i \mathbf{v}_i^\top$ 
 $\mathbf{y}_i = \mathbf{q}_i^\top \mathbf{S}_i$ | $\mathbf{S}_i^{F/B} = \text{Diag}(\lambda_i)\mathbf{S}_{i-1}^{F/B} + \mathbf{k}_i\mathbf{v}_i^\top$ 
 $\mathbf{y}_i^{F/B} = \mathbf{q}_i^\top(\mathbf{S}_i^{F/B} - \frac{1}{2}\mathbf{k}_i\mathbf{v}_i)$, $\quad \mathbf{y}_i = \mathbf{y}_i^F + \mathbf{y}_i^B$ | $\mathbf{Y}^{\mathbf{F}} = \text{Tril}(\mathbf{Q} \odot \mathbf{L}^{\mathbf{F}})(\mathbf{K} \odot (\mathbf{L}^{\mathbf{F}})^{-1})\mathbf{V}$ 
 $\mathbf{Y}^{\mathbf{B}} = \text{Triu}(\mathbf{Q} \odot \mathbf{L}^{\mathbf{B}})(\mathbf{K} \odot (\mathbf{L}^{\mathbf{B}})^{-1})\mathbf{V}$ 
 $\mathbf{Y} = \mathbf{Y}^{\mathbf{F}} + \mathbf{Y}^{\mathbf{B}}$ |
| HGRN-2 [36] | $\mathbf{S}_i = \text{Diag}(\lambda_i)\mathbf{S}_{i-1} + (1-\lambda_i) \mathbf{v}_i^\top$ 
 $\mathbf{y}_i = \mathbf{q}_i^\top \mathbf{S}_i$ | $\mathbf{S}_i^{F/B} = \text{Diag}(\lambda_i)\mathbf{S}_{i-1}^{F/B} + (1-\lambda_i)\mathbf{v}_i^\top$, $\quad \mathbf{k}_i = (1-\lambda_i)$ 
 $\mathbf{y}_i^{F/B} = \mathbf{q}_i^\top(\mathbf{S}_i^{F/B} - \frac{1}{2}\mathbf{k}_i\mathbf{v}_i)$, $\quad \mathbf{y}_i = \mathbf{y}_i^F + \mathbf{y}_i^B$ | $\mathbf{Y}^{\mathbf{F}} = \text{Tril}(\mathbf{Q} \odot \mathbf{L}^{\mathbf{F}})(\mathbf{K} \odot (\mathbf{L}^{\mathbf{F}})^{-1})\mathbf{V}$ 
 $\mathbf{Y}^{\mathbf{B}} = \text{Triu}(\mathbf{Q} \odot \mathbf{L}^{\mathbf{B}})(\mathbf{K} \odot (\mathbf{L}^{\mathbf{B}})^{-1})\mathbf{V}$ 
 $\mathbf{Y} = \mathbf{Y}^{\mathbf{F}} + \mathbf{Y}^{\mathbf{B}}$ |
| xLSTM [3] | $\mathbf{S}_i = f_i \mathbf{S}_{i-1} + i_i \mathbf{k}_i \mathbf{v}_i^\top$ 
 $\mathbf{z}_i = f_i \mathbf{z}_{i-1} + i_i \mathbf{k}_i$ 
 $\mathbf{y}_i = \mathbf{q}_i^\top \mathbf{S}_i / \max(|\mathbf{q}_i^\top \mathbf{z}_i|, 1)$ | $\mathbf{S}_i^{F/B} = f_i \mathbf{S}_{i-1}^{F/B} + i_i \mathbf{k}_i \mathbf{v}_i^\top$, 
 $\mathbf{y}_i = \mathbf{q}_i^\top \mathbf{S}_i$ 
 $\mathbf{z}_i^{F/B} = f_i \mathbf{z}_{i-1}^{F/B} + i_i\mathbf{k}_i$, $\quad c_i^{F/B} = \mathbf{q}_i^\top(\mathbf{z}_i^{F/B} - \frac{1}{2}\mathbf{k}_i)$ 
 $\mathbf{y}_i^{F/B} = \mathbf{q}_i^\top(\mathbf{S}_i^{F/B} - \frac{1}{2}\mathbf{k}_i\mathbf{v}_i)$, $\quad \mathbf{y}_i = \frac{\mathbf{y}_i^F + \mathbf{y}_i^B}{\max(c_i^F + c_i^B, 1)}$ | $\mathbf{Y} = \text{SCALE}_{max}(\mathbf{Q}\mathbf{U}^\top) \odot \mathbf{M})\mathbf{V}$, 
 $\mathbf{M}_{ij} = \begin{cases} \Pi_i^{j+1} f_k & \text{if } i > j, \\ \Pi_{i+1}^j f_k & \text{if } i < j, \\ 1 & \text{if } i = j, \end{cases}$ |
| DeltaNet [52] | $\mathbf{S}_i = (1-\beta_i \mathbf{k}_i\mathbf{k}_i^\top)\mathbf{S}_{i-1} + \beta_i \mathbf{k}_i \mathbf{v}_i^\top$ 
 $\mathbf{y}_i = \mathbf{q}_i^\top \mathbf{S}_i$ | $\mathbf{S}_i^{F/B} = (1-\beta_i \mathbf{k}_i\mathbf{k}_i^\top)\mathbf{S}_{i-1}^{F/B} + \beta_i \mathbf{k}_i\mathbf{v}_i^\top$ 
 $\mathbf{y}_i^{F/B} = \mathbf{q}_i^\top(\mathbf{S}_i^{F/B} - \frac{1}{2}\mathbf{k}_i\mathbf{v}_i)$, $\quad \mathbf{y}_i = \mathbf{y}_i^F + \mathbf{y}_i^B$ | $\mathbf{Y} = (\mathbf{Q}\mathbf{K}^\top)\mathbf{T}\mathbf{V}$, 
 $\mathbf{T} = \frac{1}{2}(\mathbf{T}^F + \mathbf{T}^B)$, 
 $\mathbf{T}^F = (\mathbf{I} + \text{tril}(\text{diag}(\beta_i)\mathbf{K}_i\mathbf{K}_i^\top, -1))^{-1}\text{diag}(\beta_i)$, 
 $\mathbf{T}^B = (\mathbf{I} + \text{triu}(\text{diag}(\beta_i)\mathbf{K}_i\mathbf{K}_i^\top, -1))^{-1}\text{diag}(\beta_i)$ |

Figure 1: Memory-speed tradeoffs for LION in three representations on Imagenet classification task: Full attention, RNN, and chunkwise from for *Left)* LION-LIT, *Middle)* LION-D, and *Right)* LION-S.

**Theorem 3.2.** *(LION-Chunk) Considering the chunks for queries, keys and values as* $\mathbf{Q}_{[i]}, \mathbf{K}_{[i]}, \mathbf{V}_{[i]} \in \mathbb{R}^{C \times d}$ *with chunk size being $C$ and total number of $N = \frac{L}{C}$ chunks, we can chunk the full Linear Attention as:*

$$\mathbf{A}_{[ij]} = \mathbf{Q}_{[i]}\mathbf{K}_{[j]}^\top \odot \mathbf{M}_{[ij]}, \quad \mathbf{C}_{[ij]} = \mathbf{C}_{[ij-1]} + Sum(\mathbf{A}_{[ij]}), \quad \mathbf{S}_{[ij]} = \mathbf{S}_{[ij-1]} + \mathbf{A}_{[ij]}\mathbf{V}_{[j]}, \quad \mathbf{Y}_{[i]} = \frac{\mathbf{S}_{[iN]}}{\mathbf{C}_{[iN]}} \quad (21)$$

*where Sum operations applies summation over the row of the input matrix.*

The chunk hidden states $\mathbf{C}_{[ij]}$ and $\mathbf{S}_{[ij]}$ iterate over $j$, with the final output for chunk $i$ computed using their last values at $j = N$. The chunk mask $\mathbf{M}_{[ij]}$ corresponds to a submatrix of the full attention mask from Eq. (13), defined as:

$$\mathbf{M}_{[ij]} = \mathbf{M}_{iC+1:i(C+1),jC+1:j(C+1)} \in \mathbb{R}^{C \times C}.$$

Below, we construct both fixed and selective chunked masks $\mathbf{M}_{[ij]}$. For the fixed mask, we have:

$$\mathbf{M}_{[ij]} = \begin{cases} \lambda^{|i-j|}(\frac{1}{\mathbf{L}} \mathbf{L}^\top) & \text{if } i > j, \\ \lambda^{|i-j|}(\mathbf{L} \frac{1}{\mathbf{L}^\top}) & \text{if } i < j, \quad \mathbf{L}_i = \lambda^i, \quad \mathbf{\Gamma}_{ij} = \lambda^{|i-j|} \\ \mathbf{\Gamma} & \text{if } i = j, \end{cases} \quad (22)$$

Table 2: *Image classification results.* **Left)** We report Top-1 accuracy and relative training time ($\downarrow$) on ImageNet. $\natural$ denotes models with mutliple scans. Best and second-best (excluding ViT) are in **bold** and underline, respectively. **Right)** Accuracy vs. Training Speed scatter plot in small scale.

| Model | #Params | | Top-1 Acc. (%) | | Train Time ($\downarrow$) | |
|---|---|---|---|---|---|---|
| | Small | Base | Small | Base | Small | Base |
| ViT | 22M | 86M | 72.2 | 77.9 | $\times 1$ | $\times 1$ |
| DeiT | 22M | 86M | 79.8 | 81.8 | $\times 1$ | $\times 1$ |
| Hydra | 22M | 91M | 78.6 | 81.0 | $\times 2.50$ | $\times 2.51$ |
| Vim | 26M | 98M | **80.3** | **81.9** | $\times 14.95$ | $\times 10.86$ |
| LION-LIT | 22M | 86M | 72.4 | 74.7 | $\times$**0.74** | $\times$**0.73** |
| LION-D | 22M | 86M | 73.5 | 77.8 | $\times 1.49$ | $\times 1.39$ |
| LION-D$^\natural$ | 22M | 86M | 79.9 | 80.2 | $\times 1.66$ | $\times 1.48$ |
| LION-S | 22M | 86M | 74.0 | 76.3 | $\times 2.03$ | $\times 1.46$ |
| LION-S$^\natural$ | 22M | 86M | 79.6 | 79.9 | $\times 2.72$ | $\times 1.68$ |

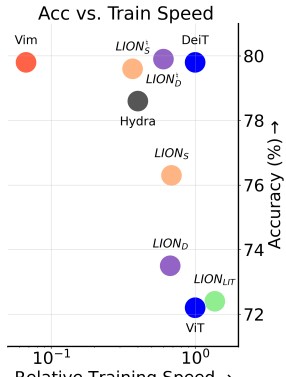

with $\mathbf{L} \in R^C$ and $\mathbf{\Gamma} \in \mathbb{R}^{C \times C}$ being the vector and matrix used for creating the mask $\mathbf{M}_{[ij]}$ and they are only depending on the decay parameter $\lambda$ and the chunk size $C$. For the selective mask we have:

$$\mathbf{M}_{[ij]} = \begin{cases} \mathbf{L}^F_{[i]} \frac{1}{\mathbf{L}^F_{[j]}}^\top & \text{if } i > j, \\ \mathbf{L}^B_{[j]} \frac{1}{\mathbf{L}^B_{[i]}}^\top & \text{if } i < j, \\ \text{Tril}\left(\mathbf{L}^F_{[i]} \frac{1}{\mathbf{L}^F_{[i]}}^\top\right) + \text{Triu}\left(\mathbf{L}^B_{[i]} \frac{1}{\mathbf{L}^B_{[i]}}^\top\right) - \mathbf{I} & \text{if } i = j, \end{cases} \quad \begin{aligned} \mathbf{L}^F_{[i]} &= \text{cumprod}(\lambda)_{iC+1:(i+1)C}, \\ \mathbf{L}^B_{[i]} &= \text{cumprod}(\text{Flip}(\lambda))_{iC+1:(i+1)C}. \end{aligned} \quad (23)$$

The chunkwise form in Eq. (21) can be parallelized over index $i$, reducing sequential operations. As bidirectional tasks require per-token outputs, memory remains subquadratic at $\mathcal{O}(LC)$. We use this as the default chunkwise mode for LION. Further visualizations and detailed proofs of LION-chunk are presented at Appendix B.9. LION full framework is shown at Figure 2 and tradeoff between different representation of LION is shown at Figure 1.

## 4 Experiments

To validate the speed and efficiency of LION we focused on well-known bidirectional sequence modeling tasks: Image Classification on ImageNet-1K [38] and Masked Language Modeling (MLM) on the C4 dataset [8]. We also conduct experiments on the LRA dataset to ensure the stability of LION. For Causal Language Modeling and additional experiments, we refer to Appendix A and C.

### 4.1 Detail of LION and baseline for MLM and image classification tasks

We evaluate the LION framework through its bidirectional variants, LION-S, LION-D, and LION-LIT, on image classification and MLM task. For vision tasks, we compare against softmax-based Transformers (ViT [9], DeiT [47]) and state-of-the-art bidirectional SSMs (Vision Mamba [53], Hydra [21]). For Masked Language Modeling, we benchmark against BERT [10] and Hydra. *We replace the attention in the original DeiT and BERT codebases with LION as a drop-in substitution, without modifying any other parts such as configurations, optimizer settings, or data augmentation parameters.*

Moreover, Linear Transformers do not use external positional embeddings in vision tasks, they often capture spatial structure through different scans by traversing the image along different paths, resulting in different token orderings [35]. Each scan induces a different decay mask $\mathbf{M}$. We show that LION naturally supports such variations through alternate token sequences, leading to models like LION-D$^\natural$ and LION-S$^\natural$, which capture spatial information similarly to existing vision Linear Transformers and SSMs [1]. Details of this strategy and its implementation for LION are provided in Appendix C.14.

### 4.2 Training speed and inference efficiency for image classification

**Training speed and accuracy** To support our claims, we pre-train three LION variants (LION-LIT, LION-D, and LION-S) across Tiny, Small, and Base scales for image classification. As shown in Table 2, all LION models outperform the vanilla ViT and perform close to DeiT. Notably, LION-D$^\natural$

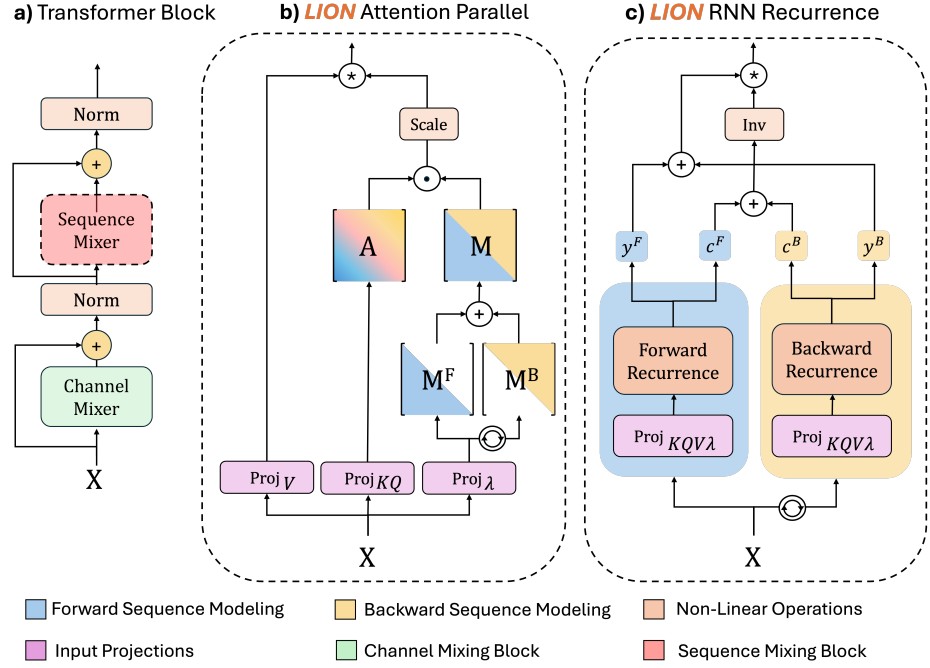

**a)** Transformer Block    **b)** *LION* Attention Parallel    **c)** *LION* RNN Recurrence

Forward Sequence Modeling    Backward Sequence Modeling    Non-Linear Operations

Input Projections    Channel Mixing Block    Sequence Mixing Block

Figure 2: (*Left*) Standard Transformer block. (*Middle*) Training mode of LION with full linear attention. (*Right*) Inference mode of LION in the equivalent bidirectional RNN.

and Vim are the only models to surpass DeiT in the Small scale, with LION-D♮ being $10\times$ **faster** to train, as shown in Table 2 (Right). Tiny scale results are in Appendix C.8, and training details in Appendix C.11. The LION-S♮ and LION-D♮ variants lie in the *optimal region of the accuracy–speed trade-off* (**top right corner** of Figure in Table 2), achieving softmax-level accuracy and outperforming state-of-the-art SSMs in training speed due to their full-attention. Notably, LION achieves its fast training *entirely through PyTorch matrix operations*, while still outperforming SSMs like Vim and Hydra despite those relying on custom kernel implementations for speedups.

### 4.3 Inference memory-speed trade off for LION in RNN, Attention and Chunk form

As we also show theoretically, LION-chunk balances the tradeoff between memory and speed. To illustrate this tradeoff, we evaluated LION using three inference strategies: (1) full linear attention (Eq. (8)), (2) bidirectional RNN (Eq. (19)), and (3) chunked attention (Eq. (21)), across all three decay variants: LION-S, LION-LIT, and LION-D. As shown in Figure 2, the RNN form is the most memory-efficient, while full attention consumes the most memory. Chunked attention strikes a balance, with memory usage between the two depending on type of mask. Both full attention and chunkwise provide significantly faster inference than the RNN form, specifically for LION-LIT and LION-D. Chunkwise form enables faster inference than full attention and is more memory-efficient, making it a practical choice for balancing speed and memory for LION-D and LION-LIT. The effect of chunk size is further explored in Figure 4. Note that LION-D (LION-D♮) achieves the highest accuracy in image classification with its three representations shown in Figure 1 (*Middle*).

We study the effect of chunk size on LION inference with $1248$-resolution images, observing a trade-off where larger chunks reduce time but increase memory, with OOM beyond size $1024$, and we typically use $8$–$16$ chunks as a balance as shown in Figure 4.

### 4.4 Masked Language Modeling (MLM)

We evaluate BERT, Hydra, and LION variants using the standard pre-training recipe widely adopted in prior work [7, 10]. We pre-train models from the LARGE family and fine-tune them on the GLUE benchmark [49] to assess LION 's performance on large-scale masked language modeling. As shown in Table 3, LION variants with 334M parameters closely match the performance of softmax-based BERT and state-of-the-art SSMs like Hydra, while being $3\times$ faster to train. Thanks to their RNN formulation, LION models are also significantly more memory-efficient than BERT during inference

Table 3: *C4 MLM and GLUE results for the LARGE scale (334M).* Best and second-best results are in **bold** and underline, respectively.

| Model | MLM Acc. | GLUE | Train. time |
|-------|----------|------|-------------|
| BERT | 69.88 | **82.95** | ×1 |
| Hydra | **71.18** | 81.77 | ×3.13 |
| LION-LIT | 67.11 | 80.76 | **×0.95** |
| LION-D | 68.64 | 81.34 | ×1.10 |
| LION-S | 69.16 | 81.58 | ×1.32 |

Table 4: *LRA Ablation.* PathX and average results for LRA benchmark (best in **bold**).

| Model | PathX | Avg. |
|-------|-------|------|
| LION-LIT | ✗ | 50.41 |
| LION-D (w/ *HIPPO*) | 97.28 | 85.63 |
| LION-S (w/ *HIPPO*) | 97.99 | **86.07** |

**Inference memory efficiency** While LION matches the accuracy and training speed of softmax-based Transformers, *it scales linearly during inference via its RNN formulation, unlike softmax attention which scales quadratically with sequence length* (Figure 3). Among all baselines, LION achieves the most efficient inference scaling. This highlights the importance of including both attention and RNN representation for Linear Transformers in bi-directional modeling to achieve the best of both worlds: fast training and efficient inference. Moreover, inference times for all models ar provided at Appendix C.10.

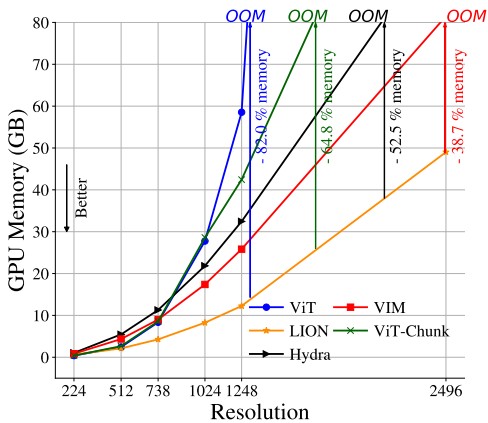

Figure 3: *Inference efficiency of* LION. GPU memory usage of LION in RNN form and baselines vs the resolution of images for in base scale.

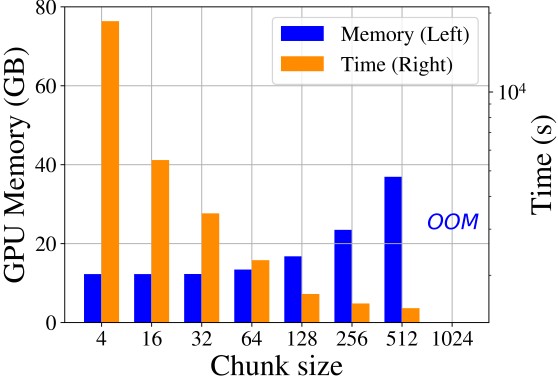

Figure 4: *Effect of chunksize in GPU memory and Speed of* LION-*chunk for* LION-D.

(Figure 9), consistent with results from image classification. LION achieves *significantly faster training than Hydra while maintaining comparable performance, marginally lower at larger scales and higher at smaller scales.* For detailed results on BASE scale and ablations please check Appendix C.6.

### 4.5  Long Range Arena stability ablation

To assess the stability of LION with diagonal decay (described in Theorem B.1), specially for long sequences, we evaluated it on the LRA benchmark. The diagonal decay factors of both LION-S and LION-D were initialized using well-known and standard HIPPO-based diagonal initialization [14, 16]. For these tasks, the selectivity of LION-S is defined as $\Lambda_{\mathbf{i}} = \exp(\sigma(\mathbf{A}_i) + \mathbf{B}_i)$ and for LION-D as $\Lambda_i = \exp(\mathbf{B}_i)$, where $B_i$ follows the HIPPO. With this setup, both LION-S and LION-D successfully solved the LRA tasks (Table 3). In contrast, LION-LIT in the absence of decay was unable to solve LRA, consistent with prior findings [45, 32]. For random initializations ablation and configurations please check Table 5 and Appendix C.3.

## 5  Conclusion

We introduce LION, as the first unified framework to adapt Linear Transformers into their bidirectional counterparts. LION supports three main representations: full linear attention, bidirectional RNN, and chunkwise parallel form. These forms are theoretically equivalent and allow the model to benefit from high-throughput training using full attention, and efficient inference using either RNN or chunkwise computation. We also provide theoretical mappings of several existing Linear Transformers into their bidirectional versions using LION (Table 1). We evaluate three representative models—LION-LIT, LION-D, and LION-S —based on different decay types, on masked language modeling and image classification tasks.

## Acknowledgments

We would like to thank the reviewers for their valuable feedback, which significantly improved both our paper and its presentation. We thank Dr. Igor Krawczuk for insightful discussions on LION models for the masked language modeling experiment. This work was partially supported by Hasler Foundation Program: Hasler Responsible AI (project number 21043), partially sponsored by the Army Research Office and was accomplished under Grant Number W911NF-24-1-0048, and partially funded by the Swiss National Science Foundation (SNSF) under grant number 200021-205011. Additionally, this work was supported under project ID #37 as part of the Swiss AI Initiative, through a grant from the ETH Domain and computational resources provided by the Swiss National Supercomputing Centre (CSCS) under the Alps infrastructure.

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

# Appendix

## Table of Contents

# A  Causal Language Modeling

We also evaluated the performance of LION-S with decay factor of $\lambda_i = \sigma(\mathbf{W}_a\mathbf{x}_i + b)$ in causal sequence modeling by replacing the softmax attention in the GPT-2 architecture with LION-S attention, following the setup of Radford et al. [37]. Note that LION-S does not use absolute positional encoding. We train our models in the OpenWebText corpus [12]. We evaluate the architectures in the 124 M parameter setup. Our implementation is based on nanoGPT[4]. We use the default GPT-2 hyperparameters and train our models for 8 days in 4 NVIDIA A100 SXM4 40 GB GPUs.

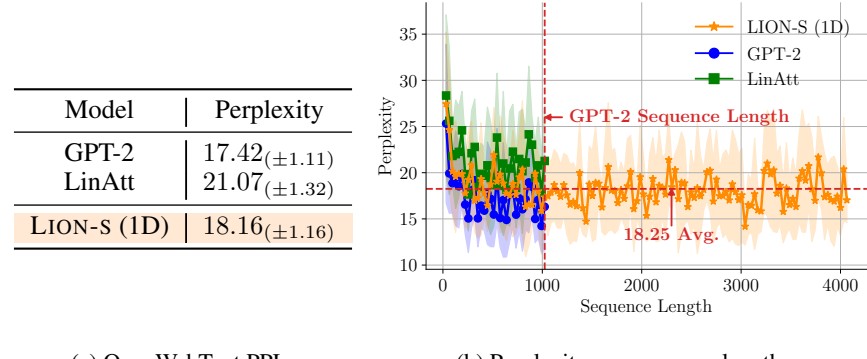

| Model | Perplexity |
|---|---|
| GPT-2 | $17.42_{(\pm1.11)}$ |
| LinAtt | $21.07_{(\pm1.32)}$ |
| LION-S (1D) | $18.16_{(\pm1.16)}$ |

(a) OpenWebText PPL

(b) Perplexity vs. sequence length

Figure 5: *Causal Language Modelling results in the GPT-2 128M size.* *(a)* Perplexity in the OpenWebText dataset. *(b)* Perplexity vs. sequence length in OpenWebText. LION-S improve over the LinAtt baseline [25] while obtaining similar performance to the GPT baseline and being able to extrapolate to larger context lengths than the one used during training.

In Figure 5 we can observe LION-S (1D) significantly improve over the LinAtt baseline, while obtain perplexity close to GPT-2. The lack of absolute positional encodings allows LION-S (1D) to scale to larger sequence lengths than the one used during training.

In Figure 6 we evaluate the latency and memory of LION-S (1D) in three modes: Attention, Attention + KV cache and RNN. While the three modes have the same output, the RNN formulation allows to save computation from previous token generations to require constant memory and latency for generating the next token. Our results align with the findings of Sun et al. [44], showing that efficient

---

[4]https://github.com/karpathy/nanoGPT

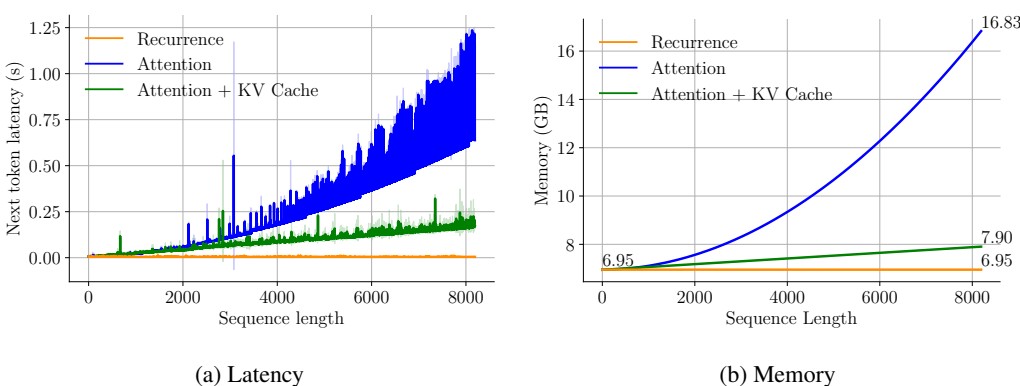

(a) Latency

(b) Memory

Figure 6: *Efficiency of the LION-S (1D) framework in the next-token generation task.* In *(a)* and *(b)* we measure respectively the latency and memory to generate the next token in the sentence. We compare three generation modes: Attention, Attention with KV cache and the Recurrence formulation. While all three produce the same output, the Recurrence formulation is the most efficient, requiring constant memory and latency to generate the next token.

models in training and inference, with a strong performance (up to a small degradation) can be obtained.

## A.1 Memory allocation in LION during Forward and Backward recurrences

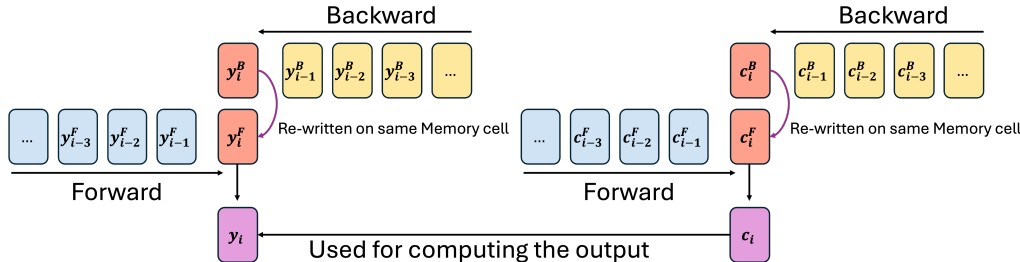

Figure 7: *Memory allocation in* LION *during Forward and Backward recurrences.* The efficient way of re-using the memory during inference is explained.

# B Proofs

## B.1 Duality between Linear Recurrence and Attention

Considering the following recurrence:

$$\mathbf{S}_i = \lambda_i \mathbf{S}_{i-1} + \mathbf{k}_i \mathbf{v}_i^\top, \tag{24}$$

$$\mathbf{z}_i = \lambda_i \mathbf{z}_{i-1} + \mathbf{k}_i, \tag{25}$$

$$\text{SCALED} : \mathbf{y}_i = \frac{\mathbf{q}_i^\top \mathbf{S}_i}{\mathbf{q}_i^\top \mathbf{z_i}} \tag{26}$$

We can calculate each output $\mathbf{y}_i$ recursively as below:

$$\mathbf{S}_1 = \mathbf{k}_1 \mathbf{v}_1^\top, \ \ \mathbf{z}_1 = \mathbf{k}_1, \ \ \mathbf{y}_1 = \mathbf{v}_1 \tag{27}$$

$$\mathbf{S}_2 = \mathbf{k}_2 \mathbf{v}_2^\top + \lambda_1 \mathbf{k}_1 \mathbf{v}_1^\top, \ \ \mathbf{z}_2 = \mathbf{k}_2 + \lambda_1 \mathbf{k}_1, \ \ \mathbf{y}_2 = \frac{\mathbf{q}_2^\top (\mathbf{k}_2 \mathbf{v}_2^\top + \lambda_1 \mathbf{k}_1 \mathbf{v}_1^\top)}{\mathbf{q}_2^\top (\mathbf{k}_2 + \lambda_1 \mathbf{k}_1)} \tag{28}$$

$$\mathbf{S}_3 = \mathbf{k}_3 \mathbf{v}_3^\top + \lambda_1 \mathbf{k}_2 \mathbf{v}_2^\top + \lambda_2 \lambda_1 \mathbf{k}_1 \mathbf{v}_1^\top, \ \ \mathbf{z}_3 = \mathbf{k}_3 + \lambda_1 \mathbf{k}_2 + \lambda_2 \lambda_1 \mathbf{k}_1, \ \ \mathbf{y}_3 = \frac{\mathbf{q}_3^\top (\mathbf{k}_3 \mathbf{v}_3^\top + \lambda_1 \mathbf{k}_2 \mathbf{v}_2^\top + \lambda_2 \lambda_1 \mathbf{k}_1 \mathbf{v}_1^\top)}{\mathbf{q}_3^\top (\mathbf{k}_3 + \lambda_1 \mathbf{k}_2 + \lambda_2 \lambda_1 \mathbf{k}_1)} \tag{29}$$

$$\Rightarrow \mathbf{y}_i = \frac{\mathbf{q}_i^\top (\sum_{j=1}^i \mathbf{M}_{ij}^C \mathbf{k}_j \mathbf{v}_j^\top)}{\mathbf{q}_i^\top (\sum_{j=1}^i \mathbf{M}_{ij}^C \mathbf{k}_j)}, \qquad \mathbf{M}_{ij}^C = \begin{cases} \Pi_{k=i}^{j+1} \lambda_k & i \geq j \\ 0 & i < j \end{cases} \tag{30}$$

This can be shown in a vectorized form as:

$$\mathbf{Y} = \text{SCALE}(\mathbf{Q}\mathbf{K}^\top \odot \mathbf{M}^C)\mathbf{V} \tag{31}$$

Where SCALE is the scaling function which scaled the attention matrix with respect to each row or can also be written as:

$$\text{SCALE}(\mathbf{A})_{ij} = \frac{\mathbf{A}_{ij}}{\sum_{j=1}^L \mathbf{A}_{ij}} \tag{32}$$

Similarly if the SCALE is applied before masking we have:

$$\mathbf{Y} = \big(\text{SCALE}(\mathbf{Q}\mathbf{K}^\top \odot \mathbf{M}_{\text{CAUSAL}}) \odot \mathbf{M}\big)\mathbf{V} \tag{33}$$

With $\mathbf{M}_{\text{CAUSAL}}$ being the causal mask used in autoregressive models [27]. This vectorized form is equivalent to:

$$\mathbf{y}_i = \frac{\mathbf{q}_i^\top (\sum_{j=1}^i \mathbf{M}_{ij} \mathbf{k}_j \mathbf{v}_j^\top)}{\mathbf{q}_i^\top (\sum_{j=1}^i \mathbf{k}_j)}, \qquad \mathbf{M}_{ij} = \begin{cases} \Pi_{k=i}^{j+1} \lambda_k & i \geq j \\ 0 & i < j \end{cases} \tag{34}$$

And the recurrence for this vectorized form can be written as:

$$\mathbf{S}_i = \lambda_i \mathbf{S}_{i-1} + \mathbf{k}_i \mathbf{v}_i^\top, \tag{35}$$

$$\mathbf{z}_i = \mathbf{z}_{i-1} + \mathbf{k}_i, \tag{36}$$

$$\text{SCALED} : \mathbf{y}_i = \frac{\mathbf{q}_i^\top \mathbf{S}_i}{\mathbf{q}_i^\top \mathbf{z_i}} \tag{37}$$

## B.2 Forward and Backward Recurrences Theoretical Details

Considering the following recurrence:

$$\mathbf{S}_i = \lambda_i \mathbf{S}_{i-1} + \mathbf{k}_i \mathbf{v}_i^\top, \tag{38}$$

$$\mathbf{z}_L = \sum_{i=1}^L \mathbf{k}_i \tag{39}$$

$$\mathbf{y}_i = \frac{\mathbf{q}_i^\top \mathbf{S}_i}{\mathbf{q}_i^\top \mathbf{z_L}} \tag{40}$$

This recurrence is the same as recurrence equation 35 but with $\mathbf{z}_L$ being fixed to the summation of all keys in the sequence, therefor the output $\mathbf{y}_i$ can simply be written as:

$$\mathbf{y}_i = \frac{\mathbf{q}_i^\top (\sum_{j=1}^i \mathbf{M}_{ij} \mathbf{k}_j \mathbf{v}_j^\top)}{\mathbf{q}_i^\top \mathbf{z}_L}, \qquad \mathbf{M}_{ij} = \begin{cases} \Pi_{k=i}^{j+1} \lambda_k & i \geq j \\ 0 & i < j \end{cases} \tag{41}$$

By replacing the $\mathbf{z}_i = \sum_{j=1}^i \mathbf{k}_j$ in the denominator of equation equation 37 with $\mathbf{z}_L$. Therefore in vectorized form, it will become:

$$\mathbf{Y} = (\mathbf{A}^C \odot \mathbf{M}) \mathbf{V} \tag{42}$$

With $\mathbf{A}^C$ being:

$$\mathbf{A}^C = \begin{pmatrix} \frac{\mathbf{q}_1^\top \mathbf{k}_1}{\mathbf{q}_1^\top \mathbf{z}_L} & & & \\ \frac{\mathbf{q}_2^\top \mathbf{k}_1}{\mathbf{q}_2^\top \mathbf{z}_L} & \frac{\mathbf{q}_2^\top \mathbf{k}_2}{\mathbf{q}_2^\top \mathbf{z}_L} & & \\ \frac{\mathbf{q}_3^\top \mathbf{k}_1}{\mathbf{q}_3^\top \mathbf{z}_L} & \frac{\mathbf{q}_3^\top \mathbf{k}_2}{\mathbf{q}_3^\top \mathbf{z}_L} & \frac{\mathbf{q}_3^\top \mathbf{k}_3}{\mathbf{q}_3^\top \mathbf{z}_L} & \\ \vdots & \vdots & \vdots & \ddots \\ \frac{\mathbf{q}_L^\top \mathbf{k}_1}{\mathbf{q}_L^\top \mathbf{z}_L} & \frac{\mathbf{q}_L^\top \mathbf{k}_2}{\mathbf{q}_L^\top \mathbf{z}_L} & \cdots & \frac{\mathbf{q}_L^\top \mathbf{k}_L}{\mathbf{q}_L^\top \mathbf{z}_L} \end{pmatrix}$$

Importantly this equation can be written as:

$$\mathbf{Y} = (\text{SCALE}(\mathbf{Q}\mathbf{K}^\top) \odot \mathbf{M}) \mathbf{V} \tag{43}$$

which despite equation equation 33 scaling is applied over the whole sequence not for the causal part of the sequence. The matrix $\mathbf{A}^C$ is helpful for driving the recurrent version of LION for Forward and Backward recurrences and the mask here $\mathbf{M}$ is equal to LION's forward mask $\mathbf{M}^F$ in equation equation 16. As shown in equation 16 the forward recurrence for the causal part of the attention can be presented as $\mathbf{Y}^B = \mathbf{A}^F \odot \mathbf{M}^F$ the matrix $\mathbf{A}^F$ can be created simply by using matrix $\mathbf{A}^C$ as bellow:

$$
\underbrace{\begin{pmatrix}
\frac{1}{2}\frac{\mathbf{q}_1^\top \mathbf{k}_1}{\mathbf{q}_1^\top \mathbf{z}_L} & & & & \\
\frac{\mathbf{q}_2^\top \mathbf{k}_1}{\mathbf{q}_2^\top \mathbf{z}_L} & \frac{1}{2}\frac{\mathbf{q}_2^\top \mathbf{k}_2}{\mathbf{q}_2^\top \mathbf{z}_L} & & & \\
\frac{\mathbf{q}_3^\top \mathbf{k}_1}{\mathbf{q}_3^\top \mathbf{z}_L} & \frac{\mathbf{q}_3^\top \mathbf{k}_2}{\mathbf{q}_3^\top \mathbf{z}_L} & \frac{1}{2}\frac{\mathbf{q}_3^\top \mathbf{k}_3}{\mathbf{q}_3^\top \mathbf{z}_L} & & \\
\vdots & \vdots & \vdots & \ddots & \\
\frac{\mathbf{q}_L^\top \mathbf{k}_1}{\mathbf{q}_L^\top \mathbf{z}_L} & \frac{\mathbf{q}_L^\top \mathbf{k}_2}{\mathbf{q}_L^\top \mathbf{z}_L} & \cdots & & \frac{1}{2}\frac{\mathbf{q}_L^\top \mathbf{k}_L}{\mathbf{q}_L^\top \mathbf{z}_L}
\end{pmatrix}}_{\mathbf{A}^F}
=
\underbrace{\begin{pmatrix}
\frac{\mathbf{q}_1^\top \mathbf{k}_1}{\mathbf{q}_1^\top \mathbf{z}_L} & & & & \\
\frac{\mathbf{q}_2^\top \mathbf{k}_1}{\mathbf{q}_2^\top \mathbf{z}_L} & \frac{\mathbf{q}_2^\top \mathbf{k}_2}{\mathbf{q}_2^\top \mathbf{z}_L} & & & \\
\frac{\mathbf{q}_3^\top \mathbf{k}_1}{\mathbf{q}_3^\top \mathbf{z}_L} & \frac{\mathbf{q}_3^\top \mathbf{k}_2}{\mathbf{q}_3^\top \mathbf{z}_L} & \frac{\mathbf{q}_3^\top \mathbf{k}_3}{\mathbf{q}_3^\top \mathbf{z}_L} & & \\
\vdots & \vdots & \vdots & \ddots & \\
\frac{\mathbf{q}_L^\top \mathbf{k}_1}{\mathbf{q}_L^\top \mathbf{z}_L} & \frac{\mathbf{q}_L^\top \mathbf{k}_2}{\mathbf{q}_L^\top \mathbf{z}_L} & \cdots & & \frac{\mathbf{q}_L^\top \mathbf{k}_L}{\mathbf{q}_L^\top \mathbf{z}_L}
\end{pmatrix}}_{\mathbf{A}^C}
-
\underbrace{\begin{pmatrix}
\frac{1}{2}\frac{\mathbf{q}_1^\top \mathbf{k}_1}{\mathbf{q}_1^\top \mathbf{z}_L} & & & & \\
& \frac{1}{2}\frac{\mathbf{q}_2^\top \mathbf{k}_2}{\mathbf{q}_2^\top \mathbf{z}_L} & & & \\
& & \frac{1}{2}\frac{\mathbf{q}_3^\top \mathbf{k}_3}{\mathbf{q}_3^\top \mathbf{z}_L} & & \\
& & & \ddots & \\
& & & & \frac{1}{2}\frac{\mathbf{q}_L^\top \mathbf{k}_L}{\mathbf{q}_L^\top \mathbf{z}_L}
\end{pmatrix}}_{\mathbf{D}^F}
$$

Or equivalently:

$$
\mathbf{Y}^F = \mathbf{A}^F \odot \mathbf{M}^F = (\mathbf{A}^C - \mathbf{D}^F) \odot \mathbf{M}^F \tag{44}
$$

Since the diagonal values of the mask $\mathbf{M}^F$ are all ones and the matrix $\mathbf{D}^F$ is diagonal, we have:

$$
\mathbf{Y}^F = (\mathbf{A}^C - \mathbf{D}^F) \odot \mathbf{M}^F = \mathbf{A}^C \odot \mathbf{M}^F - \mathbf{D}^F \tag{45}
$$

As $\mathbf{A}^C \odot \mathbf{M}^F$ corresponds to linear recurrence shown at equation 41. The vectorized form equation 45 can be presented as linear recurrence:

$$
\mathbf{y}_i = \frac{\mathbf{q}_i^\top \left( \sum_{j=1}^i \mathbf{M}_{ij} \mathbf{k}_j \mathbf{v}_j^\top \right)}{\mathbf{q}_i^\top \mathbf{z}_L} - \frac{1}{2}\frac{\mathbf{q}_i^\top \mathbf{k}_i}{\mathbf{q}_i^\top \mathbf{z}_L}, \qquad \mathbf{M}_{ij} = \begin{cases} \Pi_{k=i}^{j+1} \lambda_k & i \geq j \\ 0 & i < j \end{cases} \tag{46}
$$

This is equivalent to the linear recurrence presented in equation equation 40. The same theoretical approach applies to the backward recurrence, leading to the following linear recurrence for both recurrences:

$$
\mathbf{S}_i^F = \lambda_i \mathbf{S}_{i-1}^F + \mathbf{k}_i \mathbf{v}_i^\top, \tag{47a}
$$
$$
\mathbf{S}_i^B = \lambda_{L-i} \mathbf{S}_{i-1}^B + \mathbf{k}_{L-i+1} \mathbf{v}_{L-i+1}^\top, \tag{48a}
$$

$$
\mathbf{y}_i^F = \frac{\mathbf{q}_i^\top \mathbf{S}_i^F}{\mathbf{q}_i^\top \mathbf{z}_L} - \frac{1}{2}\frac{\mathbf{q}_i^\top \mathbf{k}_i}{\mathbf{q}_i^\top \mathbf{z}_L} \tag{47b}
$$
$$
\mathbf{y}_{L-i+1}^B = \frac{\mathbf{q}_{L-i+1}^\top \mathbf{S}_i^B}{\mathbf{q}_{L-i+1}^\top \mathbf{z}_L} - \frac{1}{2}\frac{\mathbf{q}_{L-i+1}^\top \mathbf{k}_{L-i+1}}{\mathbf{q}_{L-i+1}^\top \mathbf{z}_L} \tag{48b}
$$

However, the above equation requires access to the summation of scaling values $\mathbf{z}_L$. A naive approach would involve adding an additional scaling recurrence alongside the forward and backward recurrences to compute the summation of all keys in the sequence. This approach, however, is inefficient, as it complicates the process. While the forward and backward recurrences can traverse the sequence in parallel to obtain the forward and backward recurrences outputs $\mathbf{Y}^F$ and $\mathbf{Y}^B$, the scaling recurrence must be computed prior to these recurrences because both the forward and backward recurrences computations rely on the final scaling value $\mathbf{z}_L$ to generate their outputs.

### B.3 Efficient and Simple Method for Scaling Attention During Inference

As shown in previous section scaled attention matrix can be formulated as two recurrences equation 47 and equation 48 with an additional recurrence to sum all the keys ($\mathbf{z}_L$). This section we will proof how to avoid an extra scaling recurrence by simple modifications to equation equation 47 and equation 48.

Considering having a scaling recurrence as part of forward and backward recurrence we will have:

$$
\mathbf{S}_i^F = \lambda_i \mathbf{S}_{i-1}^F + \mathbf{k}_i \mathbf{v}_i^\top, \tag{49a}
$$
$$
\mathbf{S}_i^B = \lambda_{L-i} \mathbf{S}_{i-1}^B + \mathbf{k}_{L-i+1} \mathbf{v}_{L-i+1}^\top, \tag{50a}
$$

$$
\mathbf{z}_i^F = \mathbf{z}_{i-1}^F + \mathbf{k}_i \tag{49b}
$$
$$
\mathbf{z}_i^B = \mathbf{z}_{i-1}^B + \mathbf{k}_{L-i+1} \tag{50b}
$$

$$
c_i^F = \mathbf{q}_i^\top \mathbf{z}_i^F - \frac{1}{2}\mathbf{q}_i^\top \mathbf{k}_i \tag{49c}
$$
$$
c_i^B = \mathbf{q}_{L-i+1}^\top \mathbf{z}_i^B - \frac{1}{2}\mathbf{q}_{L-i+1}^\top \mathbf{k}_{L-i+1} \tag{50c}
$$

$$
\mathbf{y}_i^F = \mathbf{q}_i^\top \mathbf{S}_i^F - \frac{1}{2}\mathbf{q}_i^\top \mathbf{k}_i \mathbf{v}_i \tag{49d}
$$
$$
\mathbf{y}_{L-i+1}^B = \mathbf{q}_{L-i+1}^\top \mathbf{S}_i^B - \frac{1}{2}\mathbf{q}_{L-i+1}^\top \mathbf{k}_{L-i+1} \mathbf{v}_{L-i+1}^\top \tag{50d}
$$

The equations above are similar to the previous ones, with the addition of scalar states $c^F$ and $c^B$ for the backward and forward recurrences, respectively. During each recurrence, the outputs $\mathbf{y}_i^F$ and $\mathbf{y}_i^B$, along with the scalars $c_i^F$ and $c_i^B$, are saved for each token to construct the final output of each layer. *It is also important to note that there is no need to save $\mathbf{z}^F$ and $\mathbf{z}^B$ for each token; these states can simply be overwritten in memory.* The final output of each layer is equal to:

$$\mathbf{y}_i = \frac{\mathbf{y}_i^F + \mathbf{y}_i^B}{c_i^F + c_i^B} \tag{51}$$

Where $\mathbf{y}_i^F$ and $\mathbf{y}_i^B$ can be written as:

$$\mathbf{y}_i^F = \mathbf{q}_i^\top \left( \sum_{j=1}^{i} \mathbf{M}_{ij}^F \mathbf{k}_j \mathbf{v}_j^\top \right) - \frac{1}{2} \mathbf{q}_i^\top \mathbf{k}_i \mathbf{v}_i \ , \ \ \mathbf{y}_i^B = \mathbf{q}_i^\top \left( \sum_{j=i}^{L} \mathbf{M}_{ij}^B \mathbf{k}_j \mathbf{v}_j^\top \right) - \frac{1}{2} \mathbf{q}_i^\top \mathbf{k}_i \mathbf{v}_i \tag{52}$$

So the addition $\mathbf{y}_i^F + \mathbf{y}_i^B$ is equal to:

$$\mathbf{y}_i^F + \mathbf{y}_i^B = \mathbf{q}_i^\top \left( \sum_{j=1}^{i} \mathbf{M}_{ij}^F \mathbf{k}_j \mathbf{v}_j^\top \right) + \mathbf{q}_i^\top \left( \sum_{j=i}^{L} \mathbf{M}_{ij}^B \mathbf{k}_j \mathbf{v}_j^\top \right) - \mathbf{q}_i^\top \mathbf{k}_i \mathbf{v}_i \tag{53}$$

$$\Rightarrow \mathbf{y}_i^F + \mathbf{y}_i^B = \mathbf{q}_i^\top \left( \sum_{j=1}^{i} \mathbf{M}_{ij}^F \mathbf{k}_j \mathbf{v}_j^\top + \sum_{j=i}^{L} \mathbf{M}_{ij}^B \mathbf{k}_j \mathbf{v}_j^\top \right) - \mathbf{q}_i^\top \mathbf{k}_i \mathbf{v}_i \tag{54}$$

Where by considering the mask $\mathbf{M}$ as bellow:

$$\mathbf{M}_{ij} = \begin{cases} \Pi_{k=j}^{i+1} \lambda_k & i > j \\ \Pi_{k=i+1}^{j} \lambda_k & i < j \\ 1 & i = j \end{cases} = \begin{pmatrix} 1 & \lambda_2 & \lambda_2 \lambda_3 & \cdots & \lambda_2 \cdots \lambda_L \\ \lambda_1 & 1 & \lambda_3 & \cdots & \lambda_3 \cdots \lambda_L \\ \lambda_1 \lambda_2 & \lambda_2 & 1 & \cdots & \lambda_4 \cdots \lambda_L \\ \vdots & \vdots & \vdots & \ddots & \vdots \\ \lambda_{L-1} \cdots \lambda_1 & \lambda_{L-1} \cdots \lambda_2 & \lambda_{L-1} \cdots \lambda_3 & \cdots & 1 \end{pmatrix} \tag{55}$$

The above mask is equal to $\mathbf{M}^F + \mathbf{M}^B - \mathbf{I}$, allowing equation equation 53 to be rewritten as:

$$\mathbf{y}_i^F + \mathbf{y}_i^B = \mathbf{q}_i^\top \left( \sum_{j=1}^{i} \mathbf{M}_{ij}^F \mathbf{k}_j \mathbf{v}_j^\top + \sum_{j=i}^{L} \mathbf{M}_{ij}^B \mathbf{k}_j \mathbf{v}_j^\top \right) - \mathbf{q}_i^\top \mathbf{k}_i \mathbf{v}_i \tag{56}$$

$$= \mathbf{q}_i^\top \left( \sum_{j=1}^{L} \mathbf{M}_{ij} \mathbf{k}_j \mathbf{v}_j^\top \right) + \mathbf{q}_i^\top \mathbf{k}_i \mathbf{v}_i - \mathbf{q}_i^\top \mathbf{k}_i \mathbf{v}_i \tag{57}$$

$$= \mathbf{q}_i^\top \left( \sum_{j=1}^{L} \mathbf{M}_{ij} \mathbf{k}_j \mathbf{v}_j^\top \right) \tag{58}$$

So we can finally find the output of each layer $\mathbf{y}_i$ as:

$$\mathbf{y}_i = \frac{\mathbf{y}_i^F + \mathbf{y}_i^B}{c_i^F + c_i^B} \xrightarrow{\text{equation 58}} \mathbf{y}_i = \frac{\mathbf{q}_i^\top \left( \sum_{j=1}^{L} \mathbf{M}_{ij} \mathbf{k}_j \mathbf{v}_j^\top \right)}{c_i^F + c_i^B} \tag{59}$$

It can easily be shown that:

$$c_i^F = \mathbf{q}_i^\top \left(\sum_{j=1}^{i} \mathbf{k}_j\right) - \frac{1}{2}\mathbf{q}_i^\top \mathbf{k}_i \ , \ \ c_i^B = \mathbf{q}_i^\top \left(\sum_{j=i}^{L} \mathbf{k}_j\right) - \frac{1}{2}\mathbf{q}_i^\top \mathbf{k}_i \tag{60}$$

$$\Rightarrow c_i^F + c_i^B = \mathbf{q}_i^\top \left(\sum_{j=1}^{L} \mathbf{k}_j\right) + \mathbf{q}_i^\top \mathbf{k}_i - \frac{1}{2}\mathbf{q}_i^\top \mathbf{k}_i - \frac{1}{2}\mathbf{q}_i^\top \mathbf{k}_i \tag{61}$$

$$\Rightarrow c_i^F + c_i^B = \mathbf{q}_i^\top \left(\sum_{j=1}^{L} \mathbf{k}_j\right) + \mathbf{q}_i^\top \mathbf{k}_i - \mathbf{q}_i^\top \mathbf{k}_i = \mathbf{q}_i^\top \left(\sum_{j=1}^{L} \mathbf{k}_j\right) = \mathbf{q}_i^\top \mathbf{z}_L \tag{62}$$

So the final output of the layer is:

$$\mathbf{y}_i = \frac{\mathbf{y}_i^F + \mathbf{y}_i^B}{c_i^F + c_i^B} = \frac{\mathbf{q}_i^\top \left(\sum_{j=1}^{L} \mathbf{M}_{ij}\mathbf{k}_j\mathbf{v}_j^\top\right)}{\mathbf{q}_i^\top \left(\sum_{j=1}^{L} \mathbf{k}_j\right)} \tag{63}$$

Alternatively, in vectorized form, it can be expressed as:

$$\mathbf{Y} = \mathbf{Y}^F + \mathbf{Y}^B = \left(\text{SCALE}(\mathbf{Q}\mathbf{K}^\top) \odot \mathbf{M}\right)\mathbf{V} \tag{64}$$

with $\mathbf{M}$ being the attention mask created by $\lambda_i$s as in equation 55.

### B.4  Flipping Operation in Backward recurrence

Here we define the operation which flip the matrices $\mathbf{A}^B, \mathbf{M}^B$ for the reverse reccurence th goal is to find the $F(.)$ such that:

$$\mathbf{A}^B = \begin{pmatrix} \frac{1}{2}\mathbf{q}_1^\top \mathbf{k}_1 & \mathbf{q}_1^\top \mathbf{k}_2 & \cdots & \mathbf{q}_1^\top \mathbf{k}_L \\ & \frac{1}{2}\mathbf{q}_2^\top \mathbf{k}_2 & \cdots & \mathbf{q}_2^\top \mathbf{k}_L \\ & & \ddots & \vdots \\ & & & \frac{1}{2}\mathbf{q}_L^\top \mathbf{k}_L \end{pmatrix} \rightarrow F(\mathbf{A}^B) = \begin{pmatrix} \frac{1}{2}\frac{\mathbf{q}_L^\top \mathbf{k}_L}{\mathbf{q}_L^\top \mathbf{z}_L} & & \\ \frac{\mathbf{q}_{L-1}^\top \mathbf{k}_L}{\mathbf{q}_2^\top \mathbf{z}_L} & \frac{1}{2}\frac{\mathbf{q}_{L-1}^\top \mathbf{k}_{L-1}}{\mathbf{q}_2^\top \mathbf{z}_L} & \\ \vdots & \vdots & \ddots \\ \frac{\mathbf{q}_1^\top \mathbf{k}_L}{\mathbf{q}_1^\top \mathbf{z}_L} & \frac{\mathbf{q}_1^\top \mathbf{k}_{L-1}}{\mathbf{q}_1^\top \mathbf{z}_L} & \cdots & \frac{1}{2}\frac{\mathbf{q}_1^\top \mathbf{k}_1}{\mathbf{q}_1^\top \mathbf{z}_L} \end{pmatrix} \tag{65}$$

$$\mathbf{M}^B = \begin{pmatrix} 1 & \lambda_2 & \lambda_2\lambda_3 & \cdots & \lambda_2 \cdots \lambda_L \\ & 1 & \lambda_3 & \cdots & \lambda_3 \cdots \lambda_L \\ & & 1 & \cdots & \lambda_4 \cdots \lambda_L \\ & & & \ddots & \vdots \\ & & & & 1 \end{pmatrix} \rightarrow F(\mathbf{M}^B) = \begin{pmatrix} 1 & & & \\ \lambda_L & 1 & & \\ \lambda_L\lambda_{L-1} & \lambda_{L-1} & 1 & \\ \vdots & \vdots & \vdots & \ddots \\ \lambda_L \cdots \lambda_2 & \lambda_L \cdots \lambda_3 & \lambda_L \cdots \lambda_4 & \cdots & 1 \end{pmatrix} \tag{66}$$

The above can be achieved by:

$$F(\mathbf{A}) = \mathbf{J}_L \mathbf{A}\mathbf{J}_L, \ \ , \mathbf{J}_L = \begin{pmatrix} & & 1 \\ & 1 & \\ & \cdot^{\cdot^{\cdot}} & \\ 1 & & \end{pmatrix} \tag{67}$$

### B.5  LION **framework for diagonal decay**

The recurrence in the causal form for a diagonal case is presented as:

$$\mathbf{S}_i = \mathbf{\Lambda}_i \mathbf{S}_{i-1} + \mathbf{k}_i \mathbf{v}_i^\top \tag{68}$$

which followed by Dao and Gu [6], Hwang et al. [21] it can be shown in matrix form as:

$$\mathbf{Y} = \mathbf{P}\,\mathbf{V} \tag{69}$$

$$\mathbf{P}_{ij} = \mathbf{Q}_i^\top\,\mathbf{\Lambda}_i...\mathbf{\Lambda}_{j+1}\,\mathbf{K}_j = \mathbf{Q}_i^\top\,\mathbf{\Lambda}_{i:j}^\times\,\mathbf{K}_j \tag{70}$$

$$\mathbf{\Lambda}_{i:j}^\times = \begin{cases} \Pi_{k=j+1}^i \mathbf{\Lambda}_k, & i > j \\ 0, & i < j \end{cases} , \mathbf{\Lambda}_{i:i} = 1 \tag{71}$$

with $\mathbf{P}$ being the sequence mixer of the Linear Transformer. We can build Full Attention starting by decomposing the decay factros $\mathbf{\Lambda}_i$ as followes:

$$\mathbf{M}_{ij} = \mathbf{Q}_i^\top\,\mathbf{\Lambda}_i\ldots\mathbf{\Lambda}_{j+1}\,\mathbf{K}_j = \tag{72}$$
$$\mathbf{Q}_i^\top\,\mathbf{\Lambda}_0\mathbf{\Lambda}_1\ldots\mathbf{\Lambda}_i \times (\mathbf{\Lambda}_0)^{-1}(\mathbf{\Lambda}_1)^{-1}\ldots(\mathbf{\Lambda}_j)^{-1}\,\mathbf{K}_j$$
$$\mathbf{M}_{ij} = \mathbf{Q}_i^\top\,\mathbf{\Lambda}_{0:i}^\times(\mathbf{\Lambda}_{0:j}^\times)^{-1}\,\mathbf{K}_j \tag{73}$$

In the above decomposition blue parts are only depending on the index $i$ where the magenta depending on $j$ this implies by defining $\mathbf{Q}_i^* = \mathbf{\Lambda}\times_{0:i}\mathbf{Q}_i$ and $\mathbf{K}_j^* = (\mathbf{\Lambda}_{0:j}^\times)^{-1}\,\mathbf{K}_j$ the Equation equation 72 can be written exactly as attention in parallel form $\mathbf{M} = \mathbf{Q}^*\,\mathbf{K}^{*\top}$. Note that since $i > j$, all terms with indices $k < j$ cancel out, leaving only the matrices $\mathbf{\Lambda}_k$ with $j+1 \le k \le i$ to be multiplied in the expression $\mathbf{\Lambda}_{0:i}^\times(\mathbf{\Lambda}_{0:j}^\times)^{-1}$. However it is important that the matrices $\mathbf{Q}^*$ and $\mathbf{K}^*$ be fast and memory efficient to built we show that indeed they are starting by following remark:

**Remark.** For diagonal matrices $\mathbf{A}_0, \mathbf{A}_1, \ldots, \mathbf{A}_i$, the product $\mathbf{A}_{0:i}^\times = \mathbf{A}_0\,\mathbf{A}_1\cdots\mathbf{A}_i$ is given by

$$\mathbf{A}_{0:i} = \mathbf{a}_0 \odot \mathbf{a}_1 \odot \mathbf{a}_2 \odot \cdots \odot \mathbf{a}_i, \tag{74}$$

where $\mathbf{a}_i \in \mathbb{R}^N$ is a vector containing the diagonal elements of the matrix $\mathbf{A}_i \in \mathbb{R}^{N\times N}$ or $\mathbf{a}_i = \mathrm{DIAG}(\mathbf{A}_i)$.

Utilizing Remark , we can rewrite equation equation 72 as follows:

$$\mathbf{Y} = \mathbf{M}\,\mathbf{V}$$
$$\mathbf{M}_{ij} = \mathbf{Q}_i^\top\,\mathbf{\Lambda}_{0:i}^\times(\mathbf{\Lambda}_{0:j}^\times)^{-1}\,\mathbf{K}_j =$$
$$\mathbf{Q}_i^\top\,(\boldsymbol{\lambda}_0 \odot \boldsymbol{\lambda}_1 \odot \boldsymbol{\lambda}_2 \ldots \boldsymbol{\lambda}_i)(\boldsymbol{\lambda}_0 \odot \boldsymbol{\lambda}_1 \odot \boldsymbol{\lambda}_2 \ldots \boldsymbol{\lambda}_j)^{-1}\,\mathbf{K}_j =$$
$$(\mathbf{Q}_i \odot \boldsymbol{\lambda}_0 \odot \boldsymbol{\lambda}_1 \odot \boldsymbol{\lambda}_2 \ldots \boldsymbol{\lambda}_i)^\top(\boldsymbol{\lambda}_0 \odot \boldsymbol{\lambda}_1 \odot \boldsymbol{\lambda}_2 \ldots \boldsymbol{\lambda}_j)^{-1} \odot \mathbf{K}_j \tag{75}$$

where in the above, $(\boldsymbol{\lambda}_0 \odot \boldsymbol{\lambda}_1 \odot \boldsymbol{\lambda}_2 \ldots \boldsymbol{\lambda}_j)^{-1}$ indicates the element-wise inverse (Inv) of the vector $\boldsymbol{\lambda}_0 \odot \boldsymbol{\lambda}_1 \odot \boldsymbol{\lambda}_2 \ldots \boldsymbol{\lambda}_j$. During training, we have access to all $L$ tokens. Thus, we can compute the vectors $\boldsymbol{\lambda}_i$ for all $i$ and construct a matrix $\mathbf{D} \in \mathbb{R}^{L\times N}$, where the $i$-th row $\mathbf{D}_i = \boldsymbol{\lambda}_i = \mathrm{DIAG}(\mathbf{\Lambda}_i)$.

Our goal is to find a matrix $\mathbf{L} \in \mathbb{R}^{L\times N}$ whose $i$-th row $\mathbf{L}_i$ is given by $\boldsymbol{\lambda}_0 \odot \boldsymbol{\lambda}_1 \odot \boldsymbol{\lambda}_2 \odot \cdots \odot \boldsymbol{\lambda}_i$. This allows us to simplify the term $\mathbf{Q}_i \odot (\boldsymbol{\lambda}_0 \odot \boldsymbol{\lambda}_1 \odot \cdots \odot \boldsymbol{\lambda}_i)$ in equation 72 to $\mathbf{Q}_i \odot \mathbf{L}_i$.

Since $\mathbf{L}_{ip} = \prod_{k=0}^i \boldsymbol{\lambda}_{kp}$, it can be viewed as the cumulative product of the vectors $\boldsymbol{\lambda}_i$. Consequently, we can obtain $\mathbf{L}$ by applying a `cumprod` operation along the sequence dimension of $\mathbf{D}$:

$$\mathbf{L} = \texttt{cumprod}(\mathbf{D}, \texttt{dim} = 0). \tag{76}$$

A similar approach applies to $(\boldsymbol{\lambda}_0 \odot \boldsymbol{\lambda}_1 \odot \cdots \odot \boldsymbol{\lambda}_j)^{-1}$. Instead of computing the cumulative product of $\boldsymbol{\lambda}_i$ directly, we compute the cumulative product of the element-wise inverse of these vectors, leading to the definition of the matrix $\mathbf{U}$:

$$\mathbf{L}_i = \boldsymbol{\lambda}_0 \odot \boldsymbol{\lambda}_1 \odot \cdots \odot \boldsymbol{\lambda}_i \tag{77}$$

$$\mathbf{U}_i = (\boldsymbol{\lambda}_0 \odot \boldsymbol{\lambda}_1 \odot \cdots \odot \boldsymbol{\lambda}_i)^{-1} \tag{78}$$

Note that $\mathbf{U} = \mathrm{Inv}(\mathbf{L})$ is the element-wise inverse of $\mathbf{L}$, and thus $\mathbf{L} \odot \mathbf{U} = \mathbf{1}$, where $\mathbf{1} \in \mathbb{R}^{L \times N}$ is an all-ones matrix.

By analogy with Transformers [48], we can store the states $\mathbf{Q}_i, \mathbf{K}_i, \Lambda_i$ for all tokens in matrix form as $\mathbf{Q}, \mathbf{K}, \mathbf{D} \in \mathbb{R}^{L \times N}$. Under these conditions, a selective diagonal SSM (Equation equation 69) can be simplified to:

$$\mathbf{Y} = \mathrm{TRIL} \left[ \underbrace{(\mathbf{Q} \odot \mathbf{L})}_{\mathbf{Q}^*} \underbrace{(\mathbf{K} \odot \mathrm{Inv}(\mathbf{L}))^\top}_{\mathbf{K}^{*\top}} \right] \mathbf{V} \tag{79}$$

$$\mathbf{L} = \mathtt{cumprod}(\mathbf{D}), \quad \text{with } \mathbf{D}_i = \mathrm{DIAG}(\Lambda_i) \tag{80}$$

**Theorem B.1.** LION-*Diagonal Full Linear Attention: For bi-directional case the above alike Eq.* (11) *can be expressed as:*

$$\mathbf{Y} = \left( \mathrm{TRIL} \left[ (\mathbf{Q} \odot \mathbf{L}^F) \, (\mathbf{K} \odot \mathrm{Inv}(\mathbf{L}^F))^\top \right] + \mathrm{TRIU} \left[ (\mathbf{Q} \odot \mathbf{L}^B) \, (\mathbf{K} \odot \mathrm{Inv}(\mathbf{L}^B))^\top \right] \right) \mathbf{V} \tag{81}$$

$$\mathbf{L}^F = \mathtt{cumprod}(\mathbf{D}), \quad \text{with } \mathbf{D}_i = \mathrm{DIAG}(\Lambda_i) \tag{82}$$

$$\mathbf{L}^B = \mathtt{cumprod}(\text{Flip}(\mathbf{D})), \quad \text{with } \mathbf{D}_i = \mathrm{DIAG}(\Lambda_i) \tag{83}$$

*The* TRIL *function in the above formulation is indicating that the output* $\mathbf{Y}$ *is computed only from previous tokens, thereby preserving the causal structure analogous to causal Linear Transformers, where* $\mathbf{Y} = \mathrm{TRIL}(\mathbf{Q} \, \mathbf{K}^\top) \mathbf{V}$.

Note that this parallel form is less straightforward than the scalar case, but the masks $\mathbf{L}^F, \mathbf{L}^B$ can still be decomposed from the queries and keys as shown in Eq. (11). Theorem B.1 extends LION's Full Linear Attention to the diagonal decay case. While scaling can be applied directly to $\mathbf{Y}$, we omit it for simplicity, as most Linear Transformers with diagonal decay do not use attention scaling [13, 16, 33].

**Theorem B.2.** *(*LION-*RNN) The equivalent bi-directional RNN for Theorem* B.1 *is same as* LION-*RNN presented at Theorem* 3.1 *without scaling:*

$$\mathbf{S}_i^{F/B} = \Lambda_i \mathbf{S}_{i-1}^{F/B} + \mathbf{k}_i \mathbf{v}_i^\top, \quad \mathbf{y}_i^{F/B} = \mathbf{q}_i^\top \mathbf{S}_i^{F/B} - \frac{\mathbf{q}_i^\top \mathbf{k}_i}{2} \mathbf{v}_i, \quad \text{OUTPUT: } \mathbf{y}_i = \frac{\mathbf{y}_i^F + \mathbf{y}_i^B}{c_i^F + c_i^B} \tag{84}$$

## B.6 Mapping Existing autoregressive models into LION

As noted, other autoregressive recurrent models can also be integrated into our bidirectional framework, benefiting from parallelization during training and fast bidirectional inference. Here, we demonstrate how to map several well-known Linear Transformers into the bidirectional form of LION, along with their corresponding masked attention matrix and inference linear recurrence.

**Linear Transformer (**LION-LIT**).** According to [25] the linear transformer has a recurrence:

$$\mathbf{S}_i^F = \mathbf{S}_{i-1}^F + \mathbf{k}_i \mathbf{v}_i^\top, \tag{85}$$

$$\mathbf{z}_i^F = \mathbf{z}_{i-1}^F + \mathbf{k}_i, \tag{86}$$

$$\mathrm{SCALED} : \mathbf{y}_i^F = \frac{\mathbf{q}_i^\top \mathbf{S}_i^F}{\mathbf{q}_i^\top \mathbf{z}_i^F} \tag{87}$$

$$\mathrm{NON\text{-}SCALED} : \mathbf{y}_i^F = \mathbf{q}_i^\top \mathbf{S}_i^F \tag{88}$$

As observed, this is a special case of our bidirectional recurrence defined in equation 19 with $\lambda_i = 1$, as LION resembles the scaled masked attention. In the case of the linear transformer, we require attention without scaling for the recurrence. The vectorized form for the scaled version can then be derived easily as follows:

$$\mathbf{S}_i^{F/B} = \mathbf{S}_{i-1}^{F/B} + \mathbf{k}_i \mathbf{v}_i^\top, \tag{89}$$

$$\mathbf{z}_i^{F/B} = \mathbf{z}_{i-1}^{F/B} + \mathbf{k}_i \tag{90}$$

$$c_i^{F/B} = \mathbf{q}_i^\top \mathbf{z}_i^{F/B} - \frac{1}{2}\mathbf{q}_i^\top \mathbf{k}_i, \tag{91}$$

$$\mathbf{y}_i^{F/B} = \mathbf{q}_i^\top \mathbf{S}_i^{F/B} - \frac{1}{2}\mathbf{q}_i^\top \mathbf{k}_i \mathbf{v}_i \tag{92}$$

$$= \qquad \mathbf{Y} = \text{SCALE}(\mathbf{QK}^\top \mathbf{V}) \tag{93}$$

For the non-scaled variant, we simply remove the scaling state $\mathbf{z}$ as well as the scaling parameter $c$. Consequently, the bidirectional linear transformer, which is equivalent to and parallelizable with attention without scaling, can be expressed as follows:

$$\mathbf{S}_i^{F/B} = \mathbf{S}_{i-1}^{F/B} + \mathbf{k}_i \mathbf{v}_i^\top, \tag{94}$$

$$\mathbf{y}_i^{F/B} = \mathbf{q}_i^\top \mathbf{S}_i^{F/B} - \frac{1}{2}\mathbf{q}_i^\top \mathbf{k}_i \mathbf{v}_i \tag{95}$$

$$= \qquad \mathbf{Y} = \mathbf{QK}^\top \mathbf{V} \tag{96}$$

The final output for scaled version can be extracted as $\mathbf{y}_i = \frac{\mathbf{y}_i^B + \mathbf{y}_i^B}{c_i^B + c_i^B}$ for scaled and as $\mathbf{y}_i = \mathbf{y}_i^B + \mathbf{y}_i^B$ for non-scaled version. Variations of linear transformers, such as Performer [5], which employ different non-linearities $\phi(.)$ for keys and queries, can be adapted to a bidirectional format using the framework established for linear transformers.

**Retentive Network** (LION-D). According to [44] the forward equation for a retentive network can be written as:

$$\mathbf{S}_i^F = \lambda \mathbf{S}_{i-1}^F + \mathbf{k}_i \mathbf{v}_i^\top, \tag{97}$$

$$\mathbf{y}_i^F = \mathbf{q}_i^\top \mathbf{S}_i^F \tag{98}$$

This architecture can also be expanded to bi-directional setting simply by not scaling the attention in our framework and only using the mask with non input-dependent $\lambda_i = \lambda$ values:

$$\mathbf{S}_i^{F/B} = \lambda \mathbf{S}_{i-1}^{F/B} + \mathbf{k}_i \mathbf{v}_i^\top, \tag{99}$$

$$\mathbf{y}_i^{F/B} = \mathbf{q}_i^\top \mathbf{S}_i^{F/B} - \frac{1}{2}\mathbf{q}_i^\top \mathbf{k}_i \mathbf{v}_i \tag{100}$$

$$= \qquad \mathbf{Y} = (\mathbf{QK}^\top \odot \mathbf{M}^R)\mathbf{V} \tag{101}$$

Note that: $\mathbf{M}_{ij}^R = \lambda^{|i-j|}$, and LION-D uses scaled version of above.

**xLSTM** (LION-LSTM). According to [3] the recurrence for forward recurrence of xLSTM can be written as:

$$\mathbf{S}_i^F = f_i \mathbf{S}_{i-1}^F + i_i \mathbf{k}_i \mathbf{v}_i^\top, \tag{102}$$

$$\mathbf{z}_i^F = f_i \mathbf{z}_{i-1}^F + i_i \mathbf{k}_i, \tag{103}$$

$$\mathbf{y}_i^F = \frac{\mathbf{q}_i^\top \mathbf{S}_i^F}{\mathbf{q}_i^\top \mathbf{z}_i^F} \tag{104}$$

The above recurrence is equivalent to equation 19 by considering $i_i \mathbf{k}_i$ as a new key. The term $i_i \mathbf{k}_i$ can be easily vectorized by aggregating all $i_i$ values for each token into a vector $\mathbf{i}$. Thus, we can express the vectorized form of the bidirectional xLSTM and its equivalence to attention as follows:

$$\mathbf{S}_i^{F/B} = f_i\mathbf{S}_{i-1}^{F/B} + i_i\mathbf{k}_i\mathbf{v}_i^\top, \quad (105)$$

$$\mathbf{z}_i^{F/B} = f_i\mathbf{z}_{i-1}^{F/B} + i_i\mathbf{k}_i \quad (106)$$

$$c_i^{F/B} = \mathbf{q}_i^\top\mathbf{z}_i^{F/B} - \frac{1}{2}\mathbf{q}_i^\top\mathbf{k}_i, \quad (107)$$

$$\mathbf{y}_i^{F/B} = \mathbf{q}_i^\top\mathbf{S}_i^{F/B} - \frac{1}{2}\mathbf{q}_i^\top\mathbf{k}_i\mathbf{v}_i \quad (108)$$

$$\text{Output: } \mathbf{y}_i = \frac{\mathbf{y}_i^F + \mathbf{y}_i^B}{\max(c_i^F + c_i^B, 1)} \quad (109)$$

$$= \quad \mathbf{Y} = \text{SCALE}_{max}(\mathbf{Q}(\mathbf{i} \odot \mathbf{K}^\top)) \odot \mathbf{M}^f)\mathbf{V} \quad (110)$$

where the mask $\mathbf{M}^f$ is equal to the LION mask equation 55 just by replacing $\lambda_i = f_i$. And where operation $\text{SCALE}_{max}$ consider the maximum of operation in the denominator as:

$$\text{SCALE}_{max}(\mathbf{A})_{ij} = \frac{\mathbf{A}_{ij}}{\max(\sum_{j=1}^L \mathbf{A}_{ij}, 1)} \quad (111)$$

**Gated RFA** (LION-S) Gated RFA [51] in autoregressive mode exhibits a recurrence similar to that of xLSTM, with only minor differences:

$$\mathbf{S}_i^F = g_i\mathbf{S}_{i-1}^F + (1 - g_i)\mathbf{k}_i\mathbf{v}_i^\top, \quad (112)$$

$$\mathbf{z}_i^F = g_i\mathbf{z}_{i-1}^F + (1 - g_i)\mathbf{k}_i, \quad (113)$$

$$\mathbf{y}_i^F = \frac{\mathbf{q}_i^\top\mathbf{S}_i^F}{\mathbf{q}_i^\top\mathbf{z_i}^F} \quad (114)$$

Thus, the bidirectional version of the model retains a similar output, achieved by replacing the vector $\mathbf{i}$ in equation 110 with $1 - \mathbf{g}$, where $\mathbf{g}$ represents the vectorized form of all scalar values $g_i$.

$$\mathbf{S}_i^{F/B} = g_i\mathbf{S}_{i-1}^{F/B} + (1 - g_i)\mathbf{k}_i\mathbf{v}_i^\top, \quad (115)$$

$$\mathbf{z}_i^{F/B} = g_i\mathbf{z}_{i-1}^{F/B} + (1 - g_i)\mathbf{k}_i \quad (116)$$

$$c_i^{F/B} = \mathbf{q}_i^\top\mathbf{z}_i^{F/B} - \frac{1}{2}\mathbf{q}_i^\top\mathbf{k}_i, \quad (117)$$

$$\mathbf{y}_i^{F/B} = \mathbf{q}_i^\top\mathbf{S}_i^{F/B} - \frac{1}{2}\mathbf{q}_i^\top\mathbf{k}_i\mathbf{v}_i \quad (118)$$

$$= \quad \mathbf{Y} = \text{SCALE}(\mathbf{Q}((1 - \mathbf{g}) \odot \mathbf{K}^\top) \odot \mathbf{M})\mathbf{V} \quad (119)$$

Note that for LION-S we just use different non-linearty compared to original GRFA and the term $(1 - g_i)$ can be considered as part of $\mathbf{k}_i$.

**Mamba2** Mamba2 [6] is the selective SSM with scalar decay factor $\lambda_i = \text{SoftPlus}(\mathbf{W}\mathbf{x_i})$ we can observe that Mamba simply can be considered as variant of LION-S with different activation and without scaling therefor we can extend it to bi-directional setting via:

$$\mathbf{S}_i^{F/B} = \lambda_i\mathbf{S}_{i-1}^{F/B} + \mathbf{k}_i\mathbf{v}_i^\top, \quad (120)$$

$$\mathbf{y}_i^{F/B} = \mathbf{q}_i^\top\mathbf{S}_i^{F/B} - \frac{1}{2}\mathbf{q}_i^\top\mathbf{k}_i\mathbf{v}_i \quad (121)$$

$$= \quad \mathbf{Y} = (\mathbf{Q}\mathbf{K}^\top \odot \mathbf{M})\mathbf{V} \quad (122)$$

Here, $\mathbf{M}$ is the selective mask used in LION-S. This variant is closely related to Hydra [21], but differs architecturally. Hydra incorporates convolutions and gating, and uses SSDs for training, whereas LION-S trains using full attention without any gating or convolutions.

**RWKV-6**: RWKV-6 [33] is a Linear Transformer with diagonal decay it causal recurrence can be written as:

$$\mathbf{S}_i = \boldsymbol{\Lambda_i}\mathbf{S}_{i-1} + \mathbf{k}_i\mathbf{v}_i^\top, \tag{123}$$

$$\mathbf{y}_i = \mathbf{q}_i^\top\mathbf{S}_i \tag{124}$$

as seen this choice is exactly aligned with our proof on Theorem B.1 which allows the bi-directional form of:

$$
\begin{aligned}
\mathbf{S}_i^{F/B} &= \Lambda_i\mathbf{S}_{i-1}^{F/B} + \mathbf{k}_i\mathbf{v}_i^\top, \quad (125)\\
\mathbf{y}_i^{F/B} &= \mathbf{q}_i^\top\mathbf{S}_i^{F/B} - \frac{1}{2}\mathbf{q}_i^\top\mathbf{k}_i\mathbf{v}_i \\
&\hspace{4cm}(126)
\end{aligned}
=
\begin{aligned}
\mathbf{Y} &= \left(\textsc{Tril}\left[(\mathbf{Q}\odot\mathbf{L}^F)\,(\mathbf{K}\odot\mathrm{Inv}(\mathbf{L}^F))^\top\right] + \right.\\
&\left.\textsc{Triu}\left[(\mathbf{Q}\odot\mathbf{L}^B)\,(\mathbf{K}\odot\mathrm{Inv}(\mathbf{L}^B))^\top\right]\right)\mathbf{V} \\
&\hspace{6cm}(127)
\end{aligned}
$$

**HGRN-2**: HGRN-2 [36] has similar recurrence as RWKV-6 just by replacing the $\mathbf{k}_i = 1 - \mathbf{Diag}(\boldsymbol{\Lambda}_i)$ therefor its equal bi-directional form via LION is alike Appendix B.6.

### B.7    Discussion on DeltaNet

For the special case of DeltaNet [52], our theorem supports an equivalent bidirectional RNN formulation with LION-style correction terms to avoid double counting:

$$\mathbf{S_i^{F/B}} = \mathbf{S_{i-1}^{F/B}}\left(\mathbf{I} - \beta_\mathbf{i}\mathbf{k_i}\mathbf{k_i^\top}\right) + \beta_\mathbf{i}\mathbf{k_i}\mathbf{v_i^\top}, \quad \mathbf{y_i^{F/B}} = \mathbf{q_i^\top}\mathbf{S_i^{F/B}} - \tfrac{1}{2}\mathbf{q_i^\top}\mathbf{k_i}\mathbf{v_i^\top}.$$

The chunkwise parallel form of DeltaNet (Eqs. 8–11 in the original paper) can also be extended to the bidirectional setting by removing causal masks and applying Theorem 3.2 for the chunkwise form of LION. However, a key bottleneck in DeltaNet arises in its full attention formulation, where the attention is computed as

$$\mathbf{A} = \mathbf{Q}\mathbf{K}^\top\mathbf{T},$$

with $T$ (Eq. 10 in the original paper) requiring a matrix inverse that scales cubically with sequence length. This inverse undermines the fast-training advantage of LION, so we avoid using the fully parallel form for training DeltaNet.

### B.8    Mask $\mathrm{M}^F\&\mathrm{M}^B$ are Semiseperable with rank-1

For the lower triangular part of the selective mask $\mathbf{M}^F$, the upper triangular part can be filled such that it creates a full matrix with rank 1, which aligns with the definition of a semi-separable matrix with rank 1, as below:

where TRIL is the function which masks the upper part of the matrix and set it to zero. Same is applied for the upper triangular part of the matrix $\mathbf{M}^B$ as well. Also since decay mask is the specific case of selective mask by setting $\lambda_i = \lambda$ all the proofs above also holds for the fixed decay mask used in RetNet.

### B.9    Details for LION of Chunkwise Parallel

As the full linear attention is written as:

$$\mathbf{Y} = \textsc{Scale}(\mathbf{Q}\mathbf{K}^\top\odot\mathbf{M})\mathbf{V} \tag{128}$$

by apply chunking to queries/keys/values and defining the:

$$\mathbf{Q}_{[i]}, \mathbf{K}_{[i]}, \mathbf{V}_{[i]} = \mathbf{Q}_{iC+1:i(C+1)}, \mathbf{K}_{iC+1:i(C+1)}, \mathbf{V}_{iC+1:i(C+1)} \in \mathbb{R}^{C\times d}$$

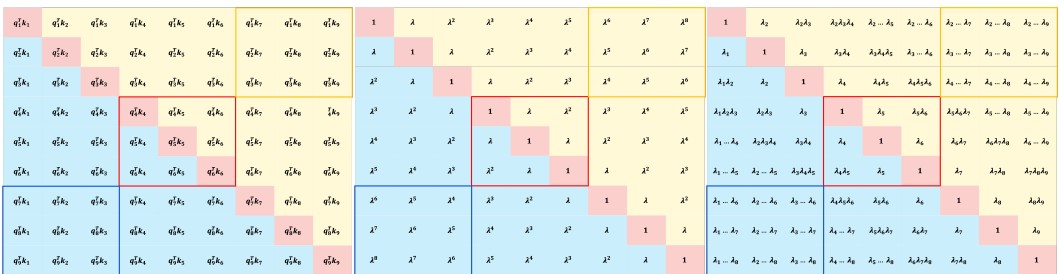

Figure 8: Visualization of attention (Left), LION-D (Middle), and LION-S (Right) chunkwise forms.

we can simply rewrite the Eq. (128) in chunkwise form as:

$$\mathbf{A}_{[ij]} = \mathbf{Q}_{[i]}\mathbf{K}_{[j]}^\top \odot \mathbf{M}_{[ij]}, \quad \mathbf{C}_{[ij]} = \mathbf{C}_{[ij-1]} + \text{Sum}(\mathbf{A}_{[ij]}), \tag{129}$$

$$\mathbf{S}_{[ij]} = \mathbf{S}_{[ij-1]} + \mathbf{A}_{[ij]}\mathbf{V}_{[j]}, \quad \mathbf{Y}_{[i]} = \frac{\mathbf{S}_{[iN]}}{\mathbf{C}_{[iN]}} \tag{130}$$

where $N$ is the number of total chunks and $N = \frac{L}{C}$ and Sum is the summation over the rows of the matrix. Since the full attention matrix needs to be scaled according to the full row of attention we need to update the scaling value for each chunk as stored in $\mathbf{C}_{ij}$ and the final output for chunk $i$ is computed by using the last chunkwise hidden state $\mathbf{S}_{[i]}$ divided by the scaling for that chunk $\mathbf{C}_{[i]}$ which are equal to $\mathbf{S}_{[iN]}, \mathbf{C}_{[iN]}$.

To construct the chunkwise mask $\mathbf{M}_{ij}$, we define the chunk-level selective parameters as:

$$\mathbf{L}_{[i]}^F = \texttt{cumprod}(\lambda^{\mathbf{F}})_{iC+1:(i+1)C}, \quad \mathbf{L}_{[i]}^B = \texttt{cumprod}(\lambda^{\mathbf{B}})_{iC+1:(i+1)C}.$$

Since the full mask is composed of lower and upper triangular components:

$$\mathbf{M}^F = \text{TRIL}(\mathbf{L}^F \frac{1}{\mathbf{L}^F}), \quad \mathbf{M}^B = \text{TRIU}(\mathbf{L}^B \frac{1}{\mathbf{L}^B}),$$

we determine the appropriate chunkwise form based on relative chunk positions:

- If $i > j$, the chunk falls entirely within the lower triangular part, requiring only $\mathbf{M}^F$, which can be efficiently computed as $\mathbf{L}_{[i]}^F \frac{1}{\mathbf{L}_{[j]}^F}$.

- If $i < j$, the chunk is fully in the upper triangular region, needing only $\mathbf{M}^B$, which follows from $\mathbf{L}_{[j]}^B \frac{1}{\mathbf{L}_{[i]}^B}$.

- If $i = j$, the chunk lies along the diagonal and requires both the lower triangular part of $\mathbf{M}^F$ and the upper triangular part of $\mathbf{M}^B$, expressed as:

The full matrix is:

$$\mathbf{M}_{[ij]} = \begin{cases} \mathbf{L}_{[i]}^F \frac{1}{\mathbf{L}_{[j]}^F}^\top & \text{if } i > j, \\ \mathbf{L}_{[j]}^B \frac{1}{\mathbf{L}_{[i]}^B}^\top & \text{if } i < j, \\ \text{Tril}\left(\mathbf{L}_{[i]}^F \frac{1}{\mathbf{L}_{[i]}^F}^\top\right) + \text{Triu}\left(\mathbf{L}_{[i]}^B \frac{1}{\mathbf{L}_{[i]}^B}^\top\right) - \mathbf{I} & \text{if } i = j. \end{cases}$$

For the fixed decay mask, a simpler case of the general selective mask, $\mathbf{L}^F$ and $\mathbf{L}^B$ are identical and simplify to $\mathbf{L}_i = \lambda^i$. Since the full mask follows $\mathbf{M}_{ij} = \lambda^{|i-j|}$, the chunkwise mask for $i, j$ can be written as:

$$\mathbf{M}_{[ij]} = \mathbf{L}_{[i]} \frac{1}{\mathbf{L}_{[j]}} = \lambda^{|i-j|} \mathbf{L}_{[0]} \frac{1}{\mathbf{L}_{[0]}}.$$

Similarly, for the upper triangular part:

$$\mathbf{M}_{[ij]} = \lambda^{|i-j|} \frac{1}{\mathbf{L}_{[0]}^{\top}} \mathbf{L}_{[0]}.$$

For diagonal chunks, the mask remains a fixed matrix $\mathbf{\Gamma} \in \mathbb{R}^{C \times C}$, where $\mathbf{\Gamma}_{ij} = \lambda^{|i-j|}$, representing a smaller version of the full fixed decay mask $\mathbf{M} \in \mathbb{R}^{L \times L}$ with $\mathbf{M}_{ij} = \lambda^{|i-j|}$. The visualization of chunking is presented in Appendix B.9. .

## B.10 Parallel Chunkwise Materialization for Parallel Output Computation

As mentioned in Section 3.3, Eq. (129) can be further parallelized by processing all $i$ tokens simultaneously, leading to the following matrix multiplications in the parallel chunkwise form:

$$\mathbf{A}_{[i]} = \mathbf{Q}\mathbf{K}_{[j]}^{\top} \odot \mathbf{M}_{[j]}, \quad \mathbf{C}_{[j]} = \mathbf{C}_{[j-1]} + \mathrm{Sum}(\mathbf{A}_{[j]}), \tag{131}$$

$$\mathbf{S}_{[j]} = \mathbf{S}_{[j-1]} + \mathbf{A}_{[ij]}\mathbf{V}_{[j]}, \quad \mathbf{Y} = \frac{\mathbf{S}_{[N]}}{\mathbf{C}_{[N]}} \tag{132}$$

and the mask $\mathbf{M}_{[j]}$ is created based on:

$$\mathbf{M}_{[j]} = \mathrm{Tril}(\mathbf{L}^F \frac{1}{\mathbf{L}^F_{[j]}^{\top}}, \mathbf{diagonal} = -j) + \mathrm{Tril}(\mathbf{L}^B \frac{1}{\mathbf{L}^B_{[j]}^{\top}}, \mathbf{diagonal} = j) \tag{133}$$

Consider a real matrix $X \in \mathbb{R}^{n \times n}$. The operator $\mathrm{Tril}(\mathbf{X}, d)$ returns the lower-triangular region of $X$, including all elements on and below the diagonal shifted by $d$. In other words, $\mathrm{tril}(\mathbf{X}, d)_{ij} = \mathbf{X}_{ij}$ whenever $j \leq i + d$ and is 0 otherwise. Similarly, $\mathrm{Triu}(\mathbf{X}, d)$ returns the upper-triangular region, keeping elements on and above the diagonal shifted by $d$. Formally, $\mathrm{Triu}(\mathbf{X}, d)_{ij} = \mathbf{X}_{ij}$ if $j \geq i + d$ and 0 otherwise.

## B.11 Zero-Order Hold Discretization

Below we explain the zero-order hold discretization derived by [24]. An LTI system can be represented with the equation:

$$\mathbf{h}'(t) = A\mathbf{h}(t) + B\mathbf{x}(t), \tag{134}$$

which can be rearranged to isolate $\mathbf{h}(t)$:

$$\mathbf{h}'(t) - A\mathbf{h}(t) = B\mathbf{x}(t). \tag{135}$$

By multiplying the equation by $e^{-At}$, we get

$$e^{-At}\mathbf{h}'(t) - e^{-At}A\mathbf{h}(t) = e^{-At}B\mathbf{x}(t) \tag{136}$$

Since $\frac{\partial}{\partial t}e^{At} = Ae^{At} = e^{At}A$, Eq. (136) can be written as:

$$\frac{\partial}{\partial t}\left(e^{-At}\mathbf{h}(t)\right) = e^{-At}B\mathbf{x}(t). \tag{137}$$

After integrating both sides and simplifications, we get

$$e^{-At}\mathbf{h}(t) = \int_0^t e^{-A\tau}B\mathbf{x}(\tau)\,d\tau + \mathbf{h}(0). \tag{138}$$

By multiplying both sides by $e^{At}$ to isolate $\mathbf{h}(t)$ and performing further simplifications, at the end we get

$$\mathbf{h}(t) = e^{At} \int_0^t e^{-A\tau} B\mathbf{x}(\tau)\, d\tau + e^{At}\mathbf{h}(0). \tag{139}$$

To discretize this solution, we can assume sampling the system at even intervals, i.e. each sample is at $kT$ for some time step $T$, and that the input $\mathbf{x}$(t) is constant between samples. To simplify the notation, we can define $\mathbf{h}_k$ in terms of $\mathbf{h}(kT)$ such that

$$\mathbf{h}_k = \mathbf{h}(kT). \tag{140}$$

Using the new notation, Eq. (139) becomes

$$\mathbf{h}_k = e^{\mathbf{A}kT}\mathbf{h}(0) + e^{\mathbf{A}kT} \int_0^{kT} e^{-\mathbf{A}\tau}\mathbf{B}\mathbf{x}(\tau)\, d\tau. \tag{141}$$

Now we want to express the system in the form:

$$\mathbf{h}_{k+1} = \tilde{\mathbf{A}}\mathbf{h}_k + \tilde{\mathbf{B}}\mathbf{x}_k. \tag{142}$$

To start, let's write out the equation for $\mathbf{x}_{k+1}$ as

$$\mathbf{h}_{k+1} = e^{\mathbf{A}(k+1)T}\mathbf{h}(0) + e^{\mathbf{A}(k+1)T} \int_0^{(k+1)T} e^{-\mathbf{A}\tau}\mathbf{B}\mathbf{x}(\tau)\, d\tau. \tag{143}$$

After multiplying by $e^{\mathbf{A}T}$ and rearranging we get

$$e^{\mathbf{A}(k+1)T}\mathbf{h}(0) = e^{\mathbf{A}T}\mathbf{h}_k - e^{\mathbf{A}(k+1)T} \int_0^{kT} e^{-\mathbf{A}\tau}\mathbf{B}\mathbf{x}(\tau)\, d\tau. \tag{144}$$

Plugging this expression for $\mathbf{x}_{k+1}$ in Eq. (143) yields to

$$\mathbf{h}_{k+1} = e^{\mathbf{A}T}\mathbf{h}_k - e^{\mathbf{A}(k+1)T} \left( \int_0^{kT} e^{-\mathbf{A}\tau}\mathbf{B}\mathbf{x}(\tau)\, d\tau + \int_0^{(k+1)T} e^{-\mathbf{A}\tau}\mathbf{B}\mathbf{x}(\tau)\, d\tau \right), \tag{145}$$

which can be further simplified to

$$\mathbf{h}_{k+1} = e^{\mathbf{A}T}\mathbf{h}_k - e^{\mathbf{A}(k+1)T} \int_{kT}^{(k+1)T} e^{-\mathbf{A}\tau}\mathbf{B}\mathbf{x}(\tau)\, d\tau. \tag{146}$$

Now, assuming that $\mathbf{x}(t)$ is constant on the interval $[kT, (k+1)T)$, which allows us to take $\mathbf{B}\mathbf{x}(t)$ outside the integral. Moreover, by bringing the $e^{\mathbf{A}(k+1)T}$ term inside the integral we have

$$\mathbf{h}_{k+1} = e^{\mathbf{A}T}\mathbf{h}_k - \int_{kT}^{(k+1)T} e^{\mathbf{A}((k+1)T-\tau)}\, d\tau\, \mathbf{B}\mathbf{x}_k. \tag{147}$$

Using a change of variables $v = (k+1)T - \tau$, with $d\tau = -dv$, and reversing the integration bounds results in

$$\mathbf{h}_{k+1} = e^{\mathbf{A}T}\mathbf{h}_k + \int_0^T e^{\mathbf{A}v}\, dv\, \mathbf{B}\mathbf{x}_k. \tag{148}$$

Finally, if we evaluate the integral by noting that $\frac{d}{dt}e^{\mathbf{A}t} = \mathbf{A}e^{\mathbf{A}t}$ and assuming $\mathbf{A}$ is invertible, we get

$$\mathbf{h}_{k+1} = e^{\mathbf{A}T}\mathbf{h}_k + \mathbf{A}^{-1}\left(e^{\mathbf{A}T} - \mathbf{I}\right)\mathbf{B}\mathbf{x}_k. \tag{149}$$

Thus, we find the discrete-time state and input matrices:

$$\tilde{\mathbf{A}} = e^{\mathbf{A}T} \tag{150}$$

$$\tilde{\mathbf{B}} = \mathbf{A}^{-1}\left(e^{\mathbf{A}T} - \mathbf{I}\right)\mathbf{B}. \tag{151}$$

And the final desecrate state space representation is:

$$\mathbf{h_k} = e^{\mathbf{A}T}\mathbf{h}_{k-1} + \mathbf{A}^{-1}\left(e^{\mathbf{A}T} - \mathbf{I}\right)\mathbf{B}_k\mathbf{x}_k. \tag{152}$$

As in case of LION-S (similar to choice of Mamba2 [6]) the matrix $\mathbf{A}$ is identity while the time step $T$ is selective and equal to $a_i$. And simply for LION-S scenario the term $Bx(t)$ will change into $\mathbf{k}_i \mathbf{v}_i^\top$ therefor considering Linear Transformer as continuous system like:

$$\mathbf{S}'_{(t)} = \mathbf{S}_{(t)} + \mathbf{k}_{(t)} \mathbf{v}_{(t)}^\top, \tag{153}$$

$$\mathbf{z}_{(t)} = \mathbf{z}_{(t)} + \mathbf{k}_{(t)}, \tag{154}$$

$$\tag{155}$$

By applying the ZOH discritization the final descreate LION-S will be equal to:

DISCRETE

$$\mathbf{S}_i = e^{a_i} \mathbf{S}_{i-1} + (e^{a_i} - 1)\mathbf{k}_i \mathbf{v}_i^\top, \tag{156}$$

$$\mathbf{z}_i = e^{a_i} \mathbf{z}_{i-1} + (e^{a_i} - 1)\mathbf{k}_i, \tag{157}$$

And it applies to both directions forward and backward.

## C   Additional experimental validation

### C.1   LRA full dataset results for LION

Table 5: *Training Stability on Long Range Arena Task.* Our family of models can solve the LRA benchmark with the appropriate initialization using full attention.

| Model (input length) | ListOps 2048 | Text 4096 | Retrieval 4000 | Image 1024 | Pathfinder 1024 | PathX 16K | Avg. |
|---|---|---|---|---|---|---|---|
| LION-LIT | 16.78 | 65.21 | 54.00 | 43.29 | 72.78 | ✗ | 50.41 |
| LION-D (w/o *HIPPO*) | 32.5 | 64.5 | 50.0 | 47.4 | 73.6 | ✗ | 53.6 |
| LION-S (w/o *HIPPO*) | 36.34 | 60.87 | 55.0 | 42.6 | 74.2 | ✗ | 53 |
| LION-D (w/ *HIPPO*) | 62.0 | 88.78 | 90.12 | 85.66 | 90.25 | 97.28 | 85.63 |
| LION-S (w/ *HIPPO*) | 62.25 | 88.10 | 90.35 | 86.14 | 91.30 | 97.99 | **86.07** |

### C.2   LRA Configurations for LION

For the LRA task, we utilized the same model dimensions as specified in the S5 [43] paper, following the guidelines from the S5 GitHub repository[5]. Our state matrix was represented as a vector $\mathbf{\Lambda}_i = \boldsymbol{\lambda}_i$, where each element contains a scalar non-input dependent value $e^a$. The value $a$ was initialized based on *HIPPO* theory, alongside the input-dependent $a_i$, as described in main body.

We employed the ADAMW optimizer with an initial learning rate of $5 \times 10^{-4}$ and a cosine learning rate scheduler [30]. The weights for the queries and keys, as well as the selective component of $\Lambda$, were initialized using a Gaussian distribution with a standard deviation of 0.1. For the values $\mathbf{v}$, we initialized $W_\mathbf{v}$ using zero-order hold discretization, represented as $W_\mathbf{v}^{\text{init}} = \left(\Lambda^{-1} \cdot (\Lambda - \mathbf{I})\right)$. The non-selective parts of $\Lambda$ were initialized based on the *HIPPO* [43] matrix.

### C.3   Ablation Studies on LRA dataset

We have did an ablation study for choosing the activation functions and using non-scalar decay factor for LION.

We have observed that bounding the keys and queries significantly enhances the model's ability to solve tasks. This finding is consistent with the observations in [52]. As demonstrated in variation [1], it can successfully tackle the LRA task even without scaling, while the RELU activation fails to do so. Additionally, we found that scaling plays a crucial role, particularly when it comes to scaling the masked attention. The approach used in LION, which scales the attention before applying the mask expressed as $\mathbf{Y} = \text{SCALE}(\mathbf{Q}\mathbf{K}^\top) \odot \mathbf{M}$ has proven ineffective in addressing the challenging PathX task, as shown in Table 6 [5].

---

[5]https://github.com/lindermanlab/S5

Table 6: *Effects of different parameter choices and non-linearities in* LION-S *on LRA tasks.* Codes: [1] Sigmoid non-linearity was applied to the **k** and **q** values with unscaled masked attention; [2] ReLU non-linearity was utilized, and the masked attention was scaled; [3] The parameter $a_i$ was selected as a scalar instead of a vector; [4] LION-S model parameters were used without scaling; [5] The attention matrix of LION-S was scaled, but attention values were adjusted without the factor of $\lambda_i$; [6] The selective component of $a_i$ was removed; [7] SoftPlus activation function was employed for the $a_i$ values. We used the HIPPO [14] initialisation for LRA task since random initalisation of LION-S and LION-D can not solve LRA.

| Model (input length) | ListOps 2048 | Text 2048 | Retrieval 4000 | Image 1024 | Pathfinder 1024 | PathX 16K | Avg. |
|---|---|---|---|---|---|---|---|
| [1] $\phi(x) = \sigma(x)$ w.o scaling | 61.02 | 88.02 | 89.10 | 86.2 | 91.06 | 97.1 | 85.41 |
| [2] $\phi(x) = \text{RELU}(x)$ w. scaling | 36.37 | 65.24 | 58.88 | 42.21 | 69.40 | ✗ | 54.42 |
| [3] $a_i$ only scalar | 36.23 | 60.33 | 60.45 | 58.89 | 70.00 | ✗ | 57.17 |
| [4] LION w.o scaling | 58.76 | 67.22 | 59.90 | 60.0 | 65.51 | ✗ | 62.27 |
| [5] scaled attention w.o mask | 60.12 | 87.67 | 87.42 | 88.01 | 89.23 | ✗ | 82.49 |
| [6] $a_i$ From *HIPPO* w.o selectivity | 60.12 | 88.00 | 89.22 | 83.21 | 91.0 | 96.30 | 84.64 |
| [7] $a_i = \text{SOFTPLUS}(x)$ | 16.23 | 59.90 | 60.00 | 45.12 | 70.07 | ✗ | 50.26 |
| LION-S (w/ *HIPPO*) | **62.25** | **88.10** | **90.35** | **86.14** | **91.30** | **97.99** | **86.07** |

## C.4 LRA full Results

We have evaluated LION variants against benchmarks for long-range sequence modeling across different categories, including softmax-based Transformers, RNNs, SSMs, and Linear Transformers.

Table 7: *Performance on Long Range Arena Tasks.* For each column (dataset), the best and the second best results are highlighted with **bold** and underline respectively. Note that the MEGA architecture has roughly $10\times$ the number of parameters as the other architectures.

| Category | Model (input length) | ListOps 2048 | Text 4096 | Retrieval 4000 | Image 1024 | Pathfinder 1024 | PathX 16K | Avg. |
|---|---|---|---|---|---|---|---|---|
| Transformer | Transformer | 36.37 | 64.27 | 57.46 | 42.44 | 71.40 | ✗ | 54.39 |
| | MEGA ($\mathcal{O}(L^2)$) | **63.14** | **90.43** | 91.25 | **90.44** | 96.01 | 97.98 | **88.21** |
| | MEGA-chunk ($\mathcal{O}(L)$) | 58.76 | 90.19 | 90.97 | 85.80 | 94.41 | 93.81 | 85.66 |
| SSM | DSS | 57.60 | 76.60 | 87.60 | 85.80 | 84.10 | 85.00 | 79.45 |
| | S4 (original) | 58.35 | 86.82 | 89.46 | 88.19 | 93.06 | 96.30 | 85.36 |
| | S5 | 62.15 | 89.31 | **91.40** | 88.00 | 95.33 | **98.58** | 87.46 |
| | Mamba (From [3]) | 32.5 | N/A | 90.2 | 68.9 | **99.2** | N/A | N/A |
| RNN | LRU | 60.2 | 89.4 | 89.9 | 89.0 | 95.1 | 94.2 | 86.3 |
| | xLSTM (From [3]) | 41.1 | N/A | 90.6 | 69.5 | 91.9 | N/A | N/A |
| Linear Transformer | Local Att. | 15.82 | 52.98 | 53.39 | 41.46 | 66.63 | ✗ | 46.06 |
| | Sparse Transformer | 17.07 | 63.58 | 59.59 | 44.24 | 71.71 | ✗ | 51.24 |
| | Longformer | 35.63 | 62.85 | 56.89 | 42.22 | 69.71 | ✗ | 53.46 |
| | Linformer | 16.13 | 65.90 | 53.09 | 42.34 | 75.30 | ✗ | 50.55 |
| | Reformer | 37.27 | 56.10 | 53.40 | 38.07 | 68.50 | ✗ | 50.67 |
| | Sinkhorn Trans. | 33.67 | 61.20 | 53.83 | 41.23 | 67.45 | ✗ | 51.48 |
| | BigBird | 36.05 | 64.02 | 59.29 | 40.83 | 74.87 | ✗ | 55.01 |
| | Linear Trans. | 16.13 | 65.90 | 53.09 | 42.34 | 75.30 | ✗ | 50.55 |
| | Performer | 18.01 | 65.40 | 53.82 | 42.77 | 77.05 | ✗ | 51.41 |
| | FNet | 35.33 | 65.11 | 59.61 | 38.67 | 77.80 | ✗ | 55.30 |
| | Nyströmformer | 37.15 | 65.52 | 79.56 | 41.58 | 70.94 | ✗ | 58.95 |
| | Luna-256 | 37.25 | 64.57 | 79.29 | 47.38 | 77.72 | ✗ | 61.24 |
| | H-Transformer-1D | 49.53 | 78.69 | 63.99 | 46.05 | 68.78 | ✗ | 61.41 |
| | LION-LIT | 16.78 | 65.21 | 54.00 | 43.29 | 72.78 | ✗ | 50.41 |
| | LION-D (w/ *HIPPO*) | 62.0 | 88.78 | 90.12 | 85.66 | 90.25 | 97.28 | 85.63 |
| | LION-S (w/ *HIPPO*) | 62.25 | 88.10 | 90.35 | 86.14 | 91.30 | 97.99 | 86.07 |

## C.5 Experimental details for the MLM/GLUE tasks

**Architectures** We train the BASE (110M parameters) and LARGE (334M parameters) model families from the original BERT paper [7]. For the LION models, we replace the standard self-attention blocks with LION-LIT/LION-D/LION-S blocks while keeping all hyperparameters the

Table 8: GLUE finetuning recipes employed in this work. All recipes finetune on RTE, STSB and MRPC from the weights finetuned in MNLI and the rest from the C4-pretrained weights. All recipes use a sequence length of 128 tokens except BERT24 and Hydra, that use 256. D. AdamW stands for decoupled AdamW.

| Recipe | Param. | Dataset | | | | | | | |
|---|---|---|---|---|---|---|---|---|---|
| | | MNLI | QNLI | QQP | RTE | SST2 | MRPC | COLA | STSB |
| BERT24 [22] | LR | $5 \cdot 10^{-5}$ | $1 \cdot 10^{-5}$ | $3 \cdot 10^{-5}$ | $1 \cdot 10^{-5}$ | $3 \cdot 10^{-5}$ | $8 \cdot 10^{-5}$ | $5 \cdot 10^{-5}$ | $3 \cdot 10^{-5}$ |
| | WD | $5 \cdot 10^{-6}$ | $1 \cdot 10^{-5}$ | $3 \cdot 10^{-6}$ | $1 \cdot 10^{-6}$ | $3 \cdot 10^{-6}$ | $8 \cdot 10^{-5}$ | $5 \cdot 10^{-6}$ | $3 \cdot 10^{-6}$ |
| | Epochs | 3 | 10 | 5 | 3 | 3 | 10 | 10 | 10 |
| | Optimizer | D. AdamW | D. AdamW | D. AdamW | D. AdamW | D. AdamW | D. AdamW | D. AdamW | D. AdamW |
| M2-BASE [10] | LR | $5 \cdot 10^{-5}$ | $5 \cdot 10^{-5}$ | $3 \cdot 10^{-5}$ | $1 \cdot 10^{-5}$ | $3 \cdot 10^{-5}$ | $8 \cdot 10^{-5}$ | $8 \cdot 10^{-5}$ | $8 \cdot 10^{-5}$ |
| | WD | $5 \cdot 10^{-6}$ | $1 \cdot 10^{-5}$ | $3 \cdot 10^{-6}$ | $1 \cdot 10^{-6}$ | $3 \cdot 10^{-6}$ | $8 \cdot 10^{-5}$ | $5 \cdot 10^{-6}$ | $3 \cdot 10^{-6}$ |
| | Epochs | 3 | 10 | 5 | 3 | 3 | 10 | 10 | 10 |
| | Optimizer | D. AdamW | D. AdamW | D. AdamW | D. AdamW | D. AdamW | D. AdamW | D. AdamW | AdamW |
| M2-LARGE [10] | LR | $5 \cdot 10^{-5}$ | $5 \cdot 10^{-5}$ | $3 \cdot 10^{-5}$ | $5 \cdot 10^{-5}$ | $3 \cdot 10^{-5}$ | $8 \cdot 10^{-5}$ | $5 \cdot 10^{-5}$ | $8 \cdot 10^{-5}$ |
| | WD | $5 \cdot 10^{-6}$ | $1 \cdot 10^{-6}$ | $3 \cdot 10^{-6}$ | $1 \cdot 10^{-6}$ | $3 \cdot 10^{-6}$ | $8 \cdot 10^{-6}$ | $1 \cdot 10^{-6}$ | $3 \cdot 10^{-5}$ |
| | Epochs | 3 | 10 | 5 | 2 | 3 | 10 | 10 | 8 |
| | Optimizer | D. AdamW | D. AdamW | D. AdamW | AdamW | D. AdamW | D. AdamW | D. AdamW | D. AdamW |
| Hydra [21] | LR | $10^{-4}$ | $5 \cdot 10^{-5}$ | $5 \cdot 10^{-5}$ | $10^{-5}$ | $5 \cdot 10^{-5}$ | $8 \cdot 10^{-5}$ | $10^{-4}$ | $3 \cdot 10^{-5}$ |
| | WD | $5 \cdot 10^{-6}$ | $10^{-6}$ | $3 \cdot 10^{-6}$ | $10^{-6}$ | $3 \cdot 10^{-6}$ | $8 \cdot 10^{-6}$ | $8 \cdot 10^{-6}$ | $3 \cdot 10^{-6}$ |
| | Epochs | 2 | 7 | 3 | 3 | 2 | 10 | 10 | 8 |
| | Optimizer | D. AdamW | D. AdamW | D. AdamW | AdamW | D. AdamW | D. AdamW | D. AdamW | D. AdamW |
| Modified (Ours) | LR | $10^{-5}$ | $10^{-5}$ | $10^{-5}$ | $10^{-5}$ | $10^{-5}$ | $10^{-5}$ | $10^{-5}$ | $10^{-5}$ |
| | WD | $5 \cdot 10^{-6}$ | $1 \cdot 10^{-6}$ | $3 \cdot 10^{-6}$ | $1 \cdot 10^{-6}$ | $3 \cdot 10^{-6}$ | $8 \cdot 10^{-6}$ | $1 \cdot 10^{-6}$ | $3 \cdot 10^{-5}$ |
| | Epochs | 3 | 10 | 5 | 2 | 3 | 10 | 10 | 8 |
| | Optimizer | D. AdamW | D. AdamW | D. AdamW | AdamW | D. AdamW | D. AdamW | D. AdamW | D. AdamW |

same. For LION-LIT, we incorporate LayerNorm [2] after the attention block to enhance stability. For Hydra, we take the default hyperparameters in [21] and increase the number of layers to 45 to match the number of parameters in the LARGE scale. Our implementation is based on the M2 repository [10], i.e., `https://github.com/HazyResearch/m2`.

**Pretraining** All our pretraining hyperparameters follow Fu et al. [10]: We employ the C4 dataset [8], a maximum sequence length during pretraining of 128 and a masking probability of 0.3 and 0.15 for the training and validation sets respectively. We train our model for 70,000 steps with a batch size of 4096. We employ the decoupled AdamW optimizer with a learning rate of $8 \cdot 10^{-4}$, $\beta_1 = 0.9$, $\beta_2 = 0.98$, $\epsilon = 10^{-6}$ and weight decay $10^{-5}$. As a scheduler, we perform a linear warm-up for 6% of the training steps and a linear decay for the rest of training until reaching 20% of the maximum learning rate.

Our only change in the pretraining hyperparameters is setting the learning rate to $2 \cdot 10^{-4}$ for the LARGE model family. In our preliminary experiments, we found that training diverged when using a learning rate of $8 \cdot 10^{-4}$ for BERT-LARGE.

For completeness, we present the results with the BERT pretraining[6] and BERT 24 finetuning[7] recipes available in the M2 repository.

**Finetuning** For the GLUE finetuning experiments, we employ five different configurations:

- **BERT24**: Available in Izsak et al. [22] and the file this link.
- **M2-BASE**: Available in Fu et al. [10], Section C.1 and the file this link.
- **M2-LARGE**: Available in Fu et al. [10], Section C.1 and the file this link.
- **Hydra**: Available in Hwang et al. [21], Section D.2 and the file this link.
- **Modified**: Same as M2-LARGE but all learning rates are set to $10^{-5}$.

The recipes are summarized in Appendix C.5. The Modified hyperparameter set was devised as M2-LARGE was found to diverge for BERT-LARGE.

---

[6]`https://github.com/HazyResearch/m2/blob/main/bert/yamls/pretrain/hf-transformer-pretrain-bert-base-uncased.yaml`
[7]`https://github.com/HazyResearch/m2/blob/main/bert/yamls/finetune-glue/hf-transformer-finetune-glue-bert-base-uncased.yaml`

Table 9: Combining positional embeddings with LION-D and LION-S. Both pretrained models improve in the validation MLM acc. when employing positional embeddings.

| Model | Pos. Emb. | MLM Acc. | MNLI | RTE | QQP | QNLI | SST2 | STSB | MRPC | COLA | Avg. |
|-------|-----------|----------|------|-----|-----|------|------|------|------|------|------|
| LION-D | ✗ | 66.62 | 82.85 | 52.49 | 89.63 | 88.43 | 91.86 | 85.96 | 83.94 | 53.58 | 78.59 |
|        | ✓ | 66.97 | 83.37 | 54.08 | 89.52 | 88.32 | 92.35 | 83.58 | 79.40 | 54.53 | 78.15 |
| LION-S | ✗ | 67.05 | 83.17 | 53.50 | 89.35 | 88.89 | 93.00 | 37.73 | 77.87 | 53.18 | 72.09 |
|        | ✓ | 67.35 | 83.26 | 52.42 | 89.82 | 88.38 | 92.58 | 83.87 | 79.54 | 55.25 | 78.14 |

## C.6 Ablation studies in the MLM/GLUE tasks

**Combining positional embeddings with LION.** We compare the GLUE performance of LION-D and LION-S when including positional embeddings. We pretrain the BASE models and finetune them with the M2-BASE recipe.

In Table 9 we can observe that adding positional embeddings increased the MLM acc. in around $0.3$ percentage points. In the GLUE benchmark, we observe that for LION-D performance degraded in $0.44$ percentage points, while for LION-S, performance improved in $6.05$ percentage points. We attribute this behavior in GLUE to the dependence on the finetuning recipe.

**Recipe selection.** In this section, we select the best finetuning recipe for each model family and size. For the BASE models, we test the M2-BASE and Modified recipes. For the LARGE models, we test the M2-LARGE and Modified recipes.

In Table 10, firstly, we observe that the M2-BASE recipe generally provides a higher GLUE score than the Modified recipe for the BASE models, e.g., $82.25$ v.s. $80.26$ for the BERT model. Secondly, we observe that for the LARGE model family, the M2-LARGE recipe fails, providing poor performances between $60.96$ and $72.41$ GLUE points. When reducing the learning rate to $10^{-5}$ (Modified recipe), training is more stable and performance reaches between $80.76$ and $82.95$ GLUE points. We find that small changes in the finetuning recipe have a large effect in the performance. Our results in standard recipes show that the LION family of models can obtain a high performance without extensive tuning and closely follow the performance of the BERT family models, at $80.31$ v.s. $82.25$ for the BASE model size and $81.58$ v.s. $82.95$ for the LARGE model size.

Table 10: Recipe selection for the GLUE benchmark.

| Model | MLM Acc. | Recipe | MNLI | RTE | QQP | QNLI | SST2 | STSB | MRPC | COLA | Avg. |
|-------|----------|--------|------|-----|-----|------|------|------|------|------|------|
| BERT | 67.70 | M2-BASE | 84.63 | 64.33 | 89.99 | 89.80 | 92.51 | 86.69 | 89.62 | 60.42 | 82.25 |
|      |       | Mod. | 83.09 | 58.27 | 89.35 | 89.88 | 92.16 | 86.56 | 87.78 | 55.02 | 80.26 |
| LION-LIT | 65.47 | M2-BASE | 82.50 | 63.47 | 89.72 | 89.27 | 91.74 | 87.18 | 89.37 | 49.22 | 80.31 |
|          |       | Mod. | 80.88 | 54.95 | 88.80 | 88.83 | 91.32 | 85.42 | 87.07 | 46.98 | 78.03 |
| LION-D | 66.62 | M2-BASE | 82.85 | 52.49 | 89.63 | 88.43 | 91.86 | 85.96 | 83.94 | 53.58 | 78.59 |
|        |       | Mod. | 80.52 | 52.85 | 88.93 | 88.36 | 91.55 | 82.05 | 84.48 | 49.13 | 77.23 |
| LION-S | 67.05 | M2-BASE | 83.17 | 53.50 | 89.35 | 88.89 | 93.00 | 37.73 | 77.87 | 53.18 | 72.09 |
|        |       | Mod. | 78.14 | 56.39 | 88.68 | 88.52 | 92.39 | 51.22 | 77.60 | 49.75 | 72.84 |
| BERT$_{LARGE}$ | 69.88 | M2-LARGE | 84.97 | 69.10 | 31.59 | 49.15 | 91.93 | 53.61 | 87.87 | 51.16 | 64.92 |
|                |       | Mod. | 85.68 | 67.44 | 89.90 | 91.89 | 93.04 | 88.63 | 90.89 | 56.14 | 82.95 |
| Hydra$_{LARGE}$ | 71.18 | Hydra | 84.24 | 60.44 | 89.24 | 89.73 | 91.70 | 88.21 | 88.99 | 47.72 | 80.03 |
|                 |       | Mod. | 84.39 | 59.42 | 90.38 | 91.31 | 93.43 | 87.19 | 88.57 | 59.46 | 81.77 |
| LION-LIT $_{LARGE}$ | 67.11 | M2-LARGE | 83.20 | 54.51 | 89.08 | 84.90 | 90.44 | 68.57 | 85.25 | 23.35 | 72.41 |
|                     |       | Mod. | 83.73 | 57.18 | 89.85 | 89.93 | 91.86 | 88.02 | 90.18 | 55.36 | 80.76 |
| LION-D $_{LARGE}$ | 68.64 | M2-LARGE | 83.82 | 52.85 | 41.48 | 53.67 | 91.13 | 36.87 | 82.41 | 45.79 | 61.00 |
|                   |       | Mod. | 83.82 | 60.72 | 89.72 | 89.79 | 92.93 | 87.29 | 89.66 | 56.83 | 81.34 |
| LION-S $_{LARGE}$ | 69.16 | M2-LARGE | 83.71 | 50.04 | 38.81 | 53.98 | 91.59 | 36.98 | 82.29 | 50.27 | 60.96 |
|                   |       | Mod. | 84.38 | 57.69 | 89.57 | 90.30 | 92.93 | 87.68 | 90.57 | 59.54 | 81.58 |

## C.7 Additional experimental results for the MLM/GLUE tasks

In this section, we present our bidirectional MLM results in the BASE scale using the BERT pretraining recipe described in Appendix C.5 and BERT24 [22] finetuning recipes (Table 11), we

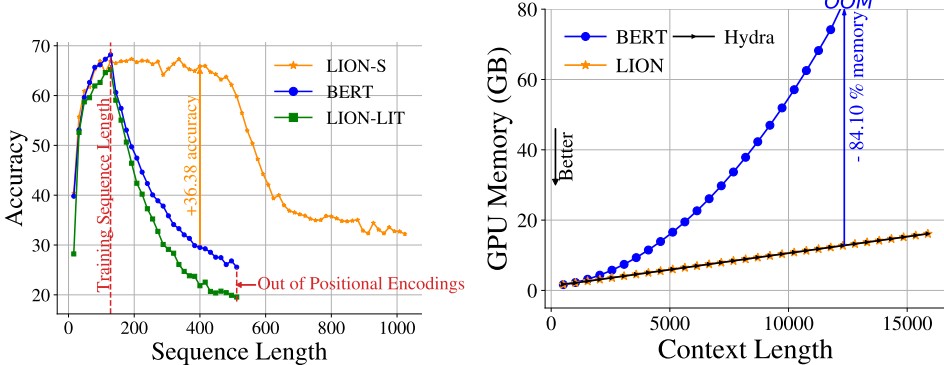

Figure 9: *(Left) MLM Acc. for different sequence lengths.* LION-S is able to generalize to larger sequence lengths and does not require positional encoddings. *(Right) GPU Memory for different sequence lengths.* Results for the LARGE (334M) scale. The memory employed by LION and Hydra scales linearly with the sequence length, being able to process a sequence length of $\sim 16.000$ tokens with less than 20GB. Contrarily, the memory employed by BERT scales quadratically, going out of memory (80GB) at a sequence length of $\sim 12.000$ tokens.

present the per-task GLUE results omitted in the main text (Table 11) and present the length scaling capabilities of the LION-S model (Figure 9).

Table 11: *C4 Masked Language Modeling and GLUE results for the BASE scale (110M) and LARGE scale (334M). For each column (dataset), the best and the second best results are highlighted with* **bold** *and* underline *respectively.*

| Scale | Model | MLM Acc. | MNLI | RTE | QQP | QNLI | SST2 | STSB | MRPC | COLA | Avg. |
|---|---|---|---|---|---|---|---|---|---|---|---|
| BASE | BERT | 67.23 | 84.26 | 59.21 | 89.87 | 90.24 | 92.35 | 88.12 | 90.24 | 56.76 | 81.38 |
| | Hydra* | 69.10 | 84.50 | 57.20 | 91.30 | 90.00 | 93.50 | 91.20 | 88.90 | 77.50 | 84.30 |
| | LION-LIT | 65.08 | 82.37 | 55.81 | 89.49 | 89.57 | 91.74 | 86.27 | 88.25 | 44.46 | 78.50 |
| | LION-D | 66.62 | 82.85 | 52.49 | 89.63 | 88.43 | 91.86 | 85.96 | 83.94 | 53.58 | 78.59 |
| | LION-S | 66.19 | 82.50 | 57.47 | 89.38 | 87.88 | 92.70 | 82.42 | 82.46 | 53.39 | 78.40 |
| LARGE | BERT | 69.88 | **85.68** | **67.44** | 89.90 | **91.89** | 93.04 | **88.63** | **90.89** | 56.14 | **82.95** |
| | Hydra | **71.18** | 84.39 | 59.42 | **90.38** | 91.31 | **93.43** | 87.19 | 88.57 | 59.46 | 81.77 |
| | LION-LIT | 67.11 | 83.73 | 57.18 | 89.85 | 89.93 | 91.86 | 88.02 | 90.18 | 55.36 | 80.76 |
| | LION-D | 68.64 | 83.82 | 60.72 | 89.72 | 89.79 | 92.93 | 87.29 | 89.66 | 56.83 | 81.34 |
| | LION-S | 69.16 | 84.38 | 57.69 | 89.57 | 90.30 | 92.93 | 87.68 | 90.57 | **59.54** | 81.58 |

\* Results extracted from the original paper [21].

### C.8 ImageNet classification results for Tiny scale

In Table 12, we present the image classification results of LION models on the tiny scale models an compare them against the baseline models. Results indicate that the high training speed and competitive performance of LION models is also applicable in the tiny scaled models.

### C.9 Inference time comparison for image classification tasks

In Figure 10, we compare the baseline models on the image classification task. At lower resolutions, all models exhibit similar inference times, with Vim and Hydra being slightly slower than ViT and LION-D chunking. However, at higher resolutions, vision SSMs (Vim and Hydra) become faster. This trend arises because vision SSMs leverage specialized kernels, whereas ViT and LION-D rely on plain Python implementations, leading to increasing overhead as resolution grows.

### C.10 Ablation studies with image classification

**Choice of $\lambda_i$ values.** In this section, we study the properties of the selectivity parameter $a_i$ on CIFAR-100 dataset. We tested, three cases: ($i$) fixed mask with scalars $a_i = a^i$, ($ii$) vector, input-dependent $\mathbf{a}_i \in \mathbb{R}^d$ (*cf.*, Appendix B.5) and iii) input dependent scalar $\mathbf{a}_i \in \mathbb{R}$. The results, presented

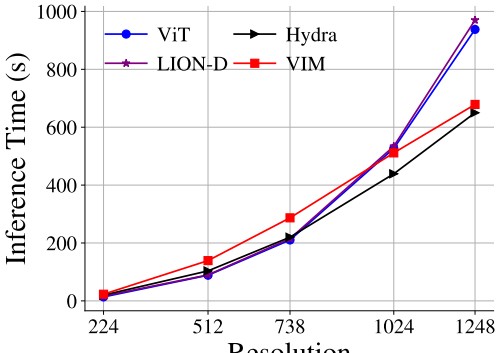

Figure 10: *Inference times comparison.* The inference time of LION-D with chunking, Vim, Hydra and ViT models are presented for different resolutions.

Table 12: *Image classification task in Tiny scale.* Top-1 accuracy on the validation data. * represents changes in patch orders. Best and second best results are in **bold** and underline.

| Model | #Param | Imagenet Top-1 Acc. | Train time |
|---|---|---|---|
| ViT | 5M | 70.2 | ×1 |
| DeiT | 5M | 72.2 | ×1 |
| Vim | 7M | 76.1 | ×9.48 |
| LION-LIT | 5M | 68.9 | ×0.69 |
| LION-D | 5M | 72.4 | ×1.48 |
| LION-D$^\natural$ | 5M | 74.2 | ×1.73 |
| LION-S | 5M | 72.4 | ×2.05 |
| LION-S$^\natural$ | 5M | 73.5 | ×3.83 |

in Table 13, show that while the input dependency is beneficial, the expansion of $\mathbf{a}_i$ is not necessary for image tasks. As a result, we employ option three in all image classification tasks, and the end model is called LION-S.

Table 13: *Ablation studies on image classification.* Additional ablations with CIFAR100 dataset to determine the size and input dependency of the selectivity parameter of the model LION-S.

| Models | Top-1 Acc. |
|---|---|
| Fixed mask $a_i = a^i$ | 75.66 |
| Vector $\mathbf{a}_i \in \mathbb{R}^d$ | 67.55 |
| Scalar, input dependent $\mathbf{a}_i \in \mathbb{R}$ (LION-S) | **77.56** |

**Understanding the power of non-linearity, softmax, and positional embeddings.** In Table 14, we present additional ablations on certain design elements of a Vision Transformer. We perform these experiments on CIFAR-100 data using the same hyperparameters with LION-S. We have observed that either nonlinearity or softmax is essential for the model to converge with a nice accuracy. Though positional embedding boosts the accuracy, a mask can easily replace it.

Table 14: *Ablation studies on image classification.* Additional ablations with the CIFAR-100 dataset to understand the contribution of softmax, nonlinearities in a model is presented. Soft., PosEmb and NonLin expresses if softmax, positional embedding, and non-linearity have been applied. ✗ means the model did not converge. The ▩ symbol denotes the adaptation of recurrent models that achieve equivalence to attention during training while utilizing recurrence during inference, as established by our theorem.

| Models | Top-1 Acc. |
|---|---|
| [1] Soft. + PosEmb + NonLin | 73.88 |
| [2] Soft. + PosEmb (ViT-T) | 77.33 |
| [3] Soft. + NonLin | ✗ |
| [4] Soft. | 73.15 |
| [5] PosEmb + Non.Lin (LION-LIT) | 73.61 |
| [6] PosEmb | 68.54 |
| [7] NonLin | 65.28 |
| [8] Base | ✗ |
| Non.Lin + Mask (LION-S) | **77.56** |

### C.11 Hyperparameters for Training Image Classifiers

All experiments were conducted on a single machine for CIFAR-100 and multiple machines for ImageNet, using NVIDIA A100 SXM4 80GB GPUs. For LION models and Hydra, the ViT architecture serves as the main structure, with Hydra following the Hydra training recipe and other models following the ViT recipe. The training and evaluation codes are adapted from [46] and [50].

### C.12 Calculation of Number of FLOPS

Below we present a theoretical number of FLOPS used in the attention of vision transformers and LION-S during inference where $L$ is the resolution/context length and $D$ is the hidden dimension. Results show that while transformer has $\mathcal{O}(L^2 + LD^2)$ LION-S has $\mathcal{O}(LD^2)$. Note that in this calculation, the exponentials and other nonlinearities are considered as 1 FLOP whereas in reality, the Softmax introduces additional complexities. The same calculations should also apply to other bi-directional models.

The number of FLOPs in the one head of the one layer attention for a vision transformer:

- Calculating $\mathbf{Q}, \mathbf{K}, \mathbf{V}$: $6LD^2$,
- Attention $A = \mathbf{Q}\mathbf{K}^T : 2L^2D$
- Softmax (assuming 1 FLOP for exp): $2L^2$
- Calculating $\mathbf{Y}$: $2L^2D$
- **TOTAL:** $L(6D^2 + 4LD + 2L)$

The number of FLOPs in the attention module for LION:

- Calculating $\mathbf{Q}, \mathbf{K}, \mathbf{V}, \lambda$: $6LD^2 + 2LD$,
- For each token in one forward/backward recurrence:
    - Updating $\mathbf{S}_i^{F/B}$: $3D^2$
    - Updating $\mathbf{z}_i^{F/B}$: $2D$
    - Calculating $c_i^{F/B}$: $4D + 2$
    - Calculating $\mathbf{y}_i^{F/B}$: $2D^2 + 4D + 1$
    - Total: $5D^2 + 10D + 3$
- L forward + backward recurrences: $2L(5D^2 + 10D + 3)$
- Calculating $\mathbf{Y}$: $2L(D + 1)$
- **TOTAL:** $L(16D^2 + 24D + 7)$

### C.13 Distillation Results of LION-S

We have also used the same recipe from DeiT distillation [47] and distilled the RegNet network into LION-S. We observed that the distillation outperforms the original ViT-Tiny on the ImageNet dataset. The results are shown in the table below:

### C.14 Different scans in vision Linear Transformers

When processing images, both the spatial relationships among neighboring pixels and their positions are as critical as the pixel values themselves. Positional embeddings provide a way to incorporate these spatial relationships. The common approach in Transformers involves flattening, also called scanning, the image row-by-row, from left to right and top to bottom as illustrated in the left panel of Figure 11.

Linear Transformers and SSMs such as xLSTM [1], VMamba [29], and Vim [53], often traverse (or scan) the image patches using a the same flattening pattern. To enhance spatial understanding, scans are usually applied in both vertical and horizontal directions [1, 29]. These scans act as separate

 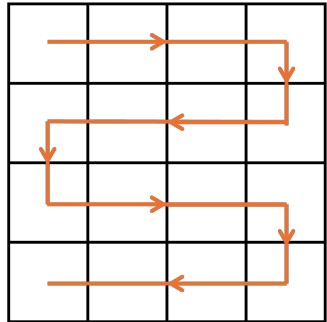 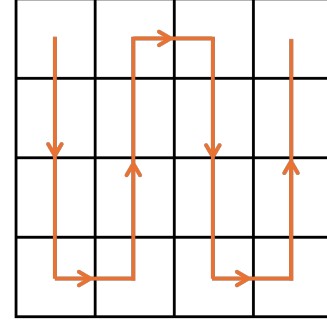

Figure 11: *Reordering of patches.* Left is the naive approach to flatten images, also used in LION-S. Center and right figures are the new approaches applied in LION-S$^\natural$ to consider further spatial information.

recurrences within the model, each capturing different spatial patterns within the image through their respective decay factors.

However, we argue that this method of flattening is suboptimal and can be enhanced to include additional contextual information. To address this, we propose a new scanning scheme for patch values. In the attention module, the patch values are traversed following the patterns depicted in the center and right panels of Figure 11. Forward and backward passes are then executed based on this new ordering, adhering to established procedures. The outputs from these two passes are subsequently averaged to generate the final result.

We indicate this method with $\natural$ throughout the paper. This approach demonstrated a notable improvement in accuracy for image classification tasks while maintaining the efficiency and flexibility inherent to the method. A similar concept has been previously explored in Vision-LSTM [1].

## C.15 Ablation studies on importance of bi-directionality on image classification

To highlight the importance of bi-directionality and demonstrate the versatility of the LION framework, we conducted additional experiments examining the processing directions of the blocks. We evaluated four settings: (i) all blocks process patches in the forward direction only (Forward), (ii) all blocks process patches in the backward direction only (Backward), (iii) odd-numbered blocks process patches in the forward direction while even-numbered blocks process them in the backward direction (Forward-Backward), and (iv) all blocks process patches in both directions (Bi-directional). The results reveal that incorporating both directions improves performance by approximately 4%, while full bi-directionality achieves a significant boost of up to 10%.

Table 15: *Distillation results of* LION-S.

| Models | Top-1 Acc. |
|---|---|
| LION-S | 67.95 |
| VIT-Tiny | 70.23 |
| LION-S (Distilled) | **70.44** |

Table 16: Results for LION-S and LION-S$^\natural$ with different directional settings on CIFAR-100. Incorporating both directions improves performance by approximately 4%, while full bi-directionality achieves a significant boost of up to 10%.

| Model | Top-1 Acc. |
|---|---|
| LION-S (Forward) | 71.08 |
| LION-S (Backward) | 69.61 |
| LION-S (Forward-backward) | 73.93 |
| LION-S **(Bi-directional)** | **77.56** |
| LION-S$^\natural$ (Forward) | 70.24 |
| LION-S$^\natural$ (Backward) | 70.42 |
| LION-S$^\natural$ **(Bi-directional)** | **80.07** |

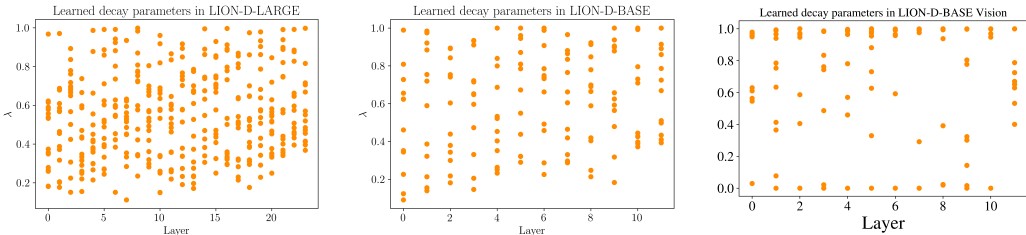

| (a) Chunkwise Parallel LION-LIT | (b) Chunkwise Parallel LION-D | (c) Chunkwise Parallel LION-S |

Figure 12: LION-D *Decay factor distribution.*

## C.16   LION-D **Decay factor distribution**

We visualize the distribution of decay factors in LION-D across layers for both image classification and MLM tasks. Interestingly, the model learns a diverse range of decay values, highlighting their importance across different heads and layers.

## C.17   **Generation of the Mask**

Below we present the Python code used for the creation of the bidirectional mask $\mathbf{M}$ as described in previous sections.

```python
#caption=Code for generation of the selective bidirectional mask
                                of \lion ,
def create_matrix_from_tensor(tensor):
    cumsum = torch.exp(torch.cumsum(tensor, dim=-1))
    A = torch.matmul(cump.unsqueeze(-1) ,
    1/ ( cump.unsqueeze(-1).transpose(-1,-2)))
    return torch.tril(A)

def Mask_selective(vec):
    vec_shape = vec.shape
    A_for = create_matrix_from_tensor(vec)
    A_back = create_matrix_from_tensor(flip(vec))
    return A_for + A_back - torch.eye(A_for.shape[-1])

def Mask_Decay(a_i , L):
    idx = torch.arange(L,device=a_i.device)
    I, J = torch.meshgrid(idx, idx, indexing='ij')
    E = (torch.abs((I-J)).float().view(1,1,L,L))
    M = torch.sigmoid(a_i).view(1,-1,1,1)**E
    return M
```

```python
#caption=Code for generation of the partial selective
                                bidirectional mask of \lion for
                                chunking,
def Partial_Mask_selective(vec):
    B,H,L = vec.shape
    A_for = create_matrix_from_tensor_forward(vec[...,:-1]),
                                chunk_index,chunk_len)
    A_back = create_matrix_from_tensor_backward(vec[...,1:]),
                                chunk_index,chunk_len)
    I  = torch.diag_embed(torch.ones((B,H,L-chunk_index*chunk_len)
                                ),offset = -chunk_index*chunk_len
                                )[...,:L]
    return A_for + A_back - I.to(A_for.device)
```

## D  Future Direction & Limitation

While LION maps many Linear Transformers to the bidirectional setting, we empirically focused on three examples. LION does not rely on additional design choices (e.g., gating in Mamba), but could incorporate them to further boost performance and scale to larger models, potentially suppressing softmax Transformers in multiple domains and tasks.

