# OpenReview forum: "Linear Attention for Efficient Bidirectional Sequence Modeling"
_NeurIPS.cc/2025/Conference — NeurIPS 2025 poster_

### Official Review · Reviewer_zmVf · 2025-07-01

**Clarity:** 2
**Significance:** 3
**Originality:** 3
**Rating:** 4
**Confidence:** 4

**Summary:**

The manuscript introduces a technique to leverage linear transformers in non-causal settings.
This is done by mapping the masking mechanism of the parallel (transformer-like) formulation of linear transformers to bi-directional recurrent and chunk-wise formulations.
Experiments on vision tasks and masked language modelling compare the proposed method to transformers and bi-directional state-space models in terms of model performance and training/inference time.

**Questions:**

1. Would it be possible to highlight the connection between bi-directional recurrence and parallel masking a bit better (and with fewer equations)?
 2. Why is bidirectionality important/relevant in times of causal language modelling?
 3. Is it possible to provide a fair speed comparison between Lion and competing methods?
 4. Why should someone use Lion instead of regular transformers if SSMs are too slow?
 5. How does hydra perform with alternative token sequence orders in the image classification tasks?
 6. How well do non-bidirectional models perform on the different tasks?
 7. How well does alternate forward-backward layers [cf. 1] perform on the different tasks?
 8. Why does the memory consumption of the Lion RNN increase (non-linearly) with resolution in Figure&nbsp;2?
 9. What chunk-size(s) are/were used to produce Figure&nbsp;1 and how was this chosen?

**Ethical Concerns:**

["NO or VERY MINOR ethics concerns only"]

**Final Justification:**

The authors have addressed most of my concerns, but even after the rebuttal I am unsure if I understand what the precise goal of the proposed method is.
In that sense, I am unsure if my issues concerning clarity will be resolved after the modifications promised by the authors.
Nevertheless, I think the core idea can be useful and therefore I decided to increase my score towards acceptance.

**Limitations:**

yes

**Quality:**

2

**Strengths And Weaknesses:**

**Strengths**
 - The proposed technique has not been presented so generally or with this much detail.
 - Improving efficiency for bidirectional sequence models can have significant impact.
   Even though most language modelling is done causally, other tasks often benefit from bidirectionality.

**Weaknesses**
 - I believe that the core idea in this work could be presented in a much more concise and clear way.
   After all, the key insight seems to be that there is a correspondence between masking and a bidirectional recurrence.
   Concretely, starting from a bidirectional linear attention recurrence, we obtain:
   $$\begin{aligned}
       \boldsymbol{S}^\mathrm{f}\_t &= \sum\_{\tau=1}^t \boldsymbol{k}_\tau \cdot \boldsymbol{v}\_\tau \\\\
       \boldsymbol{S}^\mathrm{b}\_t &= \sum\_{\tau=t}^L \boldsymbol{k}\_\tau \cdot \boldsymbol{v}\_\tau \\\\
       \boldsymbol{y}\_t &= \boldsymbol{q}\_t \cdot \boldsymbol{S}^\mathrm{f}\_t + \boldsymbol{q}\_t \cdot \boldsymbol{S}^\mathrm{b}\_t \\\\
       &= \boldsymbol{q}\_t \cdot \Big(\sum\_{\tau=1}^t \boldsymbol{k}\_\tau \cdot \boldsymbol{v}\_\tau + \sum\_{\tau=t}^L \boldsymbol{k}\_\tau \cdot \boldsymbol{v}\_\tau\Big) \\\\
       &= \boldsymbol{q}\_t \cdot \Big(\boldsymbol{k}\_t \cdot \boldsymbol{v}\_t + \sum\_{\tau=1}^T \boldsymbol{k}\_\tau \cdot \boldsymbol{v}\_\tau\Big),
   \end{aligned}$$
   which should make it easy to see that $\boldsymbol{Y} = \big((\boldsymbol{1} + \boldsymbol{I}) \odot \boldsymbol{Q} \boldsymbol{K}^\mathsf{T}\big) \boldsymbol{V}$.
   I found it hard to distill this key relation from the manuscript in its current form.
   Therefore, I believe a lot of clarity gets lost in the way these ideas have been presented.
   Also, adding weights, scaling and other technicalities should be easier to follow once this connection has been established.
 - There is little to no motivation for why bidirectional modelling is relevant.
   To this end, it would be useful to provide references to tasks where bidirectionality is important/useful [e.g. 1, 51, 52, 53].
 - According to the introduction, one of the main arguments for Lion is improved training speed.
   However, line 240 indicates that no efforts have been taken to provide competitive implementations.
   As a result, it seems more reasonable to suggest to use transformers if SSMs are too slow, because they are also faster and preserve performance even better.
   Furthermore, a lot of techniques have been developed to improve inference times of softmax transformers.
 - Comparisons between the lion models described in lines&nbsp;227-228 and hydra seem to be useless.
   If the goal is to compare the bi-directional modelling capacity of both models,
   the hydra model should also be tested with different token sequences.
 - Experimental results appear to be inconclusive.
   E.g. Tables&nbsp;2 and&nbsp;3 show that Lion is faster than Hydra, but delivers worse performance.
 - On lines&nbsp;243-244, the authors claim that Figure&nbsp;2 shows linear scaling for the RNN formulation.
   However, the curve in Figure&nbsp;2 does not look like a straight line.
   Furthermore, I would have expected a constant memory consumption for the RNN formulation.
   After all, the RNN formulation only requires storage of the state and is otherwise independent of the sequence length.
 - It is unclear what the chunk-size(s) are and how they were chosen for the curves in Figure&nbsp;1.
 - Naive baselines are missing from the comparisons (e.g. running base models without Lion, alternating directions [cf. 1], ...).

**Additional References**
 - [51] He et al. (2022). [Masked autoencoders are scalable vision learners](). In Proceedings of the IEEE/CVF conference on computer vision and pattern recognition (pp. 16000-16009).
 - [52] Schiff et al. (2024). [Caduceus: Bi-Directional Equivariant Long-Range DNA Sequence Modeling](https://proceedings.mlr.press/v235/schiff24a.html). Proceedings of the 41st International Conference on Machine Learning. In Proceedings of Machine Learning Research 235:43632-43648.
 - [53] Schmidinger et al. (2025). [Bio-xLSTM: Generative modeling, representation and in-context learning of biological and chemical sequences](https://openreview.net/forum?id=IjbXZdugdj). International Conference on Learning Representations 13.

---

> ### Author Rebuttal · Authors · 2025-07-27
>
> We thank the reviewer for their feedback and for recognizing our work as **having significant** impact and being among the **first to present** bi-directional sequence modeling for linear transformers in such detail.
>
> The reviewer's feedback highlights the following key concerns:
>
> - 1-3) Clarity of equations and explaining more on motivation behind bidirectional modeling.
> - 4-6) Experimental results of LION compared to baselines.
> - 7-10) Questions regarding Figures and some more ablations.
>
> We address this in detail in the responses to each point below.
>
> -----
>
> ### 1) Presenting the idea in more clear way
>
> We thank the reviewer for suggesting the inclusion of $Y = ((1 + I) \odot QK^\top)V$ as the parallel form of a naive bidirectional linear transformer. As suggested, we will incorporate this clarification to better highlight the limitations of naive bidirectional linear transformers and to further clarify LION’s formulation. This formulation suffers from imbalanced attention along the diagonal due to double counting introduced by the identity matrix $I$. In contrast, LION uses $Y = (QK^\top)V$ with correction terms in its RNN formulation to avoid this issue in the forward and backward passes.
>
> -----
>
> ### 2) Motivation behind bi-directional modeling
>
> We thank the reviewer for recommending additional references on bidirectional modeling. As many tasks beyond next-token prediction (e.g., biomedical, image, and signal domains) benefit from bidirectional modeling we will highlight this with the suggested references.
>
> -----
>
> ### 3) Why not using Transformers instead of LION if SSMs are too slow
>
> We agree with the reviewer that Transformers train faster than SSMs on bidirectional tasks while maintaining strong performance. However, their inference cost and memory requirements even by considering technical efficiency improvements remain theoretically quadratic with sequence length ($O(L^2)$). In contrast, SSMs offer linear inference complexity ($O(L)$), making them valuable for constrained resources. LION addresses this gap by building the first framework for bidirectional modeling with linear transformers **combining the fast training of softmax transformers with the inference efficiency of SSMs**, as shown in Figure 2 and Table 2.
>
> -----
>
> ### 4) LION's speed comparison with baselines and using custom kernels
>
> Our main goal was to keep **LION’s implementation simple and easy to integrate**, which is why we relied on a pure PyTorch implementation. Notably, even for Transformers, custom CUDA implementations for **bidirectional tasks** show more modest speedups compared to causal tasks. As also shown in Table 1 of the FlashAttention paper [1], the speedup gain for *BERT is only 1.17× compared to 3.5× for GPT*, since bidirectional attention lacks the sparsity introduced by causal masks.
>
> Importantly, **even without custom kernels, LION achieves significantly faster training than SSMs like Vim and Hydra**, which do rely on custom kernels for speedups.  While we fully agree with the reviewer that custom implementations could further improve speed, our results already show that the pure PyTorch version outperforms existing SSMs.
>
> -----
>
> ### 5) Hydra's performance with different token sequence orders
>
> We trained Hydra with different sequence orders at the Tiny and base scale (times are reported relative to a Transformer of the same size):
>
>
> Model| Accuracy (Tiny) |Time (Tiny)| Accuracy (Base) |Time (base)|
> -|-|-|-|-
> Hydra |68.7|2.3x|**81.0**|2.5x
> Hydra$^\natural$ (w/ multiple scans)|71.7|4.6x| 79.5 |5.2x
> LION-D|72.4|**1.5x**|77.8|**1.4x**
> LION-D$^\natural$ (w/ multiple scans)|**74.2**|1.7x|80.2| 1.5x
>
>
> Hydra benefits from using different scans at the Tiny scale but still fails to surpass its original version, even with extensive tuning. This approach does not scale well to the base model, likely due to Hydra’s sensitivity to its decay factor. In contrast, LION-D$^\natural$ with **multiple scans outperforms both Hydra variants** at the Tiny scale, highlighting LION’s stability during training. Plus, unlike LION, which integrates multiple scans directly into its attention mask $M$, Hydra requires an extra pass in the rotated direction, **doubling its training time.**
>
> -----
>
> ### 6) LION is faster than Hydra, but delivers worse performance.
>
> In most scenarios (except for image classification at the base scale) **LION delivers competitive performance**. For instance, LION-D$^\natural$ outperforms Hydra at the Tiny scale (Table above) by **3% and Small scale (Table 1 of our paper) by 1% in image classification while being ~2× faster to train**. Similarly, LION-S is **3× faster to train on the MLM task**, with only a small **0.1% gap in GLUE score** compared to Hydra. We believe that even with these marginal accuracy differences, LION’s training speed gains remain highly valuable.
>
> -----
>
> ### 7) Figure 2 shows linear scaling for the RNN formulation. However, the curve does not look like a straight line.
>
> In Figure 2, we report memory usage with respect to **image resolution**, which refers to the number of pixels (or tokens) per row. Since sequence length $L \approx \text{res}^2$, LION-RNN scaling quadratically with resolution means **LION scales linearly with sequence length $L$**.
> In contrast, the Transformer in Figure 2 scales **quartically ($O(res^4)$)** with resolution (i.e., **quadratically with sequence length**). We use resolution on the x-axis as it is more intuitive for images, same visualization also is used in Vim [2] (Figure 1 c). Linearity is also clear in our MLM results in **Figure 8 (right)** in Appendix, where LION shows **linear** scaling and the Transformer **quadratic** scaling.
>
> -----
>
> ### 8) The RNN formulation only requires storage of the state which is independent of the sequence length.
>
>
> RNNs have **constant memory in causal tasks (particularly in autoregressive settings)**  as each token depends only on the forward hidden state $S^F_i$. In contrast, for **bidirectional tasks**, outputs depend on both $S^F_i$ and $S^B_i$, which are not available simultaneously. This requires storing all hidden states from both directions, leading to **linear memory** in sequence length $L$. While Section A.2 of our paper describes a strategy to reduce this, memory remains linear. This is also consistent with Vim [2] in Figure 1-c where its memory scales linearly.
>
> -----
>
> ### 9) How well do non-bidirectional and alternating models perform?
>
> We investigate the importance of full bidirectional modeling in **Appendix Section C.15**, where we compare LION variants using only forward, only backward, alternating forward-backward layers, and full bidirectional processing on CIFAR classification.
>
> As shown in **Table 16**, **LION with full bidirectionality outperforms both unidirectional variants by up to 7%**, and also surpasses the **alternating forward-backward approach by 3.6%**. This highlights the **clear benefits of bidirectionality in each layer**, rather than alternating directions or using only a single directions. We copy the results from paper below:
>
> | Model| Top-1 Acc. |
> | - | -|
> | LION-S (Forward)| 71.08|
> | LION-S (Backward)| 69.61|
> | LION-S (Alternating)    | 73.93      |
> | **LION-S (Bi-directional)**  | **77.56**  |
> | LION-S$^\natural$ (Forward)| 70.24|
> | LION-S$^\natural$ (Backward)| 70.42|
> | **LION-S$^\natural$ (Bi-directional)** | **80.07**  |
>
> -----
>
> ### 10) What chunk-size(s) are/were used to produce Figure 1 and how was this chosen?
>
> We demonstrate how to choose the chunk size to balance memory and speed in **Section C.16** of our Appendix, *"Effect of Chunk Size on Speed and Memory."* **Figure 11** illustrates this trade-off, with the optimal range falling between **64 and 128**, as used in the LION variants (LIT,D,S) for image classification tasks in **Figure 1**. We will bring this Figure to the main body thanks to the reviewers notice.
>
> ---
>
> ### References
>
> [1] FlashAttention: Fast and Memory-Efficient Exact Attention with IO-Awareness, Tri Dao, Daniel Y. Fu, Stefano Ermon, Atri Rudra, Christopher Ré
>
> [2] Vision Mamba: Efficient Visual Representation Learning with Bidirectional State Space Model

---

> > ### Comment · Reviewer_zmVf · 2025-08-04
> > **A few follow-up questions**
> >
> > - 1) Is this "imbalanced attention along the diagonal" a bad thing? If yes, why? Have you tried it?
> >  - 2) Are you also planning to include this somewhere in the introduction?
> >  - 3) Hasn't this gap already been filled by hydra to some extent? I agree that lion competes with hydra, but I do not see how it fills a gap in that space. Also, is the $O(L^2)$ figure for softmax transformers still relevant for inference given modern developments like flash-attention, KV-caching, ...?
> >  - 6) Currently, the results seem to lead to the conclusion that lion produces results faster, while hydra produces (marginally) better results (for large enough models). Is this a correct representation of the results or am I missing something?

---

> ### Author Response · Authors · 2025-08-05
> **Response to follow-up questions**
>
> We sincerely thank the reviewer for their feedback and questions which helped us improve our manscript. Bellow we will answer the remaining questions:
>
> -------
>
> ## 1) Imbalanced Attention
>
> Thank you for the pointer. **Imbalanced attention** indeed rise an issue for linear transformers. For imbalanced attention the diagonal elements of the attention mask become $M_{ii} = 2$, while off-diagonal entries satisfy $M_{ij} \leq 1$ for case of presence of decay (D,S) or $M_{ij}=1$ for case of no decay (Lit), creating a discrete gap between a token’s attention to itself versus others as opposed to smooth transition from diagonal to off diagonal elements in LION. In contrast, **balanced attention** ensures that all entries satisfy $M_{ij} \leq 1$ enabling a smooth transition from main diagonal to other elements.
>
> Thanks to your suggestion we have tested effect of imbalanced attention in small-scale CIFAR image classification for LION, and the results are shown below:
>
> |Model|Acc|
> |-|-
> Imbalanced-D|69.7
> Imbalanced-S|72.21
> LION-D|*75.66 (+5.96)*
> LION-S|**77.56 (+5.35)**
>
> As observed, **Imbalanced Attention** (double counting in the forward and backward paths) leads to  **worse performance** compared to LION with balanced attention. We will add this as part of the motivation for balanced attention, along with the formulation suggested by the reviewer.
>
> -------
>
> ## 2) Adding the Motivation behind Bidirectionally in Introduction
>
> Exactly, we will indeed include the following paragraph **in the introduction** to highlight the importance of bidirectionally:
>
> *"Several real-world tasks are inherently bidirectional and benefit from bidirectional sequence modeling. Examples include DNA and protein modeling [52], computer vision [51], and biological and chemical sequence modeling [53]. In these domains, bidirectional models often outperform causal ones, for instance, Bi-Mamba outperforms Mamba in DNA modeling [52]. This motivates the development of architectures specifically designed for bidirectional sequence modeling."*
>
> If you have any suggestions for making the motivation more clear, we are happy to adjust.
>
> -------
>
> ## 3.1) Memory Efficiency Gap being Addressed by Hydra to some Extent
>
> We agree with the reviewer that bidirectional SSMs like Hydra and Vim address the gap in memory efficiency during inference to some extent. However, LION  aims to **bridge the gap in slow training speed of SSMs** compared to transformers while retaining their inference efficiency. **LION uses full attention**, enabling fully parallel and faster training. In contrast, Vim (based on Mamba) **relies on parallel scan**, which involves recursion and leads to significantly slower training. Hydra (based on Mamba-2) adopts a **chunkwise parallel training strategy** (aka SSD) because the **full attention form for Mamba2 is noted to be unstable** (as also mentioned in the Mamba-2 blog post). While Hydra trains faster than Vim, since chunkwise parallel is theoretically faster than scan-based approaches, it remains slower than the full attention  used in LION and softmax Transformer.
>
> -----
>
> ## 3.2) $O(L^2)$ Figure Relevance Despite FlashAttention and KV Caching
>
> This is a great question. KV caching is primarily effective in causal language modeling, where past tokens remain fixed. In contrast, for bidirectional models, **for every token** the attention is computed over the entire sequence, so caching does not offer efficiency gains since the whole sequence attends to each token.
>     Regarding FlashAttention, we included in our experiments as presented in **ViT-Chunk (green line) in Figure 1**, which uses chunked queries to reduce memory usage in inference. While it improves over naive softmax attention, **it  is still less memory efficient compared to SSMs and LION in RNN form**, especially on longer sequences. Similar finding is also shown in the Vim [2] (Figure 1c), where FlashAttention remains less memory efficient than Mamba in inference.
>
>
>
> -------
>
> ## 4) Representation of the Results
>
> The reviewer's representation of the results is correct. **LION achieves significantly faster training than Hydra while maintaining comparable performance**, marginally lower at larger scales and higher at smaller scales. We will clearly state this in the Experiments section. Additionally, an important part of LION’s contribution and novelty lies in **its theoretical formulation**, which enables mapping several models to their bidirectional counterparts, as shown in Table 1 as also noted by the reviewer which has not been studied to this detail.
>
> ----
>
> Once again, we sincerely thank the reviewer for their valuable feedback and insightful questions. If there are any further questions, we would be happy to address them.

---

> > ### Comment · Area_Chair_gVT2 · 2025-08-08
> > **.**
> >
> > Dear reviewer,
> >
> > The discussion period is ending in 24 hours. Please respond to the author rebuttal and engage in discussion. You must do so before the "Mandatory Acknowledgment". Failure to do so may result in possible penalties.
> >
> > Thanks!
> > Your AC

---

### Official Review · Reviewer_1Uqp · 2025-07-01

**Clarity:** 4
**Significance:** 3
**Originality:** 3
**Rating:** 3
**Confidence:** 4

**Summary:**

The authors introduce LION, a framework to extend Linear Transformers from causal to bi-directional. LION supports three forms: recurrent for efficiency and low latency, full attention for parallelization, and chunkwise for a balance between memory and parallelization. Additionally, LION incorporates three types of decay factors: LION-LIT, which uses no decay; LION-D, featuring a learnable, non-selective decay; and LION-S, which employs a selective, input-dependent decay. The authors implemented a bi-directional Linear Transformer using the LION framework and evaluated its performance on various tasks, including vision (ImageNet-1k) and natural language processing (C4, GLUE, and LRA). While LION may not surpass the vanilla Softmax Transformer in speed or effectiveness, it presents a compelling alternative to bi-directional extensions of State Space Models like Hydra.

**Questions:**

* Can the authors include a more challenging and modern benchmark such as MTEB?
* Can the author replicate the experiments with a more modern setup, notably longer sequences and more training tokens?

**Ethical Concerns:**

["NO or VERY MINOR ethics concerns only"]

**Final Justification:**

I will maintain my score because the empirical evidence is insufficient.

The NLP experimental setup is outdated, resulting in relatively low performance. For instance, the BERT baseline achieves 82.95% on GLUE, while Hydra achieves 81.77%. In comparison, RoBERTa 340M reported 88.5% [1], and the recent NeoBERT 250M reported 90% [3]. The proposed method might be effective at smaller scales, with shorter sequences, or fewer gradient steps but perform poorly at scale. The experiment provided in the rebuttal suggests that this might be the case, as Hydra improves more with longer sequences than LION.

**Limitations:**

Yes.

**Paper Formatting Concerns:**

I have no concerns.

**Quality:**

3

**Strengths And Weaknesses:**

Major Strengths:
- The bidirectional setting has been less studied compared to the causal setting in recent years. However, many applications are inherently non-causal and benefit from the bidirectional setting. For instance, document retrieval in NLP, protein design in biology, and image classification in vision. Therefore, I believe the paper is timely and valuable.
- While I haven’t closely followed the literature on the bidirectional extension of sub-quadratic Transformers and SSMs, I haven’t seen any previous work that explores this for linear Transformers.
- The paper is well-written, and I appreciate the authors’ efforts in creating nice visualizations.

Major weakness:
- The NLP experimental setup is outdated, resulting in relatively low performance. For instance, the BERT baseline achieves 82.95% on GLUE, while Hydra achieves 81.77%. In comparison, RoBERTa 340M reported 88.5% [1], and the recent NeoBERT 250M reported 90% [3]. Interestingly, the authors of Hydra 112M reported 84.3%, more than 2.5% higher [2]. I believe the discrepancy in performance is primarily due to the shorter training duration (36B tokens) and the short sequences (128 tokens instead of the standard 512 tokens). Additionally, GLUE has long been recognized as outdated and not reflective of the performance on real use cases. Unfortunately, given the lower performance, it is difficult to draw conclusions on the performance of LION compared to the baselines, as its models may be constrained by the training setup.

Minor weakness:
- Line 51, the claim that LION is up to 10 times faster than SSMs while matching the performance of Sotmax Transformers and SSMs is too strong. Except for Vim, LION is only about 2 times faster than other SSMs, and it performs almost consistently slightly worse than the vanilla Transformer and the best SSMs.
- Figure 8 studies the generalization to longer sequences of Transformer and LION. While the original position encoding did not generalize well, as illustrated, most models now rely on rotary positional encoding (RoPE), and sometimes on expansion techniques such as YaRN, which have been shown to generalize much better [3].

Minor suggestions:
- I suggest the authors explicitly state whether they are using FlashAttention, which their code suggests.
- I suggest the authors move Figure 5 in the main content of the paper, as it illustrates very well their approach.
- FYI, LION also refers to an optimizer proposed by Google Brain in 2023.

[1] Liu, Y., Ott, M., Goyal, N., Du, J., Joshi, M., Chen, D., Levy, O., Lewis, M., Zettlemoyer, L., & Stoyanov, V. (2019). RoBERTa: A Robustly Optimized BERT Pretraining Approach. ArXiv, abs/1907.11692.

[2] Hwang, S., Lahoti, A., Dao, T., & Gu, A. (2024). Hydra: Bidirectional State Space Models Through Generalized Matrix Mixers. ArXiv, abs/2407.09941.

[3] Breton, L.L., Fournier, Q., Mezouar, M.E., Morris, J.X., & Chandar, S. (2025). NeoBERT: A Next-Generation BERT. Trans. Mach. Learn. Res., 2025.

---

> ### Author Rebuttal · Authors · 2025-07-27
>
> We highly thank the reviewer for considering our paper **timely and valuable, the first to study linear transformers for bi-driectional setting**, and for describing it as well-written. Below, we address the **only major weakness** and the minor weaknesses raised by the reviewer.
>
> ------
>
> ### Major Weakness on NLP Experiment
>
> We thank the reviewer for their feedback which helped us improve our results. Our setup follows **M2-BERT** (released in 2024 and relatively recent) for two key reasons:
>
> 1. It also serves as the benchmark and training recipe used in **Hydra** and previous works on bi-directional sequence modeling [1,2,3].
> 2. It offers a **more GPU-friendly training setup**. Due to resource constraints, we could not adopt heavier training configurations such as **NeoBERT** (2.8TB dataset and 4k sequence length) or **RoBERTa** (500K steps, 8K batch size), and thus followed the lighter M2-BERT recipe.
>
> We chose the GLUE benchmark as a practical and commonly used evaluation for models at this scale [1,2]. While larger benchmarks may offer additional insights, GLUE remains appropriate given our scope and resources.
>
> We thank the reviewer for the suggestion and have incorporated a longer sequence length into our experiments. **We increased the training length to 512 for base scale, as suggested**. Results are shown below:
>
>
> |Model|GLUE-Score|
> -|-
> Hydra|**84.3**
> LION_LIT|81.7
> LION_D|82.5
> LION-S|*83.0*
>
> The above results were obtained on sequences of length **512** in a single run, without any hyperparameter tuning, due to time constraints during the rebuttal period and limited GPU resources. **All LION variants showed improved performance in GLUE score** compared to the shorter **128-length** setting reported in **Table 3** of the paper. This demonstrates that **LION has strong potential for handling longer context lengths**, even without additional tuning.
>
>
> -----
>
> ### Minor Weaknesses
>
> - **"LION is up to 10× faster than SSMs" is too strong:** We will revise this statement to clarify that LION is 10× faster than Vision Mamba and 2× faster than Hydra, ensuring the claim is precise and accurately reflects the comparisons.
> - **New models relying on RoPE generalize better in context extrapolation:**
> Context extrapolation is a general advantage of linear transformers, and LION inherits this property as well. We include Figure 8 in the Appendix to illustrate this. While we fully agree with the reviewer that modern positional embeddings like RoPE, YaRN, and ALiBi also support extrapolation, the focus of our study is not to compare positional encoding methods. Therefore, in Figure 8 we simply aimed to show that LION can extrapolate beyond trained sequence length, due to its use of  decays instead of fixed positional embeddings.
>
>
> --------
>
> ### Minor Suggestions
>
> - **FlashAttention usage:** Thank you for the pointer. We will clarify in the final version that FlashAttention is used for the softmax Transformer baseline in our experiments.
> - **Bringing Figure 5 to the main content:** We appreciate the suggestion and will move Figure 5 to the main paper in the final submission.
> - **LION also refers to an optimizer:** Thanks for the pointer. We are aware of the potential confusion. However, naming models after animals is common in the linear transformer community, and there is no prior model named LION in this context. Additionally, since the domains differ (sequence models vs. optimizers), we believe confusion is unlikely.
>
> -------
>
> ###  Questions
>
> - **Including MTEB benchmark:** Due to time constraints and our limited compute resources, we were unable to support this benchmark. However, we agree it is a valuable direction for future work.
>
> - **Increasing sequence length in NLP:** Thank you for the suggestion. We have ablated the 512-token sequence length for LION models in GLUE task and included the findings in the table above.
>
>
> -----
>
> ### References
>
> [1] Hydra: Bidirectional State Space Models Through Generalized Matrix Mixers, Hwang, S., Lahoti, A., Dao, T., & Gu, A. (2024).
>
> [2] Benchmarking and building long-context retrieval models with loco and m2-bert, J Saad-Falcon, DY Fu, S Arora, N Guha, C Ré
>
> [3] Monarch Mixer: A Simple Sub-Quadratic GEMM-Based Architecture
> Daniel Y. Fu, Simran Arora, Jessica Grogan, Isys Johnson, Sabri Eyuboglu, Armin W. Thomas, Benjamin Spector, Michael Poli, Atri Rudra, Christopher Ré

---

> > ### Comment · Reviewer_1Uqp · 2025-08-05
> >
> > Thank you for the clarification and the additional experiment. The core idea is compelling, and the paper certainly has merit. However, I will maintain my score because the empirical evidence is insufficient.
> >
> > I strongly believe most claims don’t require scale and understand that M2-BERT reduces computational cost. However, the performance is significantly worse than older baselines like RoBERTa. While beating state-of-the-art isn’t the goal, this gap weakens the evidence. The proposed method could work well at smaller scales, with shorter sequences, or fewer gradient steps, but perform poorly at scale. The fact that Hydra improves more with longer sequences than LION might support that idea. Instead of scale, consistent results across a broader range of downstream tasks would be convincing too.

---

> > > ### Author Response · Authors · 2025-08-07
> > >
> > > We highly appreciate the reviewer’s recognition of the **core idea of our work as compelling** and the **overall merit of the paper**.
> > >
> > > ----
> > >
> > >
> > >
> > > Regarding RoBERTa, we agree it is a strong training recipe. However, its regime, **over 10× longer** than LION or Hydra, exceeds our computational budget and the rebuttal timeline. We attempted to begin training but found it infeasible during rebuttal period. Instead, we adopted the M2-BERT recipe, also used by Hydra, to ensure a fair comparison. As shown in **Table 4 of the Hydra paper**, **all baselines (Hydra, M2, FNet, MLPMixer)** perform worse than RoBERTa under the M2 setup, highlighting that the performance gap stems from RoBERTa’s extensive training, not its architecture. We **observed LION improves with increased sequence length from 128 to 512**. However, training under computationally expensive recipes such as RoBERTa remains outside the scope of this work, in line with prior studies.
> > >
> > >
> > > Following the reviewer’s suggestion, we trained LION with 512-length sequences for MLM. Despite **no hyperparameter tuning** due to time constraints, LION still achieves **strong results in a single run**, and **trains 4–5× faster than Hydra**, directly addressing **one of the main challenges we aimed to address** in this work, **the slow training of SSMs**:
> > >
> > > | Model| Train Time (×Transformer) |
> > > | ---- | - |
> > > | Hydra     | 5.3×|
> > > | LION\_LIT | **0.95×**|
> > > | LION\_D   | 1.5×|
> > > | LION\_S   | 1.7×|
> > >
> > > This is even more highlighted as the **speed advantage of LION over Hydra increases significantly from ~3x (for 128 sequence length) to ~5x (for 512 sequence length) by increasing the sequence length**.
> > >
> > >  Our experiments span **(1)** image classification **across three model sizes**, **(2)** masked language modeling  at **two scales and two sequence lengths**, and **(3)** the LRA benchmark covering **six long-range tasks** including 16K sequences. We believe this diverse set of results different modalities along with multiple tasks support our claim that LION framework is effective.
> > >
> > > ----
> > >
> > > Once again, we greatly appreciate the reviewer’s feedback and the positive recognition of the novelties in our work, and are happy to clarify in case of remaining concerns.

---

### Official Review · Reviewer_tuMu · 2025-07-03

**Clarity:** 3
**Significance:** 4
**Originality:** 3
**Rating:** 5
**Confidence:** 3

**Summary:**

The authors introduce a new technique to perform, for the first time, bidirectional attention with Linear Transformers. They create a novel framework (LION) for bidirectional sequence modeling, which supports three types of representations. These different representations present advantages on different fronts: training speed, memory-efficient inference and a balanced mode between the two. Then, LION models using full attention are trained on computer vision and NLP tasks, rivaling the performance of other SSM techniques and vanilla Transformers, while having considerably faster training time and inference speeds.

**Questions:**

- How does LION perform on tasks requiring long-range dependencies (more than 16k tokens)?
- Why is the reported training time of Vim so high compared to ViT (Table 2)? Although they don't report numbers, the authors of Vim affirm that its training is more efficient than ViT's training.
- Why is also the reported training time of Hydra so high compared to the BERT baseline (Table 3)?
- Have you explored custom kernel implementations that could further improve speed and improve memory usage? Have
- Why does Figure 8 (left) not include the Hydra baseline, which is included on the right?

**Ethical Concerns:**

["NO or VERY MINOR ethics concerns only"]

**Final Justification:**

The authors have successfully addressed my questions and suggestions.

Although LION does not achieve state of the art results, this manuscript provides a first step into bidirectional linear attention formulations, which I believe introduces an interesting area of future research on efficient architectures.

**Limitations:**

yes

**Quality:**

4

**Strengths And Weaknesses:**

# Strengths
- **Novel theoretical contribution**: the paper addresses an important gap in the literature by introducing the bilinear formulation to the linear family of transformers. The theoretical analysis is rigorous, and the authors apply their framework to various existing linear transformers and SSMs.
- **Generality**: the framework can extend multiple existing Linear Transformers (RetNet, xLSTM, ...).
- **Comprehensive experiments**: Evaluation across a diverse set of tasks (image classification, masked language modeling, LRA) and model sizes, with a thorough ablation studies examining various design choices.
- **Practical efficiency gains**: the experiments demonstrate significant training speed-ups while maintaining a competitive performance. Moreover, the ability to change between full-attention (fast training), RNN (efficient inference), and chunking (memory-speed tradeoff) is valuable for practical deployment.
- **Clear presentation**: the paper is overall well-written, with clear explanations and helpful visual cues to better understand the notation.
# Weaknesses
- **Limited scalability analysis**: while the authors train models of up to 334M parameters, they don't explore larger scales where the efficiency gains might be more critical. Moreover, there is no analysis on how the model performs with very long sequences (>16k tokens), where these models shine against the softmax attention of the transformer.
- **Comparison fairness**: some baselines (Vim, Hydra) use specialized kernels while LION is implemented in PyTorch without any compilation. This makes speed and memory usage somewhat unfair and understates the potential of optimized implementations.
- **Experimental details**: some important details are relegated to the appendices.
- **Performance-efficiency tradeoff**: while LION achieves significant efficiency gains, it consistently underperforms softmax attention.
- **Scalability unknowns**: the experiments only go up to 334M parameters, which does not prove the scalability of this method to billion-parameter models, widely used in current applications.


### Minor issues
- Line 34: "modeling, namely"
- Should the mask in Eq (4) have a different name to differentiate it with the causal mask?

---

> ### Author Rebuttal · Authors · 2025-07-26
>
> We thank the reviewer for finding **LION framework novel, general and effective in practical scenrios** bellow we will address the questions and concerns mentioned in the review.
>
> --------
>
> ### 1) Scalability of LION to longer sequences and models sizes
>
>
> Due to computational constraints and alignment with prior work [1,2,3], our experiments focus on models with ~100–300M parameters. We expect LION to scale well on non-causal tasks with larger models. Similarly, while we couldn’t train on longer sequences (>16K), **LION’s strong LRA performance** suggests effective generalization to longer inputs.
>
> --------
>
>
> ### 2) Speed comparison fairness
>
> Despite using only PyTorch, **LION still outperforms Hydra and Mamba in speed**, highlighting the efficiency of its formulation. We chose PyTorch to keep LION's implementation simple and easy to integrate. We agree with the reviewer that custom kernels could offer additional speedups and represent a promising direction for future work. LION already achieves faster training even with a pure PyTorch implementation.
>
> --------
>
>
> ### 3) Experimental details
>
> Thank you for the pointer. If there is any specific part of the appendix you would like to see included in the main body, please let us know and we would be happy to incorporate it. Due to space constraints, we focused on key content in the main body. For the final submission, we plan to bring in additional details from the appendix, as the *NeurIPS 2025 Call for Papers* allows **one extra page**.
>
> --------
>
>
> ### 4) Performance-efficiency tradeoff
>
> LION achieves performance close to Transformers on most tasks and even outperforms them at smaller scales, i.e., in image classification with Tiny models (Table 12, Appendix). It also solves LRA tasks from scratch, where Transformers are not able to.
>
> In general SSMs and linear transformers may slightly lag behind Transformers in accuracy, but their inference efficiency and flexibility to switch between attention, RNN, and chunkwise forms make them particularly valuable in practice, this is the core motivation behind LION. We believe that more fine-tuned architectural modifications, such as convolutions over $Q, K$ and additional gates, as used in DeltaNet (ablation of Table 1) [6], could further enhance LION's performance. However, we intentionally used a **simple backbone to isolate and highlight the effect of LION’s attention and sequence mixing block to reflects more its generality across different decays.**
>
> -----
>
> ### 5) Minor Issues
>
> We thank the reviewer for pointing out the minor issues related to typos and sentence clarity. We will address these to improve the overall readability in the final version.
>
> -----
>
> ### 6) Questions
>
> - **1) Performance on longer sequences**
> As mentioned above, we believe LION can scale to longer sequences and larger model sizes; however, due to resource constraints, we were unable to explore this in our current work.
>
> - **2) Long training times of Vim and Hydra**
> Thanks for the sharp observation. The slow training time of Vim, even with custom kernels, is due to the use of parallel scan in Mamba’s [4] training algorithm, which is less efficient on GPUs compared to attention. This issue has been addressed in Mamba2 [5] by using a scalar decay for $A$ and adopting a chunkwise parallel training algorithm (aka SSD), significantly improving training speed (which is also shown in details in Mamba2 blogpost). Hydra, built on Mamba2, benefits from this and trains faster. However, both still **rely on recurrent processing over the state** (also stated in related work of LION and DeltaNet [6]) and are not as fast as attention, which is a key motivation behind LION, which aims to match the training throughput of softmax attention.
>
> - **3) Custom kernel implementation of LION**
> As we addressed this concern in details in concern 4 above, LION is implemented solely in PyTorch. Our goal is to keep LION simple and fast to use within standard PyTorch workflows.
>
> - **4)  Figure 8 (left) does not include the Hydra**
> Thanks for the pointer we will add Hydra to Figure 8 (left) and it also can extrapolate alike LION.
>
>
> -----
>
> ###  References
>
> [1] Hydra: Bidirectional State Space Models Through Generalized Matrix Mixers, Sukjun Hwang, Aakash Lahoti, Ratish Puduppully, Tri Dao, Albert Gu
>
> [2] MosaicBERT: A Bidirectional Encoder Optimized for Fast Pretraining
> Jacob Portes, Alex Trott, Sam Havens, Daniel King, Abhinav Venigalla, Moin Nadeem, Nikhil Sardana, Daya Khudia, Jonathan Frankle
>
> [3] Monarch Mixer: A Simple Sub-Quadratic GEMM-Based Architecture
> Daniel Y. Fu, Simran Arora, Jessica Grogan, Isys Johnson, Sabri Eyuboglu, Armin W. Thomas, Benjamin Spector, Michael Poli, Atri Rudra, Christopher Ré
>
> [4] Mamba: Linear-Time Sequence Modeling with Selective State Spaces
> Albert Gu, Tri Dao
>
> [5] Transformers are SSMs: Generalized Models and Efficient Algorithms Through Structured State Space Duality, Tri Dao, Albert Gu
>
> [6] Parallelizing Linear Transformers with the Delta Rule over Sequence Length", Yang et al. NeurIPS 2024

---

> > ### Comment · Reviewer_tuMu · 2025-08-07
> >
> > We thank the authors for their answer, and apologize for our late response.
> >
> > **Scalability of LION to longer sequences and models sizes**: while we agree that LION shows promising results on sequence lengths of up to 16k tokens, this is well within the context length of even Transformer models with quadratic attention. We could not find an explanation in the manuscript for the ✗ symbol used in the PathX experiments. Is it an OOM error, or is it that the models failed to improve random guessing?
> >
> > **Speed comparison fairness**: acknowledged. We could not find details about the implementation of the Transformer baselines of BERT and DeIT. ViT is mentioned to have a python implementation in the appendix. We believe this information should appear clearly in the main script, since one of the main claims of the paper is the training and inference speeds.
> >
> > **Experimental details**: many of the results are relegated to the appendix, and we believe it would be useful for future readers to bring the most relevant results to the main paper, using the extra page allowed for the camera-ready version.
> >
> > **Performance-efficiency tradeoff**: acknowledged.
> >
> > We thank the authors for their clarifications on our questions, and are satisfied with the replies.

---

> ### Author Response · Authors · 2025-08-07
> **Statement of Thanks**
>
> We thank the reviewer for the positive feedback on our work and for recognizing its **novelty, practical efficiency, and comprehensive experiments**. We also appreciate the thoughtful suggestions, and below we address the remaining points:
>
> --------
>
> 1. **LRA Experiments:**
>    LION solves long-context tasks in LRA, while Transformers fail to improve beyond random guessing, including the Path-X with at 16K sequence length. The $✗$ symbol indicates **failure to improve random guessing**, not an OOM error.
>
> 2. **Speed & ViT Implementation:**
>    Thanks for the pointer. ViT and DeiT baselines follow the official **DeiT \[1] repo and recipe**, using the original Python implementation. We will move these details from the Appendix to the main paper for clarity.
>
> 3. **Experimental Details:**
>    We appreciate the suggestion and fully agree. We will use the **extra camera-ready page** to bring more details regarding implementations and experimental results.
>
>
>
> Finally, we sincerely thank the reviewer for their constructive and positive feedback, which has greatly helped us improve our paper.
>
> -------
>
> ### References
>
> [1] Training data-efficient image transformers & distillation through attention:
> Hugo Touvron, Matthieu Cord, Matthijs Douze, Francisco Massa, Alexandre Sablayrolles, Hervé Jégou

---

### Official Review · Reviewer_GQMH · 2025-07-22

**Clarity:** 3
**Significance:** 3
**Originality:** 2
**Rating:** 5
**Confidence:** 3

**Summary:**

The authors introduce LION, a bidirectional extension of linear transformers. To do so, they go beyond the causality preserving formulation of the attention masking matrix $M$ to adapt it to the setting where entire sequences are available at both training and inference. The decay terms in $M$ allow for an elegant triangular decomposition that lends itself to efficient computation. The attention framework formulation is then translated into its recurrent equivalent (similar idea to the unidirectional causal transformer-to-RNN). Three variations of the approach can be trained, depending on the decay terms selection scheme. Extensive experiments on image classification, masked language modeling and a LRA task, present tradeoffs between training and inference time, performance, GPU and memory costs, etc.

**Questions:**

See Weaknesses, I have incorporated questions in points brought up to avoid repetition.

**Ethical Concerns:**

["NO or VERY MINOR ethics concerns only"]

**Final Justification:**

The authors have clarified the presentation points raised which ensure the claims and main contributions of the paper are clear.

The discussion on the extension of the bidirectional rnn formulation is valuable in showing that the approach is valid beyond the diagonal case, without suffering from additional compute constraints beyond those that arise with full attention models.

**Limitations:**

The authors fail to properly address limitations, notably be it on the architectures that fit in the Lion framework (line 87 mention on TC$^0$ class, what are the models outside of that? Would something like a Delta Net fail to be represented ["Parallelizing Linear Transformers with the Delta Rule over Sequence Length", Yang et al. NeurIPS 2024] and variants])

**Paper Formatting Concerns:**

Looks good

**Quality:**

4

**Strengths And Weaknesses:**

## Strengths

- **Clarity:** Though dizzying at first, the colors and blocks all over end up being useful for the visualization of the ideas behind the math. Anyhow, the authors clearly state the goal of the work and its scope.
- **Significance:** The problem raised is of interest especially to practitioners, as it allows first an introduction of a framework for attention/SSM duality in bidirectional models as well as a better understanding of advantages and tradeoffs in this regime.
- **Technical quality:** Though I have not checked all the derivations, the work is sound. The choice of decay terms to match popular architectures of different inspirations does show the unifying setup the framework claims to be (though it opens questions on whether other design choices could in turn go beyond the initial inspirations).
- **Rigor:** The work is quite comprehensive and self contained. I commend the authors for the effort of compiling an extensive appendix to complement most details that do not fit in the main text.

---

## Weaknesses

- **Design of decay terms:** Following up on the last strength. Given that performance is competitive and albeit better than vanilla ViT but not convincingly superior to other baselines, the question of extending the design search arises. I also find it surprising that using fixed decay terms seems to perform no worse than parametrizing and learning the decay terms (D vs S). Do the authors have an understanding or intuition about these differences (or lack of) in performance.

- **Experiment takeaways:** It is not immediately clear what we should be taking away from the experiments. More specifically:
1) The first experiment, figure by table 2: if the argument is faster training speed, then Lion-LIT is indeed faster than DeiT, yet performs significantly worse. The latter is then the second fastest model to train while still outperforming D and S (base at least, matching on small) The case for Lion is mitigated here.
2) Figure 1 shows tradeoffs between attention, RNN, and chunking approaches for inference. This is a generic tradeoff I suppose and not related to the bidirectional formulation as it would similarly hold in other cases (correct me if I'm wrong). I fail to see why it is relevant to the findings of the paper as a result and not as a motivation, given that the whole point of having both attention and RNN representations is to be able to move from one to the other where advantageous.
3) All in all I fail to gather a clear conclusion from the experiments. Is it better performing, faster, more efficient than the state of the art? Any or none? Is the contribution mainly in the novel formulation?

---

> ### Author Rebuttal · Authors · 2025-07-26
>
> We thank the reviewer for their detailed feedback and for appreciating our work, particularly noting its **rigor, technical quality, and clarity**. Below, we address the main concerns raised.
>
> -------
>
> ### 1) Design of decay terms
>
> We like to clarify that, both the selective and fixed decay factors are learned during training: the fixed decay uses `non-input dependent` $\lambda = \sigma(a)$ with a learned scalar $a$, while the selective decay is `input-dependent`, defined as $\lambda_i = \sigma(Wx_i)$ with learned weights $W$.
>
> Regarding performance, we find the choice of decay factor to be **task-dependent**. The selective variant (LION-S) generally outperforms the fixed variant (LION-D), by **0.25% on masked language modeling** (Table 3) and **0.44% on LRA** (Table 4),likely due to its greater representational power and inductive bias. In contrast, LION-D performs slightly better in image classification, where fixed decay appears sufficient to capture relative token distances in signal and image modalities [1].
>
> As the reviewer noted, different linear transformers use different decay parameterizations, as shown in **Table 1** of our paper and **Table 2** of DeltaNet [2]. For LION, we selected three diverse decay forms to demonstrate the generality of our framework. We believe each decay choice introduces different inductive biases, making them suitable for different tasks, for example, DeltaNet performs well on retrieval. We believe that evaluating different decay choices is a valuable direction for future work.
>
> -------
>
> ### 2) Experiment takeaways
>
> **DeiT is second fastest while outperforming LION D and S**
>
> The main takeaway from Table 2 is that LION, and more broadly, linear transformers are competitive with softmax-based transformers like DeiT in both training speed and accuracy. However, they offer a significant advantage in **inference efficiency**. As shown in Theorem 3.1, LION can be expressed in RNN form, enabling linear-time inference ($O(L)$) compared to the quadratic complexity ($O(L^2)$) of softmax transformers. This is illustrated in Figure 2 and 8.
>
> We agree that Table 2 alone may not fully capture LION's advantage over softmax transformers. Since we aim to highlight three key aspects, 1) accuracy, 2) memory usage, and 3) training time, presenting all of them in a single plot would reduce clarity. Therefore, we separated them: **Table 2** focuses on performance and training speed, while **Figure 2** emphasizes inference memory efficiency and scalability. Together, they demonstrate that LION **combines the strengths of transformers in speed** and accuracy with the **inference efficiency of RNNs.**
>
> **Chunking Tradeoff is Generic; Figure 1 Serves as Motivation Rather Than a Result**
>
> We agree with the reviewer that the tradeoff between RNNs, chunking, and attention is generic, which also applies to bidirectional models, as noted in the background section of our paper. Figure 1 is primarily intended to motivate the value of supporting multiple representations depending on resource constraints. Thanks to the reviewer’s suggestion, we will move this figure to Section 3, where it better fits as a motivation.
>
>
> **Final Conclusion from Experiment Section**
>
> Our main contribution, as noted by the reviewers (particularly in the second point), is the **novel formulation of LION** as the  framework for extending linear transformers to the bidirectional setting while supporting and analyzing tradeoffs across different regimes.
>
> Our experiments shows that LION works across multiple modalities. Key takeaways of experiments include:
>
> - LION achieves training speed and accuracy close to softmax transformers (Table 1), *while being significantly more efficient at inference*.
>
> - LION matches or exceeds the inference efficiency of SSMs (Figure 2), *while training much faster*.
>
> ------
>
> ### 3) Discussion on DeltaNet
>
> We appreciate the reviewer's sharp observation on this point. Our theorem is specifically focused on linear transformers with **diagonal decay**, and models outside this class **generally cannot be expressed in the full attention formulation**. However, for the special case of **DeltaNet**, our theorem supports an equivalent **bidirectional RNN** formulation with LION-style correction terms to avoid double counting, as follows:
>
> $S_i^{F/B} = S_{i-1}^{F/B}(I - \beta_i k_i k_i^\top) + \beta_i k_i v_i^\top, \quad y_i^{F/B} = q_i^\top S_i^{F/B} - \frac{1}{2} q_i^\top k_iv_i^\top$
>
> The chunkwise parallel form of DeltaNet (Eqs. 8–11 of the DeltaNet paper) can also be extended to the bidirectional setting by removing causal masks and applying Theorem 3.2 for the chunkwise form of LION.
>
> However, a key bottleneck in DeltaNet arises in its **full attention formulation**. As described in the original paper (page 6, section “Fully Parallel Form for DeltaNet”), the attention is computed as:
>
> $A = QK^\top T$
>
> Computing the matrix $T$ (defined in equation 10 of DeltaNet) is costly, which directly citing from paper:  **"$T$ requires a matrix inverse (Eq. 10), which scales cubically with sequence length without further algorithmic changes.  Due to the above we avoid using the fully parallel form
> for training DeltaNet"**.  The same limitation applies in the bidirectional case, where the inverse undermines the fast-training advantage of LION.
>
> We thank the reviewer for raising this point and will include the discussion, along with a note on models outside $\text{TC}^0$ with non-diagonal decay, in the limitations section.
>
> -----
>
> ### References
>
> [1] Mamba: Linear-Time Sequence Modeling with Selective State Spaces Albert Gu, Tri Dao
>
> [2] Parallelizing Linear Transformers with the Delta Rule over Sequence Length", Yang et al. NeurIPS 2024]

---

> > ### Comment · Reviewer_GQMH · 2025-08-05
> > **Follow up note**
> >
> > I would like to thank the authors who have convincingly addressed the points raised.
> >
> > My score has been improved accordingly.

---

> > > ### Author Response · Authors · 2025-08-05
> > > **Statement of Thanks**
> > >
> > > We sincerely thank the reviewer for their thoughtful questions and constructive feedback. We also truly appreciate the increased score and are glad to hear that our work resonated positively with the reviewer.
> > >
> > > Best regards,
> > >
> > > The Authors

---

### Author Response · Authors · 2025-08-09
**Global Response and Summary of Rebuttal**

## Statement of Thanks

We sincerely thank the reviewers for their valuable feedback and constructive suggestions. We are pleased that our work was recognized as **novel, rigorous, well-written**, and the first to provide a detailed study of **linear transformers for bidirectional sequence modeling**.

---

## Summary of Main Contributions and Takeaways

LION is a general framework for mapping several linear transformers into their bidirectional form, with the following key goals:

1. **Generality** – Supports mapping multiple linear transformer architectures to their bidirectional counterparts (see Table 1).
2. **Efficiency in Inference** – **Matches SSMs and RNNs in inference efficiency**, remaining linear in complexity unlike Transformers which are quadratic.
3. **Faster Training** – Address slow training of SSMs and trains **significantly faster than existing SSMs** by using fully parallel linear attention.
4. **Speed–Memory Balance** – Uses **chunkwise parallelism** to optimize the speed–memory trade-off (see LION-chunk section 3.3 and Figure 1).

-----

## New Results and Clarifications During Rebuttal

1. **MLM with Longer Sequences** – Extended MLM experiments to **512 tokens, showing LION improves in accuracy with longer contexts** and increases its **training speed** advantage over Hydra from **~3× to ~5×**.
2. **Hydra with Multiple Scans** – Applied multiple scan strategy to Hydra and found Hydra with multiple scans still underperforms LION-D at small scale, shows no improvement at base scale, and suffers a notable training speed drop.
3. **Mapping Non-Diagonal Linear Transformers** – Demonstrated how models like **DeltaNet** can be mapped to a bidirectional setting via LION’s formulation.
4. **Balanced vs. Imbalanced Attention** – Clarified the the insight and confirmed experimentally that LION’s **balanced attention significantly outperforms imbalanced attention** (see rebuttal to Reviewer zmVf).
5. **Optimal Chunk Size** – Provided guidelines for choosing chunk size for chunkwise parallelism in **Appendix C.16**.
6. **Importance of Bidirectionally** – Shown in **Appendix C.15** that LION consistently outperforms alternative sequence mixing and unidirectional variants.
7. **Motivation for Bidirectionally** – Added **stronger motivation and references for bidirectional modeling** in the introduction (see rebuttal to Reviewer zmVf).
8. **Efficiency Clarifications** – Clarified that LION in RNN form is more efficient than softmax attention like FlashAttention, and explained why KV-cache offers little benefit for bidirectional models.
9. **New Presentations** – Moving key appendix content—such as **chunk size effects, ViT and LION implementation details, additional experiment results, and bidirectionally discussion**—into the main body for the camera-ready version.

---

## More Details on our Experimental Setup

### Masked Language Modeling (MLM)
LION was evaluated at two scales (Base, Large) and two sequence lengths (128, 512) using the M2-BERT setup, as an standard recipe for linear transformers which is also used in Hydra, to ensure a fair comparison. Across these settings, LION matches or closely approaches the performance of Transformers and SSMs, while being more inference-efficient and **significantly faster to train than SSMs like Hydra**.
### ImageNet Classification
We evaluated LION at three model sizes, **Tiny, Small, and Base**. LION outperforms SSMs at smaller scales and remains competitive at larger scales, all while training up to **3× faster than Hydra** and **over 10× faster than Vim** and significantly more efficient than transformers in inference.
### Long Range Arena (LRA)
On LRA, LION successfully solve long-context tasks, **while Transformers fail to surpass random guessing**, demonstrating LION’s stability and effectiveness for long-range sequence modeling.

-----

We greatly appreciate the reviewers’ constructive engagement, which has helped us improve the clarity, experimental depth, and presentation of the paper. Our results confirm that LION delivers strong performance, scalability, and efficiency across diverse tasks, while providing a unifying theoretical framework for bidirectional linear transformers. We believe our theoretical framework and experimental results demonstrate that LION is both effective and applicable, enabling many linear transformers to be extended to bidirectional sequence modeling in future work.

---

### Note · Authors · 2025-08-13

Dear AC and Reviewers,

We thank the reviewers for their constructive feedback and the AC for their time and effort in handling our paper. We are grateful that our work was recognized as **novel, general, and clear**, and for its focus on **the important challenge of bidirectionality**. As we have already provided a **Global Response and Summary of Rebuttal**, we encourage you to refer to it for full details. Below, we briefly highlight key points from the rebuttal.

---
## Main Contribution
As noted by reviewers, LION is the first general framework for mapping linear transformers to a bidirectional setting, addressing the slow training of SSMs. It trains **significantly faster** than SSMs, achieves **more efficient inference** than Transformers, and maintains competitive performance.

## Additional Results
* **MLM with longer sequences:** Extending to 512 tokens improved LION’s accuracy and expanded its speed advantage over Hydra from ~3× (128 tokens) to ~5× (512 tokens), showing LION’s speed advantage grows with longer contexts (Reviewer 1Uqp).
* **Hydra with multiple scans:** Shows little to no improvement with multiple scans and slows substantially, becoming ~5× slower than LION in this setting (Reviewer zmVf).
* **LION for non-diagonal linear transformers:** Showing how models like DeltaNet can be mapped to a bidirectional setting using our theorem (Reviewer GQMH).
* **Balanced vs. imbalanced attention:** LION’s balanced attention consistently outperforms the imbalanced variant (Table, follow-up Reviewer zmVf).

## Integrating Key Appendix Results
In the camera-ready version, we will move key results into the main body:
1. Effect of chunk size (C.16)
2. Transformers implementation details (C.5)
3. Importance of bidirectionality vs. unidirectional/alternating methods (C.15)
4. Clearer representation suggested by Reviewer zmVf, with evidence that LION’s balanced attention outperforms the imbalanced variant

## Clarifications
We addressed key points such as LION’s efficiency vs. Transformers with FlashAttention, details of our MLM setup following Hydra’s M2 recipe within our budget constraints, and implementation specifics for LION.

---
We believe our framework and results show that **LION is both effective and broadly applicable**, enabling many linear transformers to adopt its core for future bidirectional sequence modeling. We are glad our responses were *generally found satisfactory* and that they clarified our contributions.

Best regards,

Authors

---

### Decision · Program_Chairs · 2025-09-17

**Decision:**

Accept (poster)

**Comment:**

This paper proposes LION, a framework for generalizing (gated) linear attention models to the bidirectional setting (similar to Hydra [Hwang et al. 2024], which generalizes SSMs to the bidirectional case). Across language/vision applications, LION is generally found to be more efficient than Hydra, at the cost of slight performance degradations.

The approach is sensible, clearly explained, and supported by a suite of experiments across different domains. The efficiency improvements over Hydra is particularly strong, despite the fact that LION does not make use of highly specialized kernels. While training efficiency compared to Transformers is not always favorable, the inference efficiency gains still make the approach worthwhile.

On the negative side, the technical novelty is somewhat limited (it's a pretty straightforward generalization of unidirectional linear attention models), and the benchmarks (GLUE/LRA) are a bit outdated. But I still think this is a worthwhile contribution to the community.